# Universally Optimal Watermarking Schemes for LLMs: from Theory to Practice

## Abstract

Large Language Models (LLMs) boosts human efficiency but also poses misuse risks, with watermarking serving as a reliable method to differentiate AI-generated content from human-created text. In this work, we propose a novel theoretical framework for watermarking LLMs. Particularly, we jointly optimize both the watermarking scheme and detector to maximize detection performance, while controlling the worst-case Type-I error and distortion in the watermarked text. Within our framework, we characterize the *universally minimum Type-II error*, showing a fundamental trade-off between detection performance and distortion. More importantly, we identify the optimal type of detectors and watermarking schemes. Building upon our theoretical analysis, we introduce a practical, model-agnostic and computationally efficient token-level watermarking algorithm that invokes a surrogate model and the Gumbel-max trick. Empirical results on Llama-13B and Mistral-8×7B demonstrate the effectiveness of our method. Furthermore, we also explore how robustness can be integrated into our theoretical framework, which provides a foundation for designing future watermarking systems with improved resilience to adversarial attacks.

## 1 Introduction

Arising with Large Language Models (LLMs) (Touvron et al., 2023) that demonstrate stunning power are substantial risks: spreading disinformation, generating fake news, engaging in plagiarism, etc. Such risks elevate as LLMs are increasingly widely adopted for content generation. Distinguishing AI-generated content from human-written text is then critically demanded and watermarking serves as an effective solution to address this challenge.

Existing watermarking techniques for AI-generated text can be classified into two categories: post-process and in-process. Post-process watermarks (Brassil et al., 1995; Yoo et al., 2023; Yang et al., 2023; Munyer et al., 2023; Yang et al., 2022; Sato et al., 2023; Zhang et al., 2024; Abdelnabi & Fritz, 2021) are applied after the text is generated, while in-process watermarks (Wang et al., 2023; Fairoze et al., 2023; Hu et al., 2023; Huo et al., 2024; Zhang et al., 2023; Tu et al., 2023; Ren et al., 2023) are embedded during generation. Between the two types, in-process watermarking is more favorable due to its flexibility and numerous techniques have been proposed to seamlessly integrate watermarks into the generation process. Notably, an ideal in-process watermarking scheme for LLMs should have four desired properties: 1) **Detectability**: the watermarking can be reliably detected with Type-I error controlled; 2) **Distortion-free** (Christ et al., 2024; Kuditipudi et al., 2023): the watermarked text preserves the quality of the original generated text by maintaining the original text distribution; 3) **Robustness** (Zhao et al., 2023; Liu & Bu, 2024): the watermark is resistant to modifications aimed at removing it from the watermarked text; 4) **Model-agnostic** (Kirchenbauer et al., 2023a): detection does not require knowledge of the original watermarked LLMs or the prompts. Clearly, one expects a tension between these dimensions. Yet, despite the great efforts in designing watermarking and detection schemes that heuristically balance these factors, theoretical understanding of the fundamental trade-offs therein is rather limited to date.

Among existing theoretical analyses, Huang et al. (2023) frames statistical watermarking as an independent testing problem between text and watermark and analyzes the optimal watermarking scheme for a specific detection process. While their analysis can be extended to a model-agnostic setting, they do not propose a practical algorithm. In contrast, Li et al. (2024) proposes a surrogate hypothesis testing framework based on i.i.d. pivotal statistics, with the goal of identifying statistically optimal detection rules for a given watermarking scheme. However, their method depends on a

suitably chosen watermarking scheme without optimizing it, and the proposed detection rule is not necessarily optimal for the original independence testing problem. While these studies provide useful insights, they fall short of capturing the optimal watermarking scheme and detection rule, limiting their practicality and effectiveness in real-world applications.

In this paper, we formulate the LLM watermarking problem as an independence test between the text and an auxiliary variable to jointly optimize *both* the watermarking scheme and the detector, achieving universal optimality. Our theoretical framework characterizes the fundamental trade-off between detectability, distortion, and robustness by minimizing Type-II error. To capture the **detectability**, we define universal optimality in two aspects: 1) controlling the false alarm rate across *all* possible text distributions to ensure worst-case Type-I error performance, and 2) designing a universally optimal detector that remains effective across *all* text distributions. Additionally, we measure the **distortion** of a watermarked LLM using the divergence between the watermarked text distribution and the original text distribution. **Robustness**, on the other hand, depends on modifications to the watermarked text, such as replacement, deletion/insertion, or paraphrasing. Unlike existing approaches that evaluate robustness via experiments (Liu & Bu, 2024) or provide detection error bounds under specific modifications (Kuditipudi et al., 2023; Zhao et al., 2023), our framework covers a broader range of potential attacks, including those that preserve the semantics of the text.

Our contributions can be summarized as follows:

- We characterize the universally minimum Type-II error, representing the best achievable detection performance, which reveals a fundamental trade-off between detection performance and distortion. More importantly, we characterize the entire class of detectors and watermarking schemes that are universally optimal, meaning no other type can achieve the same performance.

- To balance theoretical guarantees and practical implementation, we propose a practical token-level watermarking scheme that ensures a small Type-II error (exponentially decaying under certain conditions) while controlling worst-case Type-I error. It also shows inherent robustness against token replacement attacks. The corresponding algorithm, leveraging a surrogate language model and the Gumbel-max trick, is both *model-agnostic* and *computationally efficient*.

- We conduct extensive experiments over various language models (Llama2-13B (Touvron et al., 2023), and Mistral-8×7B (Jiang et al., 2023)) on multiple datasets. Comparisons with baseline methods show the effectiveness of our algorithm, even under token replacement attacks.

- Lastly, we explore how to incorporate *robustness* against semantic-invariant attacks into our theoretical framework, providing insights for designing optimal semantic-based watermarking systems that are robust to such attacks in the future.

Proofs and additional analyses are deferred to Appendix.

**Other Related Literature.** The advancement of LLMs boosts productivity but also presents challenges like bias and misuse, which watermarking addresses by tracing AI-generated content and distinguishing it from human-created material. Currently, many watermarking methods for LLMs are proposed (Zhou et al., 2024; Fu et al., 2024; Giboulot & Teddy, 2024; Wu et al., 2023; Kirchenbauer et al., 2023b), including biased and unbiased (distortion-free) watermarking. Biased watermarks (Kirchenbauer et al., 2023a; Zhao et al., 2023; Liu & Bu, 2024; Liu et al., 2024; Qu et al., 2024) typically alter the next-token prediction distribution slightly, thereby increasing the likelihood of sampling certain tokens. For example, Kirchenbauer et al. (2023a) proposes to divide the vocabulary into green and red lists, slightly enhancing the probability of green tokens in the next token prediction (NTP) distribution. Unbiased watermarks (Zhao et al., 2024; Fernandez et al., 2023; Boroujeny et al., 2024; Christ et al., 2024) maintain the original NTP distributions unchanged, using various sampling strategies to embed watermarks. The Gumbel-max watermark (Aaronson, 2023) utilizes the Gumbel-max trick (Gumbel, 1954) for sampling the NTP distributions, while Kuditipudi et al. (2023) introduces an inverse transform method for this purpose.

Most existing watermarking schemes and detectors are heuristic and lack theoretical support. Traditional post-process watermarking schemes, which apply watermarks after generation, have been extensively studied from information-theoretic perspective (Martinian et al., 2005; **?**; Chen, 2000; Merhav & Ordentlich, 2006; Merhav & Sabbag, 2008). For in-process watermarking, while two prior works attempt to derive theoretically optimal schemes or detectors, their solutions are either not jointly optimized or lacked universal optimality as achieved in our paper. Huang et al. (2023)

designs an optimal watermarking scheme for a specific detector, but their detector is not model-non-agnostic, requiring the original NTP distributions of the watermarked LLM, with experiments also conducted under this assumption. Li et al. (2024) proposes detection rules using pivotal statistics, formulating the task as a minimax optimization, but their detector's optimality depends on the assumption that the pivotal statistics are i.i.d.. In contrast, we propose a framework that jointly optimizes the watermarking scheme and detector for an optimal configuration of both components.

## 2 PRELIMINARIES AND PROBLEM FORMULATION

**Notations.** For any set $\mathcal{X}$, we denote the space of all probability measures over $\mathcal{X}$ by $\mathcal{P}(\mathcal{X})$. For a random variable taking values in $\mathcal{X}$, we often use $P_X$ or $Q_X$ to denote its distribution, and use the lower-cased letter $x$ to denote a realization of $X$. For a sequence of random variables $X_1, X_2, \ldots, X_n$, and any $i, j \in [n]$ with $i \leq j$, we denote $X_i^j := (X_i, \ldots, X_j)$. We may use distortion function, namely, a function $\mathsf{D} : \mathcal{P}(\mathcal{X}) \times \mathcal{P}(\mathcal{X}) \to [0, +\infty)$ to measure the dissimilarity between two distributions in $\mathcal{P}(\mathcal{X})$. For example, the total variation distance, as a notion of distortion, between $\mu, \nu \in \mathcal{P}(\mathcal{X})$ is $\mathsf{D}_{\mathsf{TV}}(\mu, \nu) := \int \frac{1}{2} |\frac{d\mu}{d\nu} - 1| \, d\nu$. For any set $A \subseteq \mathcal{X}$, we use $\delta_A$ to denote its characteristic function, namely, $\delta_A(x) := \mathbb{1}\{x \in A\}$. Additionally, we denote $(x)_+ := \max\{x, 0\}$ and $x \wedge y := \min\{x, y\}$.

**Hypothesis Testing Framework for Watermark Detection.** LLMs process text through "tokenization", namely, breaking it down into words or word fragments, called "tokens". An LLM generates text token by token. Specifically, let $\mathcal{V}$ denote the token vocabulary, typically of size $|\mathcal{V}| = \mathcal{O}(10^4)$ (Liu, 2019; Radford et al., 2019; Zhang et al., 2022; Touvron et al., 2023). An *unwatermarked* LLM generates the next token $X_t$ based on a prompt $u$ and the previous tokens $x_1^{t-1}$ by sampling from a distribution $Q_{X_t|x_1^{t-1},u}$, referred to as the next-token prediction (NTP) distribution at position $t$. For simplicity, we will suppress the dependency of the generated tokens on the prompt $u$ in our notation throughout the paper. The joint distribution of a length-$T$ generated token sequence $X_1^T$ is then given by $Q_{X_1^T}(\cdot) := \prod_{t=1}^T Q_{X_t|X_1^{t-1}}(\cdot|\cdot)$, which we assume to be identical to one that governs the human-generated text.

*Watermarking LLM.* We consider a general formulation of watermarking schemes for LLMs, where the construction of the NTP distribution for the *watermarked* LLM exploits an auxiliary random sequence, as shown in Figure 1. Specifically, associated with each token position $t$, there is a random variable $\zeta_t$ taking values in some space $\mathcal{Z}$ (either discrete or continuous). The NTP distribution for the watermarked LLM is now in the form of (and denoted by) $P_{X_t|x_1^{t-1},\zeta_1^t}$, from which $X_t$ is sampled. The resulted joint distribution of the watermarked sequence $X_1^T$ is denoted by $P_{X_1^T}$. The joint structure of sampling $\zeta_1^T$ and the new NTP distribution $P_{X_t|x_1^{t-1},\zeta_1^t}$, i.e., the joint distribution $P_{X_1^T,\zeta_1^T}$, characterizes a "watermarking scheme". Here, we assess the *distortion level* of a watermarking scheme by measuring the statistical divergence between the watermarked distribution $P_{X_1^T}$ and the original distribution $Q_{X_1^T}$. Examples of such divergences include squared distance, total variation, KL divergence, and Wasserstein distance.

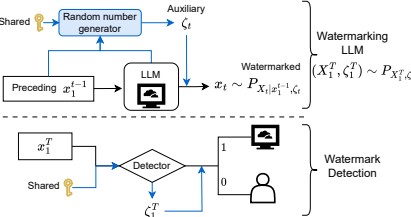

Figure 1: Generic framework of LLM watermarking and detection.

**Definition 1** ($\epsilon$-distorted watermarking scheme). *A watermarking scheme is $\epsilon$-distorted with respect to distortion $\mathsf{D}$, if $\mathsf{D}(P_{X_1^T}, Q_{X_1^T}) \leq \epsilon$, where $\mathsf{D}$ can be any distortion metric.*

Notably, the auxiliary sequence $\zeta_1^T$ is usually sampled using a shared `key` accessible during both watermarked text generation and watermark detection. Our formulation here allows it to take an arbitrary structure, which contrasts the rather restricted i.i.d. assumption considered in Li et al. (2024, Working Hypothesis 2.1). As shown in the following example, existing watermarking schemes, to the best of our knowledge, may all be seen as special cases of this formulation.

**Example 1** (Existing watermarking schemes). *In Green-Red List watermarking scheme (Kirchenbauer et al., 2023a), at each position $t$, the vocabulary $\mathcal{V}$ is randomly split into a green list $\mathcal{G}$ and a red list $\mathcal{R}$, where $|\mathcal{G}| = \rho|\mathcal{V}|$ for some $\rho \in (0, 1)$. The splitting can then be represented by a $|\mathcal{V}|$-dimensional binary auxiliary variable $\zeta_t$, indexed by $x \in \mathcal{V}$, where $\zeta_t(x) = 1$ means $x \in \mathcal{G}$; otherwise, $x \in \mathcal{R}$. The watermarking scheme is as follows. For $t = 1, 2, \ldots,$*

*– Compute a hash of the previous token $X_{t-1}$ using a hash function $h : \mathcal{V} \times \mathbb{R} \to \mathbb{R}$ and a shared secret `key`, i.e., $h(X_{t-1}, \texttt{key})$.*

- *Use $h(X_{t-1}, \mathtt{key})$ as a seed to uniformly sample the auxiliary variable $\zeta_t$ from the set $\{\zeta \in \{0,1\}^{|\mathcal{V}|} : \|\zeta\|_1 = \rho|\mathcal{V}|\}$ to construct the green list $\mathcal{G}$.*
- *Sample $X_t$ from the following NTP distribution which increases the logit of tokens in $\mathcal{G}$ by $\delta > 0$:*

$$P_{X_t | x_1^{t-1}, \zeta_t}(x) = \frac{Q_{X_t | x_1^{t-1}}(x) \exp(\delta \cdot \mathbb{1}\{\zeta_t(x)=1\})}{\sum_{x \in \mathcal{V}} Q_{X_t | x_1^{t-1}}(x) \exp(\delta \cdot \mathbb{1}\{\zeta_t(x)=1\})}.$$

*Several other watermarking schemes are also presented in Appendix A.*

*Watermark Detection.* When $X_1^T$ is generated by a watermarked LLM, it depends on $\zeta_1^T$, while human-generated $X_1^T$ and $\zeta_1^T$ are independent. Therefore, given a pair of sequences $(X_1^T, \zeta_1^T)$, the detection task boils down to discriminating between the following two hypotheses:

- H$_0$: $X_1^T$ is generated by a human, i.e., $(X_1^T, \zeta_1^T) \sim Q_{X_1^T} \otimes P_{\zeta_1^T}$;
- H$_1$: $X_1^T$ is generated by a watermarked LLM, i.e., $(X_1^T, \zeta_1^T) \sim P_{X_1^T, \zeta_1^T}$.

We consider a model-agnostic detector $\gamma : \mathcal{V}^T \times \mathcal{Z}^T \to \{0,1\}$, which maps $(X_1^T, \zeta_1^T)$ to the index of one of the two hypotheses. In our theoretical analysis, we assume the auxiliary sequence $\zeta_1^T$ can be fully recovered from $X_1^T$ and the shared $\mathtt{key}$. This assumption is however dropped in our practical implementation.

Performance is measured by Type-I (false alarm) and Type-II (missed detection) error probabilities:

$$\beta_0(\gamma, Q_{X_1^T}, P_{\zeta_1^T}) := \Pr(\gamma(X_1^T, \zeta_1^T) \neq 0 \mid \mathrm{H}_0) = (Q_{X_1^T} \otimes P_{\zeta_1^T})(\gamma(X_1^T, \zeta_1^T) \neq 0),$$

$$\beta_1(\gamma, P_{X_1^T, \zeta_1^T}) := \Pr(\gamma(X_1^T, \zeta_1^T) \neq 1 \mid \mathrm{H}_1) = P_{X_1^T, \zeta_1^T}(\gamma(X_1^T, \zeta_1^T) \neq 1). \tag{1}$$

When $Q_{X_1^T}$ and $P_{X_1^T, \zeta_1^T}$ are fixed, it is well-known that the optimal detector is a likelihood-ratio test (Cover & Thomas, 2006). However, this is a non-model-agnostic detector, as it requires the knowledge of $Q_{X_1^T}$. In contrast, in our setting, the watermarking scheme $P_{X_1^T, \zeta_1^T}$ is to be designed and both $Q_{X_1^T}$ and $P_{X_1^T, \zeta_1^T}$ are unknown to the detector.

Furthermore, since humans can generate texts with arbitrary structures, we must account for controlling Type-I error across all possible distributions $Q_{X_1^T}$. Therefore, our goal is to jointly design an $\epsilon$-distorted watermarking scheme and a model-agnostic detector that minimizes the Type-II error while ensuring the *worst-case* Type-I error $\sup_{Q_{X_1^T}} \beta_0(\gamma, Q_{X_1^T}, P_{\zeta_1^T})$ under a constant $\alpha \in (0,1)$.

Specifically, the optimization problem is:

$$\inf_{\gamma, P_{X_1^T, \zeta_1^T}} \beta_1(\gamma, P_{X_1^T, \zeta_1^T}) \quad \text{s.t.} \quad \sup_{Q_{X_1^T}} \beta_0(\gamma, Q_{X_1^T}, P_{\zeta_1^T}) \leq \alpha, \ \mathrm{D}(P_{X_1^T}, Q_{X_1^T}) \leq \epsilon. \tag{Opt-O}$$

The optimal objective value is the *universally minimum Type-II error*, denoted by $\beta_1^*(Q_{X_1^T}, \alpha, \epsilon)$.

## 3  JOINTLY OPTIMIZE WATERMARKING SCHEME AND DETECTOR

Solving the optimization in (Opt-O) is challenging due to the binary nature of $\gamma$ and the vast set of possible $\gamma$, sized $2^{|\mathcal{V}|^T |\mathcal{Z}|^T}$. To address this, we first minimize over $P_{X_1^T, \zeta_1^T}$ with a fixed detector, aiming to uncover a potential structure for the optimal detector.

Consider any model-agnostic detector $\gamma(X_1^T, \zeta_1^T) = \mathbb{1}\{(X_1^T, \zeta_1^T) \in \mathcal{A}_1\}$, where $\mathcal{A}_1 \subseteq \mathcal{V}^T \times \mathcal{Z}^T$ defines the acceptance region for H$_1$. We then rewrite the optimization as:

$$\inf_{P_{X_1^T, \zeta_1^T}} \beta_1(\gamma, P_{X_1^T, \zeta_1^T}) \quad \text{s.t.} \quad \sup_{Q_{X_1^T}} \beta_0(\gamma, Q_{X_1^T}, P_{\zeta_1^T}) \leq \alpha, \ \mathrm{D}(P_{X_1^T}, Q_{X_1^T}) \leq \epsilon. \tag{Opt-I}$$

We first derive a lower bound for the minimum Type-II error in (Opt-I) (and for the Type-II error in (Opt-O)), which is independent of the detector $\gamma$. We then identify a detector and watermarking scheme that achieves this lower bound, indicating that it represents the universally minimum Type-II error. Thus, the proposed detector and watermarking scheme is optimal, as presented in Theorem 2. The theorem below establishes this universally minimum Type-II error for all feasible watermarking schemes and detectors.

**Theorem 1** (Universally minimum Type-II error). *The universally minimum Type-II error attained from* (Opt-O) *is*

$$\beta_1^*(Q_{X_1^T}, \alpha, \epsilon) = \min_{P_{X_1^T}: D(P_{X_1^T}, Q_{X_1^T}) \leq \epsilon} \sum_{x_1^T} (P_{X_1^T}(x_1^T) - \alpha)_+. \tag{2}$$

*By setting* D *as total variation distance* $D_{TV}$, *(2) can be simplified as*

$$\beta_1^*(Q_{X_1^T}, \alpha, \epsilon) = \Big( \sum_{x_1^T} (Q_{X_1^T}(x_1^T) - \alpha)_+ - \epsilon \Big)_+, \quad \text{if } \sum_{x_1^T} (\alpha - Q_{X_1^T}(x_1^T))_+ \geq \epsilon.$$

Theorem 1 shows that, for any watermarked LLM, the fundamental performance limits of watermark detection depend on the original NTP distribution of the LLM. When the original $Q_{X_1^T}$ has lower entropy, the best achievable detection error increases. This hints that it is inherently difficult to detect low-entropy watermarked text. However, by allowing higher distortion $\epsilon$, the watermarked LLM has more capacity to reduce the detection error again. Figure 2 shows an illustration. Moreover, we find that $\beta_1^*(Q_{X_1^T}, \alpha, \epsilon)$ matches the minimum Type-II error from Huang et al. (2023, Theorem 3.2), which is

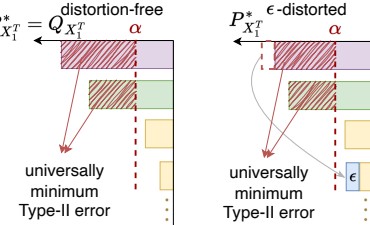

Figure 2: Universally minimum Type-II error w/o distortion

optimal under a specific detector but not universally. Our results demonstrate that $\beta_1^*(Q_{X_1^T}, \alpha, \epsilon)$ is the universally minimum Type-II error for all detectors and watermarking schemes, indicating their detector is within the set of optimal detectors.

**Optimal type of detectors and watermarking schemes.** Since we have established the universally minimum Type-II error, a natural question arises: what is the *optimal type of detectors and watermarking schemes* that achieve this universal minimum (for all $Q_{X_1^T}$ and $\epsilon$)? Let $\Pi^*(Q_{X_1^T}, \alpha, \epsilon)$ denote the set of all solutions $(\gamma^*, P^*_{X_1^T, \zeta_1^T})$ that achieve $\beta_1^*(Q_{X_1^T}, \alpha, \epsilon)$.

**Theorem 2** ((Informal Statement) Optimal type of detectors and watermarking schemes). *The optimal type of detectors is given by*

$$\Gamma^* := \{\gamma \mid \gamma(X_1^T, \zeta_1^T) = \mathbb{1}\{X_1^T = g(\zeta_1^T)\}, \text{ for some surjective } g : \mathcal{Z}^T \to \mathcal{S} \supset \mathcal{V}^T\}.$$

*For any $\gamma^* \in \Gamma^*$ and any $(Q_{X_1^T}, \epsilon)$, the corresponding optimal $\epsilon$-distorted watermarking scheme $P^*_{X_1^T, \zeta_1^T}$ is provided in Appendix D, i.e., $(\gamma^*, P^*_{X_1^T, \zeta_1^T}) \in \Pi^*(Q_{X_1^T}, \alpha, \epsilon)$.*

**Corollary 3** (Universal optimality of detectors $\Gamma^*$). *For any $\gamma \notin \Gamma^*$, there exists $(\tilde{Q}_{X_1^T}, \tilde{\epsilon})$ such that no $\tilde{\epsilon}$-distorted watermarking scheme $P_{X_1^T, \zeta_1^T}$ satisfies $(\gamma, P_{X_1^T, \zeta_1^T}) \in \Pi^*(\tilde{Q}_{X_1^T}, \alpha, \tilde{\epsilon})$.*

Theorem 2 and Corollary 3 suggest that, to guarantee the construction of an optimal watermarking scheme for any arbitrary LLM, the detector must be selected from the set $\Gamma^*$.

Using a toy example in Figure 3, we now illustrate how to construct the optimal watermarking schemes, where

$$P^*_{X_1^T} = \arg\min_{P_{X_1^T}: D(P_{X_1^T}, Q_{X_1^T}) \leq \epsilon} \sum_{x_1^T} (P_{X_1^T}(x_1^T) - \alpha)_+.$$

Constructing the optimal watermarking scheme $P^*_{X_1^T, \zeta_1^T}$ is equivalent to transporting the probability mass $P^*_{X_1^T}$ on $\mathcal{V}$ to $\mathcal{Z}$, maximizing $P^*_{X_1^T, \zeta_1^T}(x_1^T, \zeta_1^T)$ when $x_1^T = g(\zeta_1^T)$, while keeping the worst-case Type-I error below $\alpha$. Without loss of generality, by letting $T = 1$, we present Figure 3 to visualize the optimal watermarking scheme. The construction process is given step by step as follows:

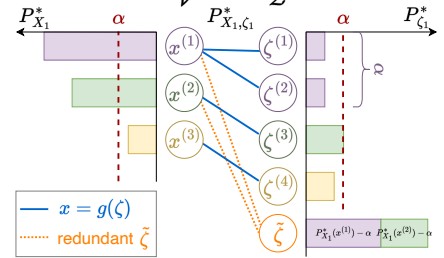

Figure 3: A toy example of the optimal detector and watermarking scheme. Links between $\mathcal{V}$ and $\mathcal{Z}$ suggest $P^*_{X_1, \zeta_1} > 0$.

– **Identify text-auxiliary pairs:** We begin by identifying text-auxiliary pairs $(x, \zeta) \in \mathcal{V} \times \mathcal{Z}$ with $\gamma(x, \zeta) = \mathbb{1}\{x = g(\zeta)\} = 1$ and connect them by blue solid lines.

– **Introducing redundant auxiliary value:** We enlarge $\mathcal{Z}$ to include an additional value $\tilde{\zeta}$ and set $\gamma(x, \tilde{\zeta}) = 0$ for all $x$. We will call $\tilde{\zeta}$ "redundant".

– **Mass allocation for** $P^*_{X_1}(x) > \alpha$**:** If $P^*_{X_1}(x) > \alpha$, we transfer $\alpha$ mass of $P^*_{X_1}(x)$ to the $\zeta$ connected by the blue solid lines. The excess mass is transferred to the redundant $\tilde{\zeta}$ (orange dashed lines). Specifically, for $x^{(1)}$, where $P^*_{X_1}(x^{(1)}) > \alpha$ and $x^{(1)} = g(\zeta^{(1)}) = g(\zeta^{(2)})$, we move $\alpha$ units of mass from $P^*_{X_1}(x^{(1)})$ to $P^*_{\zeta_1}(\zeta^{(1)})$ and $P^*_{\zeta_1}(\zeta^{(2)})$, ensuring that $P^*_{\zeta_1}(\zeta^{(1)}) + P^*_{\zeta_1}(\zeta^{(2)}) = \alpha$. The rest $(P^*_{X_1}(x^{(1)}) - \alpha)$ units of mass is moved to $\tilde{\zeta}$. Similarly, for $x^{(2)}$, where $P^*_{X_1}(x^{(2)}) > \alpha$ and $x^{(2)} = g(\zeta^{(3)})$, we move $\alpha$ mass from $P^*_{X_1}(x^{(2)})$ to $P^*_{\zeta_1}(\zeta^{(3)})$ and $(P^*_{X_1}(x^{(2)}) - \alpha)$ mass to $\tilde{\zeta}$. Consequently, the probability of $\tilde{\zeta}$ is $P_{\zeta_1}(\tilde{\zeta}) = (P^*_{X_1}(x^{(1)}) - \alpha) + (P^*_{X_1}(x^{(2)}) - \alpha)$. In this way, there is a chance for the lower-entropy texts $x^{(1)}$ and $x^{(2)}$ to be mapped to the redundant $\tilde{\zeta}$ during watermark generation.

– **Mass allocation for** $P^*_{X_1}(x) < \alpha$**:** For $x^{(3)}$, where $P^*_{X_1}(x^{(3)}) < \alpha$ and $x^{(3)} = g(\zeta^{(4)})$, we move the entire mass $P^*_{X_1}(x^{(3)})$ to $P^*_{\zeta_1}(\zeta^{(4)})$ along the blue solid line. It means that higher-entropy texts will not be mapped to the redundant $\tilde{\zeta}$ during watermark generation.

– **Outcome:** This construction ensures that $P^*_{\zeta_1}(\zeta) \leq \alpha$ for all $\zeta \in \{\zeta^{(1)}, \zeta^{(2)}, \zeta^{(3)}, \zeta^{(4)}\}$, keeping the worst-case Type-I error under control. The Type-II error is equal to $P^*_{\zeta_1}(\tilde{\zeta})$, which is exactly the universally minimum Type-II error. This scheme can be similarly generalized to $T > 1$.

In Figure 3, when there is no link between $(x, \zeta) \in \mathcal{V} \times \mathcal{Z}$, the joint probability $P^*_{X_1, \zeta_1}(x, \zeta) = 0$. By letting $\epsilon = 0$, the scheme guarantees that the watermarked LLM remains unbiased (distortion-free). Note that the detector proposed in Huang et al. (2023, Theorem 3.2) is also included in our framework, see Appendix D. Furthermore, if $P^*_{X_1^T}(x_1^T) > \alpha$ (i.e., low-entropy text), its corresponding auxiliary variable may be redundant, making it harder to detect as LLMs generated. However, this ensures better control of false alarm rates for low-entropy texts.

To better illustrate Corollary 3, we provide an example of suboptimal detectors where *no* watermarking scheme can achieve universally optimal performance.

**Example 2** (Suboptimal detectors). *Consider a detector* $\gamma(X_1^T, \zeta_1^T) = \mathbb{1}\{f(X_1^T) = \zeta_1^T\}$, *for some surjective function* $f : \mathcal{V}^T \to \mathcal{S} \subseteq \mathcal{Z}^T$. *The minimum Type-II error attained from* (Opt-I) *is* $\min_{P_{X_1^T} : D(P_{X_1^T}, Q_{X_1^T}) \leq \epsilon} \sum_{s \in \mathcal{S}} \left( \left( \sum_{x_1^T : f(x_1^T) = s} P_{X_1^T}(x_1^T) \right) - \alpha \right)_+$, *higher than* $\beta_1^*(Q_{X_1^T}, \alpha, \epsilon)$.

In the robustness discussion at the end of the paper, we will further show that this is, in fact, optimal in the presence of certain types of text modifications.

## 4 IMPLEMENTABLE TOKEN-LEVEL OPTIMAL WATERMARKING SCHEME

In our previous analysis, even with the detector having full access to the watermark sequence $\zeta_1^T$, several practical challenges remain. First, designing a proper function $g$ and alphabet $\mathcal{Z}^T$ can be difficult since $|\mathcal{V}|^T$ grows exponentially with $T$, making it hard to identify all pairs $(x_1^T, \zeta_1^T)$ such that $\mathbb{1}\{X_1^T = g(\zeta_1^T)\} = 1$. Second, these detectors are not robust to text modifications; even one changed token misclassifies the entire sequence. Third, the previous watermarking scheme is only optimal for a fixed $T$, making it unsuitable for practical scenarios where text is generated incrementally in segments with varying $T$.

To tackle these issues, we aim to design a practical detector and corresponding optimal watermarking scheme, balancing detection performance with real-world feasibility. Let's revisit examples of heuristic detectors based on specific watermarking schemes.

**Example 3** (Examples of heuristic detectors). *Two example detectors from existing works:*
- *Green-Red List watermark detector (Kirchenbauer et al., 2023a):* $\gamma(X_1^T, \zeta_1^T) = \mathbb{1}\{\frac{2}{\sqrt{T}}(\sum_{t=1}^T \mathbb{1}\{\zeta_t(X_t) = 1\} - \frac{T}{2}) \geq \lambda\}$ *where* $\lambda > 0$, $\zeta_t = (\zeta_t(x))_{x \in \mathcal{V}}$ *is uniformly sampled from* $\{\zeta \in \{0,1\}^{|\mathcal{V}|} : \|\zeta\|_1 = \rho|\mathcal{V}|\}$ *with the seed* $h(X_{t-1}, \text{key})$, $\rho \in (0, 1)$ *is the green list proportion.*
- *Gumbel-max watermark detector (Aaronson, 2023):* $\gamma(X_1^T, \zeta_1^T) = \mathbb{1}\{-\sum_{t=1}^T \log(1 - \zeta_t(X_t)) \geq \lambda\}\}$ *where* $\lambda > 0$, $\zeta_t = (\zeta_t(x))_{x \in \mathcal{V}}$ *is uniformly sampled from* $\in [0, 1]^{|\mathcal{V}|}$ *with the seed* $h(X_{t-1}^{t-n}, \text{key})$ *and* $h$ *is a hash function.*

We observe that the commonly used detectors take the non-optimal form: $\gamma(X_1^T, \zeta_1^T) = \mathbb{1}\{\frac{1}{T}\sum_{t=1}^T \text{Test Statistics of } (X_t, \zeta_t) \geq \lambda\}$. These detectors, along with corresponding watermark-

ing schemes, can effectively detect incrementally generated watermarked sequences. As $T$ increases, the earlier watermarks $\zeta_t$ and test statistics remain unchanged. Moreover, at each position $t$, the watermark alphabet only depends on the constant size $|\mathcal{V}|$. Inspired by these detectors, we propose the following detector to address the issues mentioned: for some surjective function $g : \mathcal{Z} \to \mathcal{S} \supset \mathcal{V}$,

$$\gamma(X_1^T, \zeta_1^T) = \mathbb{1}\left\{ \frac{1}{T} \sum_{t=1}^{T} \mathbb{1}\{X_t = g(\zeta_t)\} \geq \lambda \right\}. \tag{3}$$

This detector combines the advantages of existing approaches with the optimal design from Theorem 2. The test statistic for each token $(X_t, \zeta_t)$ is optimal at position $t$, enabling a token-level optimal watermarking scheme that improves the detection performance for each token.

The token-level optimal watermarking scheme is constructed following the same rule in Theorem 2, but based on the NTP distribution at each position $t$, acting only on the token vocabulary $\mathcal{V}$. The sequence-level false alarm constraint $\alpha$ is replaced by a token-level false alarm constraint $\eta \in (0, \min\{1, (\alpha/(\binom{T}{\lceil T\lambda \rceil}))^{\frac{1}{\lceil T\lambda \rceil}}\}]$, which is typically much greater than $\alpha$. For example, the joint distribution $P_{X_1^T}^*$ (cf. Theorem 2) will be replaced by $P_{X_t|x_1^{t-1}}^* = \min_{P_{X_t|x_1^{t-1}}:\mathrm{D}(P_{X_t|x_1^{t-1}}, Q_{X_t|x_1^{t-1}})\leq\epsilon} \sum_{x\in\mathcal{V}}(P_{X_t|x_1^{t-1}}(x) - \eta)_+$. Under this scheme, the previously generated watermarks remain unaffected by subsequent tokens. The details are deferred to Appendix F and the algorithm is provided in Section 5.

Subsequently, we evaluate its Type-I and Type-II errors on the entire sequence (cf. (1)).

**Lemma 4** ((Informal Statement) Token-level optimal watermarking detection errors)**.** *Under the detector $\gamma$ in* (3) *and the token-level optimal watermarking scheme, for any token-level false alarm $\eta \in (0, \min\{1, (\alpha/(\binom{T}{\lceil T\lambda \rceil}))^{\frac{1}{\lceil T\lambda \rceil}}\}]$, the worst-case Type-I error for length-$T$ sequence is upper bounded by $\alpha$. If we assume that two tokens with a positional distance greater than $n$ are independent, then with a properly chosen detector threshold, Type-II error decays exponentially in $\frac{T}{n}$.*

We show that the token-level optimal watermarking maintains good performance on the entire sequence. The formal statement is provided in Appendix G.

Furthermore, we observe that even without explicitly introducing robustness to the token-level optimal watermarking scheme, it inherently demonstrates some robustness against token replacement. The following results shows that if the watermark sequence $\zeta_1^T$ is shared between the LLM and the detector $\gamma$ (cf. (3)), the token at position $t$ can be replaced with probability $\Pr(\zeta_t$ is redundant$)$ without affecting $\frac{1}{T} \sum_{t=1}^{T} \mathbb{1}\{X_t = g(\zeta_t)\}$.

**Proposition 5** (Robustness against token replacement)**.** *Under the detector $\gamma$ in* (3) *and the token-level optimal watermarking scheme, the expected number of tokens that can be randomly replaced in $X_1^T$ without compromising detection performance is:*

$$\mathbb{E}_{\prod_{t=1}^{T} P_{X_t,\zeta_t|X_1^{t-1},\zeta_1^{t-1}}^*}\left[ \sum_{t=1}^{T} \mathbb{1}\{\zeta_t \text{ is redundant}\} \right] = \sum_{t=1}^{T} \mathbb{E}_{X_1^{t-1}}\left[ \sum_{x} \left( P_{X_t|X_1^{t-1}}^*(x|X_1^{t-1}) - \eta \right)_+ \right],$$

*where $P_{X_t|X_1^{t-1}}^*$ is induced by $P_{X_t,\zeta_t|X_1^{t-1},\zeta_1^{t-1}}^*$. When $\epsilon = 0$, we have $P_{X_t|X_1^{t-1}}^* = Q_{X_t|X_1^{t-1}}$.*

**Implementation.**  A challenge in implementing the optimal watermarking scheme is transmitting the auxiliary sequence $\zeta_1^T$ with a shared key to the detector, as $P_{\zeta_1^T}^*$ depends on $P_{X_1^T}^*$, unknown to the detector. One practical workaround is enforcing $P_{\zeta_1^T} = \mathrm{Unif}(\mathcal{Z}^T)$ and sampling $\zeta_1^T$ via a hash function with a shared key. However, this alternative watermarking scheme (cf. Appendix E) results in a higher minimum Type-II error compared to $\beta_1^*(Q_{X_1^T}, \alpha, \epsilon)$. The gap reflects the cost of pseudo-transmitting $\zeta_1^T$ via a hash function.

Another practical workaround is to use a surrogate model (much smaller than the watermarked LLMs) during detection to approximate $P_{X_1^T}^*$ based solely on the text $X_1^T$ (without prompt). Although this approximated text distribution may deviate from $P_{X_1^T}^*$, the optimal watermarking scheme and detector still exhibit superior performance. Algorithmic details are provided in Section 5. In experiments, we demonstrate that even when $\zeta_1^T$ is not fully recovered during detection, the robustness against token replacement surpasses that of benchmark watermarking schemes.

## 5 Algorithms and Experiments

**Algorithms.** To implement our proposed token-level optimal watermarking scheme and detector in Section 4, we consider the distortion-free setting ($\epsilon = 0$) and design an optimal detector (cf. (3)) that is computationally efficient by defining $g$ as the inverse of a hash function $h_{\texttt{key}}$, i.e.,

$$\gamma(X_1^T, \zeta_1^T) = \mathbb{1}\Big\{\frac{1}{T}\sum_{t=1}^{T}\mathbb{1}\{h_{\texttt{key}}(X_t) = \zeta_t\} \geq \lambda\Big\}. \tag{4}$$

Our proposed watermarking scheme relies on NTP distributions during the generation process, but these distributions are not accessible during the detection process due to the model-agnostic property. This poses a significant hurdle to practical implementation. To overcome this, we introduce a novel method using a surrogate language model (SLM) to generate surrogate NTP distributions during the detection process. The SLMs are smaller in parameter size than the watermarked LLMs but use the same tokenizer, allowing us to track NTP distributions on the same vocabulary. Along with the Gumbel-max trick to share pseudo-randomness, we present the following algorithms for sampling $\zeta_t$ during the generation and recovering it with SLM in the detection phases (cf. Figure 4).

*Watermarked text generation (Algorithm 1).* Given the detector in (4), we first define the alphabet of $\zeta_t$, which includes the unique mappings $\{h_{\texttt{key}}(x)\}_{x\in\mathcal{V}}$ derived from the vocabulary via the secret key, along with an additional redundancy $\tilde{\zeta}$. For each time step, we first construct the $P_{\zeta_t}$ from the NTP distribution $Q_{X_t|x_1^{t-1},u}$ as described in Lines 3 and 4 of Algorithm 1. Then, we employ the Gumbel-max trick (Gumbel, 1954) to sample $\zeta_t$ from $P_{\zeta_t}$. Lastly, the next token $x_t$ is sampled as $h_{\texttt{key}}^{-1}(\zeta_t)$ if $\zeta_t$ is not redundant; otherwise, it will be sampled from a multinomial distribution, as shown in Line 10.

---

**Algorithm 1** Watermarked Text Generation

**Input:** LLM $Q$, Vocabulary $\mathcal{V}$, Prompt $u$, Secret key, Token-level false alarm $\eta$.

1: $\mathcal{Z} = \{h_{\texttt{key}}(x)\}_{x\in\mathcal{V}} \cup \{\tilde{\zeta}\}$
2: **for** $t = 1,...,T$ **do**
3:     $P_{\zeta_t|x_1^{t-1},u}(\zeta) \leftarrow (Q_{X_t|x_1^{t-1},u}(h_{\texttt{key}}^{-1}(\zeta)) \wedge \eta), \forall\zeta \in \mathcal{Z}\backslash\{\tilde{\zeta}\}$.
4:     $P_{\zeta_t|x_1^{t-1},u}(\tilde{\zeta}) \leftarrow \sum_{x\in\mathcal{V}}(Q_{X_t|x_1^{t-1},u}(x) - \eta)_+$.
5:     Compute a hash of tokens $x_{t-n}^{t-1}$ with key, and use it as a seed to generate $(G_{t,\zeta})_{\zeta\in\mathcal{Z}}$ from Gumbel distribution.
6:     $\zeta_t \leftarrow \arg\max_{\zeta\in\mathcal{Z}} \log(P_{\zeta_t|x_1^{t-1},u}(\zeta)) + G_{t,\zeta}$.
7:     **if** $\zeta_t \neq \tilde{\zeta}$ **then**
8:         $x_t \leftarrow h_{\texttt{key}}^{-1}(\zeta_t)$
9:     **else**
10:         Sample $x_t \sim \left(\frac{(Q_{X_t|x_1^{t-1},u}(x)-\eta)_+}{\sum_{x\in\mathcal{V}}(Q_{X_t|x_1^{t-1},u}(x)-\eta)_+}\right)_{x\in\mathcal{V}}$
11:     **end if**
12: **end for**

**Output:** Watermarked text $x_1^T = (x_1,...,x_T)$.

---

**Algorithm 2** Watermarked Text Detection

**Input:** SLM $\tilde{Q}$, Vocabulary $\mathcal{V}$, Text $x_1^T$, Secret key, Token-level false alarm $\eta$, Threshold $\lambda$.

1: score $= 0$,   $\mathcal{Z} = \{h_{\texttt{key}}(x)\}_{x\in\mathcal{V}} \cup \{\tilde{\zeta}\}$
2: **for** $t = 1,...,T$ **do**
3:     $\tilde{P}_{\zeta_t|x_1^{t-1}}(\zeta) \leftarrow (\tilde{Q}_{X_t|x_1^{t-1}}(h_{\texttt{key}}^{-1}(\zeta)) \wedge \eta), \forall\zeta \in \mathcal{Z}\backslash\{\tilde{\zeta}\}$.
4:     $\tilde{P}_{\zeta_t|x_1^{t-1}}(\tilde{\zeta}) \leftarrow \sum_{x\in\mathcal{V}}(\tilde{Q}_{X_t|x_1^{t-1}}(x) - \eta)_+$.
5:     Compute a hash of tokens $x_{t-n}^{t-1}$ with key, and use it as a seed to generate $(G_{t,\zeta})_{\zeta\in\mathcal{Z}}$ from Gumbel distribution.
6:     $\zeta_t \leftarrow \arg\max_{\zeta\in\mathcal{Z}} \log(\tilde{P}_{\zeta_t|x_1^{t-1}}(\zeta)) + G_{t,\zeta}$.
7:     score $\leftarrow$ score $+\mathbb{1}\{h_{\texttt{key}}(x_t) = \zeta_t\}$
8: **end for**
9: **if** score $> T\lambda$ **then**
10:     **return** 1 ▷ Input text is watermarked
11: **else**
12:     **return** 0 ▷ Input text is unwatermarked
13: **end if**

---

*Watermarked text detection (Algorithm 2).* During the detection process, due to the inaccessibility of the original NTP distribution, we obtain a surrogate NTP distribution using a SLM, denoted as $\tilde{Q}_{X_t|x_1^{t-1}}$ for each $t$. We then reconstruct $P_{\zeta_t}$ approximately from $\tilde{Q}_{X_t|x_1^{t-1}}$ and sample $\zeta_t$ with the shared secret key in the same way as the generation process. At each position $t$, the score $\mathbb{1}\{h_{\texttt{key}}(x_t) = \zeta_t\} = 1$ if $\zeta_t$ is not redundant; otherwise, $\mathbb{1}\{h_{\texttt{key}}(x_t) = \zeta_t\} = 0$. At the end, we compute a final score $\frac{1}{T}\sum_{t=1}^{T}\mathbb{1}\{h_{\texttt{key}}(x_t) = \zeta_t\}$ and compare it with the given threshold $\lambda \in (0,1)$. If this score exceeds $\lambda$, the text is detected as watermarked.

**Experiment Settings.** We now introduce the setup details of our experiments.

*Implementation Details.* Our approach is implemented on two language models: Llama2-13B (Touvron et al., 2023), and Mistral-8×7B (Jiang et al., 2023). Llama2-7B serves as the surrogate model for Llama2-13B, while Mistral-7B is used as the surrogate model for Mistral-8×7B. We conduct our experiments on Nvidia A100 GPUs. In Algorithm 1, we set $\eta = 0.2$ and $T = 200$.

*Baselines.* We compare our methods with three existing watermarking methods: KGW-1 (Kirchenbauer et al., 2023a), EXP-edit (Kuditipudi et al., 2023), and Gumbel-Max (Aaronson, 2023), where

the EXP-edit and Gumbel-Max are distortion-free watermark. KGW-1 employs the prior 1 token as a hash to create a green/red list, with the watermark strength set at 2.

*Dataset and Prompt.* Our experiments are conducted using two distinct datasets. The first is an open-ended **high-entropy** generation dataset, a realnewslike subset from C4 (Raffel et al., 2020a). The second is a relatively **low-entropy** generation dataset, ELI5 (Fan et al., 2019). The realnewslike subset of C4 is tailored specifically to include high-quality journalistic content that mimics the style and format of real-world news articles. We utilize the first two sentences of each text as prompts and the following 200 tokens as human-generated text. The ELI5 dataset is specifically designed for the task of long-form question answering (QA), with the goal of providing detailed explanations for complex questions. We use each question as a prompt and its answer as human-generated text.

*Evaluation Metrics.* To evaluate the performance of watermark detection, we report the ROC-AUC score, where the ROC curve shows the True Positive Rate (TPR) against the False Positive Rate (FPR). A higher ROC-AUC score indicates better overall performance. Additionally, we provide the TPR at 1% FPR and TPR at 10% FPR to specifically evaluate detection accuracy while controlling the false classification of unwatermarked text as watermarked. The detection threshold $\lambda$ is determined empirically by the ROC-AUC score function based on 500 unwatermarked and 500 watermarked sentences. By varying $\lambda$, the ROC curve produces different false alarm rates $\alpha \in [0, 1]$.

Table 1: Watermark detection performance across different LLMs and datasets.

| Language Models | Methods | C4 | | | ELI5 | | |
|---|---|---|---|---|---|---|---|
| | | ROC-AUC | TPR@1% FPR | TPR@10% FPR | ROC-AUC | TPR@1% FPR | TPR@10% FPR |
| Llama-13B | KGW-1 | 0.995 | 0.991 | 1.000 | 0.989 | 0.974 | 0.986 |
| | EXP-edit | 0.986 | 0.968 | 0.996 | 0.983 | 0.960 | 0.995 |
| | Gumbel-Max | 0.996 | 0.993 | 0.994 | 0.999 | 0.991 | 0.994 |
| | **Ours** | 0.999 | 0.998 | 1.000 | 0.998 | 0.997 | 1.000 |
| Mistral-8 $\times$ 7B | KGW-1 | 0.997 | 0.995 | 1.000 | 0.993 | 0.983 | 0.994 |
| | EXP-edit | 0.993 | 0.970 | 0.997 | 0.994 | 0.972 | 0.996 |
| | Gumbel-Max | 0.994 | 0.989 | 0.999 | 0.987 | 0.970 | 0.990 |
| | **Ours** | 0.999 | 0.998 | 1.000 | 0.999 | 0.999 | 1.000 |

**Watermark Detection Performance.** The detection performance of unmodified watermarked text across various language models and tasks is presented in Table 1. Our watermarking method demonstrates superior performance, especially on the relatively low-entropy QA dataset. This success stems from the design of our watermarking scheme, which reduces the likelihood of low-entropy token being falsely detected as watermarked, thereby lowering the FPR. Given the tradeoff between TPR and FPR, when fixing the same FPR across different algorithms, our algorithm indeed yields a higher TPR, while other methods fail to maintain the same performance on high-entropy text. Moreover, this suggests that even without knowing the watermarked LLM during detection, we can still use the proposed SLM and Gumbel-max trick to successfully detect the watermark.

Table 2: Watermark detection performance under token replacement attack.

| Language Models | Methods | C4 | | | ELI5 | | |
|---|---|---|---|---|---|---|---|
| | | ROC-AUC | TPR@1% FPR | TPR@10% FPR | ROC-AUC | TPR@1% FPR | TPR@10% FPR |
| Llama-13B | KGW-1 | 0.965 | 0.833 | 0.952 | 0.973 | 0.892 | 0.973 |
| | EXP-edit | 0.973 | 0.857 | 0.978 | 0.967 | 0.889 | 0.975 |
| | Gumbel-Max | 0.776 | 0.396 | 0.551 | 0.733 | 0.326 | 0.556 |
| | **Ours** | 0.989 | 0.860 | 0.976 | 0.995 | 0.969 | 0.994 |
| Mistral-8 $\times$ 7B | KGW-1 | 0.977 | 0.860 | 0.962 | 0.969 | 0.890 | 0.970 |
| | EXP-edit | 0.980 | 0.861 | 0.975 | 0.983 | 0.932 | 0.988 |
| | Gumbel-Max | 0.780 | 0.402 | 0.583 | 0.753 | 0.385 | 0.556 |
| | **Ours** | 0.990 | 0.881 | 0.966 | 0.993 | 0.991 | 0.995 |

**Robustness.** We assess the robustness of our watermarking methods against a token replacement attack. As discussed in Proposition 5, the proposed token-level optimal watermarking scheme has inherent robustness against token replacement. For each watermarked text, we randomly mask 50% of the tokens and use T5-large (Raffel et al., 2020b) to predict the replacement for each masked token based on the context. For each prediction, the predicted token retains a chance of being the original one, as we do not force the replacement to differ from the original to maintain the sentence's semantics and quality. Yet, about 35% of tokens in watermarked sentences are still replaced on average. Table 2 exhibits watermark detection performance under token replacement attacks across different language models and tasks. It presents the robustness of our proposed watermarking method against the token replacement attack. Our method remains high ROC-AUC, TPR@1%FPR,

and TPR@10%FPR under this attack compared with other baselines. As pointed out in Proposition 5, it is primarily attributed to the inherent robustness of our watermark design with redundant auxiliary variables. These redundant auxiliary variables allow a certain degree of token replacement without altering the test statistics used by the detector.

**Empirical analysis on False Alarm Control.** We conduct experiments to show the relationship between theoretical FPR (i.e., $\alpha$) and the corresponding empirical FPR. As discussed in Lemma 4, we set the token-level false alarm rate as $\eta = 0.1$ and the sequence length as $T = 50$, which controls the sequence-level false alarm rate under $\alpha = \binom{T}{\lceil T\lambda \rceil}\eta^{\lceil T\lambda \rceil}$, where $\lambda$ is the detection threshold. For a given theoretical FPR $\alpha$, we calculate the corresponding threshold $\lambda$ and the empirical FPR based on 8000 unwatermarked sentences. The results, as shown in Table 3, confirm that our theoretical guarantee effectively controls the empirical false alarm rate.

Table 3: Theoretical and empirical FPR under different thresholds.

| Theoretical FPR | 9.4e-3 | 2.2e-3 | 4.9e-4 | 9.8e-5 |
|---|---|---|---|---|
| Empirical FPR | 1.1e-4 | 9.9e-5 | 9.3e-5 | 8.8e-5 |

## 6 OPTIMAL ROBUST WATERMARKING SCHEME AND DETECTOR

Thus far, we have theoretically examined the optimal detector and watermarking scheme without considering adversarial scenarios. In practice, users may attempt to modify LLM output to remove watermarks through techniques like replacement, deletion, insertion, paraphrasing, or translation. We now show that our framework can be extended to incorporating robustness against these attacks.

We consider a broad class of attacks, where the text can be altered in arbitrary ways as long as certain latent pattern, such as its *semantics*, is preserved. Specifically, let $f : \mathcal{V}^T \to [K]$ be a function that maps a sequence of tokens $X_1^T$ to a finite latent space $[K] \subset \mathbb{N}_+$; for example, $[K]$ may index $K$ distinct semantics clusters and $f$ is a function extracting the semantics. Clearly, $f$ induces an equivalence relation, say, denoted by $\equiv_f$, on $\mathcal{V}^T$, where $x_1^T \equiv_f x'^T_1$ if and only if $f(x_1^T) = f(x'^T_1)$. Let $\mathcal{B}_f(x_1^T)$ be an equivalence class containing $x_1^T$. Under the assumption that the adversary is arbitrarily powerful except that it is unable to move any $x_1^T$ outside its equivalent class $\mathcal{B}_f(x_1^T)$ (e.g., unable to alter the semantics of $x_1^T$), the "$f$-robust" Type-I and Type-II errors are then defined as

$$\beta_0(\gamma, Q_{X_1^T}, P_{\zeta_1^T}, f) := \mathbb{E}_{Q_{X_1^T} \otimes P_{\zeta_1^T}} \left[ \sup_{\tilde{x}_1^T \in \mathcal{B}_f(X_1^T)} \mathbb{1}\{\gamma(\tilde{x}_1^T, \zeta_1^T) = 1\} \right],$$

$$\beta_1(\gamma, P_{X_1^T, \zeta_1^T}, f) := \mathbb{E}_{P_{X_1^T, \zeta_1^T}} \left[ \sup_{\tilde{x}_1^T \in \mathcal{B}_f(X_1^T)} \mathbb{1}\{\gamma(\tilde{x}_1^T, \zeta_1^T) = 0\} \right].$$

Designing universally optimal $f$-robust detector and watermarking scheme can then be formulated as jointly minimizing the $f$-robust Type-II error while constraining the worst-case $f$-robust Type-I error, namely, solving the optimization problem

$$\inf_{\gamma, P_{X_1^T, \zeta_1^T}} \beta_1(\gamma, P_{X_1^T, \zeta_1^T}, f) \quad \text{s.t.} \quad \sup_{Q_{X_1^T}} \beta_0(\gamma, Q_{X_1^T}, P_{\zeta_1^T}, f) \leq \alpha, \ \mathsf{D}_{\mathsf{TV}}(P_{X_1^T}, Q_{X_1^T}) \leq \epsilon. \quad \text{(Opt-R)}$$

We prove the following theorem.

**Theorem 6** (Universally minimum $f$-robust Type-II error)**.** *The universally minimum $f$-robust Type-II error attained from* (Opt-R) *is*

$$\beta_1^*(Q_{X_1^T}, \alpha, \epsilon, f) := \min_{P_{X_1^T} : \mathsf{D}(P_{X_1^T}, Q_{X_1^T}) \leq \epsilon} \sum_{k \in [K]} \left( \left( \sum_{x_1^T : f(x_1^T) = k} P_{X_1^T}(x_1^T) \right) - \alpha \right)_+.$$

Notably, $\beta_1^*(Q_{X_1^T}, \alpha, \epsilon, f)$ aligns with the minimum Type-II error in Example 2, which is suboptimal without an adversary but becomes optimal under the adversarial settting of (Opt-R). The gap between $\beta_1^*(Q_{X_1^T}, \alpha, \epsilon, f)$ in Theorem 6 and $\beta_1^*(Q_{X_1^T}, \alpha, \epsilon)$ in Theorem 1 reflects the cost of ensuring robustness, widening as $K$ decreases (i.e., as perturbation strength increases), see Figure 5 in appendix for an illustration of the optimal $f$-robust minimum Type-II error when $f$ is a semantic mapping. Similar to Theorem 2, we derive the optimal detector and watermarking scheme achieving $\beta_1^*(Q_{X_1^T}, \alpha, \epsilon, f)$, detailed in Appendix J. These solutions closely resemble those in Theorem 2. For implementation, if the latent space $[K]$ is significantly smaller than $\mathcal{V}^T$, applying the optimal $f$-robust detector and watermarking scheme becomes more effective than those presented in Theorem 2. Additionally, a similar algorithmic strategy to the one discussed in Sections 4 and 5 can be employed to address the practical challenges discussed earlier. These extensions and efficient implementations of the function $f$ in practice are promising directions of future research.

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

# A  EXISTING WATERMARKING SCHEMES

- The **Gumbel-max watermarking scheme** (Aaronson, 2023) applies the Gumbel-max trick (Gumbel, 1954) to sample the next token $X_t$, where the Gumbel variable is exactly the auxiliary variable $\zeta_t$, which is a $|\mathcal{V}|$-dimensional vector, indexd by $x$. For $t = 1, 2, \ldots,$

  – Compute a hash of the previous $n$ tokens $X_{t-1}^{t-n}$ using a hash function $h : \mathcal{V}^n \times \mathbb{R} \to \mathbb{R}$ and a shared secret key, i.e., $h(X_{t-1}^{t-n}, \text{key})$.
  – Use $h(X_{t-1}^{t-n}, \text{key})$ as a seed to uniformly sample the auxiliary vector $\zeta_t$ from $[0, 1]^{|\mathcal{V}|}$.
  – Sample $X_t$ using the Gumbel-max trick

$$X_t = \arg\max_{x \in \mathcal{V}} \log Q_{X_t|x_1^{t-1}}(x) - \log(-\log \zeta_t(x)).$$

- In the **inverse transform watermarking scheme** (Kuditipudi et al., 2023), the vocabulary $\mathcal{V}$ is considered as $[|\mathcal{V}|]$ and the combination of the uniform random variable and the randomly permuted index vector is the auxiliary variable $\zeta_t$.

  – Use key as a seed to uniformly and independently sample $\{U_t\}_{t=1}^T$ from $[0, 1]$, and $\{\pi_t\}_{t=1}^T$ from the space of permutations over $[|\mathcal{V}|]$. Let the auxiliary variable $\zeta_t = (U_t, \pi_t)$, for $t = 1, 2, \ldots, T$.
  – Sample $X_t$ as follows

$$X_t = \pi_t^{-1}\left( \min\left\{ i \in [|\mathcal{V}|] : \sum_{x \in [|\mathcal{V}|]} \left( Q_{X_t|x_1^{t-1}}(x)\mathbb{1}\{\pi_t(x) \leq i\} \right) \geq U_t \right\} \right),$$

  where $\pi_t^{-1}$ denotes the inverse permutation.

- In **Liu & Bu (2024)**, they propose a watermarking scheme that applies a similar technique as Green-Red List, but designs $h$ as a pretrained neural network instead of a hash function. The auxiliary variable $\zeta_t$ is sampled from the set $\{\mathbf{v} \in \{0, 1\}^{|\mathcal{V}|} : \|\mathbf{v}\|_1 = \rho|\mathcal{V}|\}$ using the seed $h(\phi(X_1^{t-1}), \text{key})$, where $h$ takes the semantics $\phi(X_1^{t-1})$ of the generated text and the secret key as inputs. They sample $X_t$ following the same process as that of Green-Red List.

# B  PROOF OF THEOREM 1

According to the Type-I error constraint, we have $\forall x_1^T \in \mathcal{V}^T$,

$$\alpha \geq \max_{Q_{X_1^T}} \mathbb{E}_{Q_{X_1^T} P_{\zeta_1^T}}[\mathbb{1}\{(X_1^T, \zeta_1^T) \in \mathcal{A}_1\}]$$

$$\geq \mathbb{E}_{\delta_{x_1^T} P_{\zeta_1^T}}[\mathbb{1}\{(X_1^T, \zeta_1^T) \in \mathcal{A}_1\}]$$

$$= \mathbb{E}_{P_{\zeta_1^T}}[\mathbb{1}\{(x_1^T, \zeta_1^T) \in \mathcal{A}_1\}]$$

$$= \begin{cases} \sum_{\zeta_1^T} P_{\zeta_1^T}(\zeta_1^T)\mathbb{1}\{(x_1^T, \zeta_1^T) \in \mathcal{A}_1\}, & \mathcal{Z} \text{ is discrete;} \\ \int P_{\zeta_1^T}(\zeta_1^T)\mathbb{1}\{(x_1^T, \zeta_1^T) \in \mathcal{A}_1\} \, d\zeta_1^T, & \mathcal{Z} \text{ is continuouts;} \end{cases}.$$

In the following, for notational simplicity, we assume that $\mathcal{Z}$ is discrete. However, the derivations hold for both discrete $\mathcal{Z}$ and continuous $\mathcal{Z}$. The Type-II error is given by $1 - \mathbb{E}_{P_{X_1^T, \zeta_1^T}}[\mathbb{1}\{(X_1^T, \zeta_1^T) \in \mathcal{A}_1\}]$. We have

$$\mathbb{E}_{P_{X_1^T, \zeta_1^T}}[\mathbb{1}\{(X_1^T, \zeta_1^T) \in \mathcal{A}_1\}] = \sum_{x_1^T} \underbrace{\sum_{\zeta_1^T} P_{X_1^T, \zeta_1^T}(x_1^T, \zeta_1^T)\mathbb{1}\{(x_1^T, \zeta_1^T) \in \mathcal{A}_1\}}_{C(x_1^T)}, \tag{5}$$

where for all $x_1^T \in \mathcal{V}^T$,

$$C(x_1^T) \leq P_{X_1^T}(x_1^T) \quad \text{and} \quad C(x_1^T) \leq \sum_{\zeta_1^T} P_{\zeta_1^T}(\zeta_1^T)\mathbb{1}\{(x_1^T, \zeta_1^T) \in \mathcal{A}_1\} \leq \alpha$$

according to the Type-I error bound. Therefore,

$$\mathbb{E}_{P_{X_1^T, \zeta_1^T}}[\mathbb{1}\{(X_1^T, \zeta_1^T) \in \mathcal{A}_1\}] = \sum_{x_1^T} C(x_1^T) \le \sum_{x_1^T}(P_{X_1^T}(x_1^T) \wedge \alpha)$$

$$= 1 - \sum_{x_1^T}(P_{X_1^T}(x_1^T) - \alpha)_+ \qquad (6)$$

where (6) is maximized by taking

$$P_{X_1^T} = P^*_{X_1^T} := \underset{P_{X_1^T}:\mathsf{D}(P_{X_1^T}, Q_{X_1^T}) \le \epsilon}{\arg\min} \sum_{x_1^T}(P_{X_1^T}(x_1^T) - \alpha)_+. \qquad (7)$$

For any $P_{X_1^T}$, the Type-II error is lower bounded by

$$\mathbb{E}_{P_{X_1^T, \zeta_1^T}}[\mathbb{1}\{(X_1^T, \zeta_1^T) \notin \mathcal{A}_1\}] \ge \sum_{x_1^T}(P_{X_1^T}(x_1^T) - \alpha)_+.$$

By plugging $P^*_{X_1^T}$ into this lower bound, we obtain a Type-II lower bound that holds for all $\gamma$ and $P_{X_1^T, \zeta_1^T}$. Recall that Huang et al. (2023) proposed a type of detector and watermarking scheme that achieved this lower bound. Thus, it is actually the universal minimum Type-II error over all possible $\gamma$ and $P_{X_1^T, \zeta_1^T}$, denoted by $\beta_1^*(Q_{X_1^T}, \epsilon, \alpha)$.

Specifically, define $\epsilon^*(x_1^T) = Q_{X_1^T}(x_1^T) - P^*_{X_1^T}(x_1^T)$ and we have

$$\sum_{x_1^T:P^*_{X_1^T}(x_1^T) \ge \alpha} \epsilon^*(x_1^T) = \sum_{x_1^T:P^*_{X_1^T}(x_1^T) \ge \alpha, \epsilon^*(x_1^T) \ge 0} \epsilon^*(x_1^T) + \underbrace{\sum_{x_1^T:P^*_{X_1^T}(x_1^T) \ge \alpha, \epsilon^*(x_1^T) \le 0} \epsilon^*(x_1^T)}_{\le 0}$$

$$\le \sum_{x_1^T:P^*_{X_1^T}(x_1^T) \ge \alpha, \epsilon^*(x_1^T) \ge 0} \epsilon^*(x_1^T)$$

$$= \sum_{x_1^T:P^*_{X_1^T}(x_1^T) \ge \alpha, Q_{X_1^T}(x_1^T) \ge P^*_{X_1^T}(x_1^T)} \epsilon^*(x_1^T)$$

$$\le \sum_{x_1^T:Q_{X_1^T}(x_1^T) \ge P^*_{X_1^T}(x_1^T)} \epsilon^*(x_1^T) \le \epsilon$$

where the last inequality follows from the total variation distance constraint $\mathsf{D}_{\mathsf{TV}}(P_{X_1^T}, Q_{X_1^T}) \le \epsilon$. We rewrite $\beta_1^*(Q_{X_1^T}, \epsilon, \alpha)$ as follows:

$$\beta_1^*(Q_{X_1^T}, \epsilon, \alpha) = \underset{P_{X_1^T}:\mathsf{D}_{\mathsf{TV}}(P_{X_1^T}, Q_{X_1^T}) \le \epsilon}{\min} \sum_{x_1^T}(P_{X_1^T}(x_1^T) - \alpha)_+ \qquad (8)$$

$$= \sum_{x_1^T:P^*_{X_1^T}(x_1^T) \ge \alpha}(P^*_{X_1^T}(x_1^T) - \alpha),$$

$$= \sum_{x_1^T:P^*_{X_1^T}(x_1^T) \ge \alpha}(Q_{X_1^T}(x_1^T) - \epsilon^*(x_1^T) - \alpha)$$

$$= \sum_{x_1^T:P^*_{X_1^T}(x_1^T) \ge \alpha}(Q_{X_1^T}(x_1^T) - \alpha) - \sum_{x_1^T:P^*_{X_1^T}(x_1^T) \ge \alpha} \epsilon^*(x_1^T)$$

$$\ge \sum_{x_1^T}(Q_{X_1^T}(x_1^T) - \alpha)_+ - \epsilon,$$

where the last inequality follows from $\sum_{x_1^T : P^*_{X_1^T}(x_1^T) \geq \alpha} \epsilon^*(x_1^T) \leq \epsilon$, i.e. the TV constraint limits how much the distribution $P^*_{X_1^T}$ can be perturbed from $Q_{X_1^T}$. Since $\beta_1^*(Q_{X_1^T}, \epsilon, \alpha) \geq 0$, finally we have

$$\beta_1^*(Q_{X_1^T}, \epsilon, \alpha) \geq \left( \sum_{x_1^T} (Q_{X_1^T}(x_1^T) - \alpha)_+ - \epsilon \right)_+.$$

Notably, the lower bound is achieved when $\{x_1^T : P^*_{X_1^T}(x_1^T) \geq \alpha\} = \{x_1^T : Q_{X_1^T}(x_1^T) \geq P^*_{X_1^T}(x_1^T)\}$ and $\mathsf{D}_{\mathsf{TV}}(Q_{X_1^T}, P^*_{X_1^T}) = \epsilon$. That is, to construct $P^*_{X_1^T}$, an $\epsilon$ amount of the mass of $Q_{X_1^T}$ above $\alpha$ is moved to below $\alpha$, which is possible only when $\sum_{x_1^T} (\alpha - Q_{X_1^T}(x_1^T))_+ \geq \epsilon$. Note that Huang et al. (2023, Theorem 3.2) points out a sufficient condition for this to hold: $|\mathcal{V}|^T \geq \frac{1}{\alpha}$. The optimal distribution $P^*_{X_1^T}$ thus satisfies

$$\sum_{x_1^T : Q_{X_1^T}(x_1^T) \geq \alpha} (Q_{X_1^T}(x_1^T) - P^*_{X_1^T}(x_1^T)) = \sum_{x_1^T : Q_{X_1^T}(x_1^T) \leq \alpha} (P^*_{X_1^T}(x_1^T) - Q_{X_1^T}(x_1^T)) = \epsilon.$$

**Refined constraints for optimization.** We notice that the feasible region of (Opt-I) can be further reduced as follows:

$$\min_{P_{X_1^T}} \min_{P_{\zeta_1^T | X_1^T}} \quad \mathbb{E}_{P_{X_1^T} P_{\zeta_1^T | X_1^T}} [1 - \gamma(X_1^T, \zeta_1^T)] \qquad \text{(Opt-II)}$$

$$\text{s.t.} \quad \int P_{\zeta_1^T | X_1^T}(\zeta_1^T | x_1^T) \, d\zeta_1^T = 1, \ \forall x_1^T$$

$$\int P_{\zeta_1^T | X_1^T}(\zeta_1^T | x_1^T) \gamma(x_1^T, \zeta_1^T) \leq 1 \wedge \frac{\alpha}{P_{X_1^T}(x_1^T)}, \ \forall x_1^T \qquad (9)$$

$$\mathsf{D}_{\mathsf{TV}}(P_{X_1^T}, Q_{X_1^T}) \leq \epsilon,$$

$$\sup_{Q_{X_1^T}} \sum_{x_1^T} Q_{X_1^T}(x_1^T) \int \left( \sum_{y_1^T} P_{\zeta_1^T | X_1^T}(\zeta_1^T | y_1^T) P_{X_1^T}(y_1^T) \right) \gamma(x_1^T, \zeta_1^T) \, d\zeta_1^T \leq \alpha,$$

where (9) is an additional constraint on $P_{\zeta_1^T | X_1^T}$. If and only if (9) can be achieved with equality, the minimum of the objective function $\mathbb{E}_{P_{X_1^T} P_{\zeta_1^T | X_1^T}}[1 - \gamma(X_1^T, \zeta_1^T)]$ reaches (2).

## C  PROOF OF EXAMPLE 2

In this proof, we assume that $\mathcal{Z}$ is discrete for simplicity. However, the result holds for continuous $\mathcal{Z}$ without loss of generality. If the detector accepts the form $\gamma(X_1^T, \zeta_1^T) = \mathbb{1}\{f(X_1^T) = \zeta_1^T\}$ for some surjective function $f : \mathcal{V}^T \to \mathcal{S}$ and $\mathcal{S} \subseteq \mathcal{Z}^T$, we have for any $s \in \mathcal{S}$,

$$\alpha \geq \sup_{Q_{X_1^T}} \mathbb{E}_{Q_{X_1^T} P_{\zeta_1^T}} [\mathbb{1}\{f(X_1^T) = \zeta_1^T\}] \geq \mathbb{E}_{P_{\zeta_1^T}} [\mathbb{1}\{s = \zeta_1^T\}]$$

$$= \sum_{\zeta_1^T} P_{\zeta_1^T}(\zeta_1^T) \mathbb{1}\{s = \zeta_1^T\},$$

and (5) can be rewritten as

$$\mathbb{E}_{P_{X_1^T, \zeta_1^T}}[\mathbb{1}\{f(X_1^T) = \zeta_1^T\}] = \sum_{s \in \mathcal{S}} \underbrace{\sum_{x_1^T : f(x_1^T) = s} \sum_{\zeta_1^T} P_{X_1^T, \zeta_1^T}(x_1^T, \zeta_1^T) \mathbb{1}\{f(x_1^T) = \zeta_1^T\}}_{C(s)},$$

where

$$C(s) \leq \sum_{x_1^T : f(x_1^T) = s} P_{X_1^T}(x_1^T) \quad \text{and} \quad C(s) \leq \sum_{\zeta_1^T} P_{\zeta_1^T}(\zeta_1^T) \mathbb{1}\{s = \zeta_1^T\} \leq \alpha.$$

Therefore, the Type-II error for such type of detector $\gamma$ is lower bounded by

$$\mathbb{E}_{P_{X_1^T, \zeta_1^T}}[\mathbb{1}\{f(X_1^T) \neq \zeta_1^T\}]$$

$$= 1 - \sum_{s \in \mathcal{S}} C(s) \geq 1 - \sum_{s \in \mathcal{S}} \left( \left( \sum_{x_1^T : f(x_1^T) = s} P_{X_1^T}(x_1^T) \right) \wedge \alpha \right)$$

$$= \sum_{s \in \mathcal{S}} \left( \left( \sum_{x_1^T : f(x_1^T) = s} P_{X_1^T}(x_1^T) \right) - \alpha \right)_+$$

$$\geq \min_{P_{X_1^T} : D_{\mathsf{TV}}(P_{X_1^T}, Q_{X_1^T}) \leq \epsilon} \sum_{s \in \mathcal{S}} \left( \left( \sum_{x_1^T : f(x_1^T) = s} P_{X_1^T}(x_1^T) \right) - \alpha \right)_+,$$

where the last inequality holds with equality when

$$P_{X_1^T} = \underset{P_{X_1^T} : D_{\mathsf{TV}}(P_{X_1^T}, Q_{X_1^T}) \leq \epsilon}{\arg\min} \sum_{s \in \mathcal{S}} \left( \left( \sum_{x_1^T : f(x_1^T) = s} P_{X_1^T}(x_1^T) \right) - \alpha \right)_+.$$

This minimum achievable Type-II error is higher than $\beta_1^*(Q_{X_1^T}, \alpha, \epsilon)$ (cf. (8)).

## D   FORMAL STATEMENT OF THEOREM 2 AND ITS PROOF

**Theorem 2 [Formal]** (Optimal type of detectors and watermarking schemes). *The set of all detectors that achieve the minimum Type-II error $\beta_1^*(Q_{X_1^T}, \alpha, \epsilon)$ in Theorem 1 for all text distribution $Q_{X_1^T} \in \mathcal{P}(\mathcal{V}^T)$ and distortion level $\epsilon \geq 0$ is precisely $\Gamma^*$. After enlarging $\mathcal{Z}^T$ to include redundant auxiliary values, the detailed construction of the optimal watermarking scheme is as follows:*

$$P_{X_1^T}^* = \min_{P_{X_1^T} : D(P_{X_1^T}, Q_{X_1^T}) \leq \epsilon} \sum_{x_1^T} (P_{X_1^T}(x_1^T) - \alpha)_+,$$

*and for any $x_1^T \in \mathcal{V}^T$, $P_{\zeta_1^T | X_1^T}^*(\zeta_1^T | x_1^T)$ satisfies* (10)

$$\begin{cases} P_{X_1^T}^*(x_1^T) \sum_{\zeta_1^T} P_{\zeta_1^T | X_1^T}^*(\zeta_1^T | x_1^T) \gamma(x_1^T, \zeta_1^T) = P_{X_1^T}^*(x_1^T) \wedge \alpha, & \forall \zeta_1^T \text{ s.t. } \gamma(x_1^T, \zeta_1^T) = 1; \\ P_{X_1^T}^*(x_1^T) \sum_{\text{redundant } \zeta_1^T} P_{\zeta_1^T | X_1^T}^*(\zeta_1^T | x_1^T) = \left( P_{X_1^T}^*(x_1^T) - \alpha \right)_+, & \forall \text{ redundant } \zeta_1^T; \\ P_{\zeta_1^T | X_1^T}^*(\zeta_1^T | x_1^T) = 0, & \text{otherwise.} \end{cases}$$

*Proof.* First, we observe that the lower bound on the Type-II error in (2) is attained if and only if the constraint in (9) holds with equality for all $x_1^T$ and for the optimizer. Thus, it suffices to show that for any detector $\gamma \notin \Gamma^*$, the constraint in (9) cannot hold with equality for all $x_1^T$ given any text distributions $Q_{X_1^T}$. First define an arbitrary surjective function $g : \mathcal{Z}^T \to \mathcal{S}$, where $\mathcal{S}$ is on the same metric space as $\mathcal{V}^T$. Cases 1 and 2 prove that $\mathcal{V}^T \subset \mathcal{S}$. Case 3 proves that $\gamma$ can only be $\gamma(X_1^T, \zeta_1^T) = \mathbb{1}\{X_1^T = g(\zeta_1^T)\}$.

- **Case 1:** $\gamma(X_1^T, \zeta_1^T) = \mathbb{1}\{X_1^T = g(\zeta_1^T)\}$ but $\mathcal{S} \subset \mathcal{V}^T$. There exists $\tilde{x}_1^T$ such that for all $\zeta_1^T$, $\mathbb{1}\{\tilde{x}_1^T = g(\zeta_1^T)\} = 0$. Under this case, (9) cannot hold with equality for $\tilde{x}_1^T$ since the LHS is always 0 while the RHS is positive.

- **Case 2:** $\gamma(X_1^T, \zeta_1^T) = \mathbb{1}\{X_1^T = g(\zeta_1^T)\}$ but $\mathcal{S} = \mathcal{V}^T$. Let us start from the simple case where $T = 1$, $\mathcal{V} = \{x_1, x_2\}$, $\mathcal{Z} = \{\zeta_1, \zeta_2\}$, and $g$ is an identity mapping. Given any $Q_X$ and any feasible $P_X$ such that $D_{\mathsf{TV}}(P_X, Q_X) \leq \epsilon$, when (9) holds with equality, i.e.,

$$P_{X,\zeta}(x_1, \zeta_1) = P_X(x_1) \wedge \alpha \quad \text{and} \quad P_{X,\zeta}(x_2, \zeta_2) = P_X(x_2) \wedge \alpha,$$

then the marginal $P_\zeta$ is given by: $P_\zeta(\zeta_1) = P_X(x_1) \wedge \alpha + (P_X(x_2) - \alpha)_+$, $P_\zeta(\zeta_2) = P_X(x_2) \wedge \alpha + (P_X(x_1) - \alpha)_+$. The worst-case Type-I error is given by

$$\sup_{Q_X} \left( Q_X(x_1)\big(P_X(x_1) \wedge \alpha + (P_X(x_2) - \alpha)_+\big) + Q_X(x_2)\big(P_X(x_2) \wedge \alpha + (P_X(x_1) - \alpha)_+\big) \right)$$

$$\geq P_X(x_1) \wedge \alpha + (P_X(x_2) - \alpha)_+$$
$$> \alpha, \quad \text{if } P_X(x_1) > \alpha, P_X(x_2) > \alpha.$$

It implies that for any $Q_X$ such that $\{P_X \in \mathcal{P}(\mathcal{V}) : \mathsf{D}_{\mathsf{TV}}(P_X, Q_X) \leq \epsilon\} \subseteq \{P_X \in \mathcal{P}(\mathcal{V}) : P_X(x_1) > \alpha, P_X(x_2) > \alpha\}$, the false-alarm constraint is violated when (9) holds with equality. It can be easily verified that this result also holds for larger $(T, \mathcal{V}, \mathcal{Z})$ and other functions $g : \mathcal{Z}^T \rightarrow \mathcal{V}^T$.

- **Case 3:** Let $\Xi_\gamma(x_1^T) := \{\zeta_1^T \in \mathcal{Z}^T : \gamma(x_1^T, \zeta_1^T) = 1\}$. $\exists x_1^T \neq y_1^T \in \mathcal{V}^T$, s.t. $\Xi(x_1^T) \cap \Xi(y_1^T) \neq \emptyset$.
  For any detector $\gamma \notin \Gamma^*$ that does not fall into Cases 1 and 2, it falls into Case 3. Let us start from the simple case where $T = 1$, $\mathcal{V} = \{x_1, x_2\}$, $\mathcal{Z} = \{\zeta_1, \zeta_2, \zeta_3\}$. Consider a detector $\gamma$ as follows: $\gamma(x_1, \zeta_1) = \gamma(x_2, \zeta_1) = 1$ and $\gamma(x, \zeta) = 0$ for all other pairs $(x, \zeta) \in \mathcal{V} \times \mathcal{Z}$. Hence, $\Xi(x_1) \cap \Xi(x_2) = \{\zeta_1\}$. When (9) holds with equality, i.e.,

$$P_{X,\zeta}(x_1, \zeta_1) = P_X(x_1) \wedge \alpha \quad \text{and} \quad P_{X,\zeta}(x_2, \zeta_1) = P_X(x_2) \wedge \alpha,$$

we have the worst-case Type-I error lower bounded by

$$\sup_{Q_X} \left( Q_X(x_1) P_\zeta(\zeta_1) + Q_X(x_2) P_\zeta(\zeta_1) \right) = P_\zeta(\zeta_1) = P_X(x_1) \wedge \alpha + P_X(x_2) \wedge \alpha$$

$$> \alpha, \quad \text{if } P_X(x_1) > \alpha \text{ or } P_X(x_2) > \alpha.$$

Thus, for any $Q_X$ such that $\{P_X \in \mathcal{P}(\mathcal{V}) : \mathsf{D}_{\mathsf{TV}}(P_X, Q_X) \leq \epsilon\} \subseteq \{P_X \in \mathcal{P}(\mathcal{V}) : P_X(x_1) > \alpha \text{ or } P_X(x_2) > \alpha\}$, the false-alarm constraint is violated when (9) holds with equality.

If we consider a detector $\gamma$ as follows: $\gamma(x_1, \zeta_1) = \gamma(x_2, \zeta_1) = \gamma(x_2, \zeta_2) = 1$ and $\gamma(x, \zeta) = 0$ for all other pairs $(x, \zeta) \in \mathcal{V} \times \mathcal{Z}$. We still have $\Xi(x_1) \cap \Xi(x_2) = \{\zeta_1\}$. When (9) holds with equality, i.e.,

$$P_{X,\zeta}(x_1, \zeta_1) = P_X(x_1) \wedge \alpha \quad \text{and} \quad P_{X,\zeta}(x_2, \zeta_1) + P_{X,\zeta}(x_2, \zeta_2) = P_X(x_2) \wedge \alpha,$$

we have the worst-case Type-I error lower bounded by

$$\sup_{Q_X} \left( Q_X(x_1) P_\zeta(\zeta_1) + Q_X(x_2)(P_\zeta(\zeta_1) + P_\zeta(\zeta_2)) \right) = \sup_{Q_X} \left( P_\zeta(\zeta_1) + Q_X(x_2) P_\zeta(\zeta_2) \right)$$

$$= P_\zeta(\zeta_1) + P_\zeta(\zeta_2) = P_X(x_1) \wedge \alpha + P_X(x_2) \wedge \alpha > \alpha, \quad \text{if } P_X(x_1) > \alpha \text{ or } P_X(x_2) > \alpha,$$

which is the same as the previous result.

If we let $\mathcal{V} = \{x_1, x_2, x_3\}$, $\mathcal{Z} = \{\zeta_1, \zeta_2, \zeta_3, \zeta_4\}$ and $\gamma(x_3, \zeta_3) = 1$ in addition to the aforementioned $\gamma$, we can similarly show that the worst-case Type-I error is larger than $\alpha$ for some distributions $Q_X$.

Therefore, it can be observed that as long as $\Xi(x_1^T) \cap \Xi(y_1^T) \neq \emptyset$ for some $x_1^T \neq y_1^T \in \mathcal{V}^T$, (9) can not be achieved with equality for all $Q_{X_1^T}$ and $\epsilon$ even for larger $(T, \mathcal{V}, \mathcal{Z})$ as well as continuous $\mathcal{Z}$.

In conclusion, for any detector $\gamma \notin \Gamma^*$, the universal minimum Type-II error in (2) cannot be obtained for all $Q_{X_1^T}$ and $\epsilon$.

Since the optimal detector takes the form $\gamma(X_1^T, \zeta_1^T) = \mathbb{1}\{X_1^T = g(\zeta_1^T)\}$ for some function $g : \mathcal{Z}^T \rightarrow \mathcal{S}, \mathcal{S} \supset \mathcal{V}^T$, and the token vocabulary is discrete, it suffices to consider discrete $\mathcal{Z}$ to derive the optimal watermarking scheme.

Under the watermarking scheme $P^*_{X_1^T, \zeta_1^T}$ (cf. (7) and (10)), the Type-I and Type-II errors are given by:

**Type-I error:**

$$\because \forall y_1^T \in \mathcal{V}^T, \quad \mathbb{E}_{P^*_{\zeta_1^T}}[\mathbb{1}\{y_1^T = g(\zeta_1^T)\}] = \sum_{\zeta_1^T} P^*_{\zeta_1^T}(\zeta_1^T) \mathbb{1}\{y_1^T = g(\zeta_1^T)\}$$

$$= \sum_{\zeta_1^T} \sum_{x_1^T} P^*_{X_1^T, \zeta_1^T}(x_1^T, \zeta_1^T) \mathbb{1}\{y_1^T = g(\zeta_1^T)\}$$

$$= P^*_{X_1^T}(y_1^T) \sum_{\zeta_1^T} P^*_{\zeta_1^T | X_1^T}(\zeta_1^T | y_1^T) \mathbb{1}\{y_1^T = g(\zeta_1^T)\} = P^*_{X_1^T}(y_1^T) \wedge \alpha$$

$$\leq \alpha,$$

and since any distribution $Q_{X_1^T}$ can be written as a linear combinations of $\delta_{y_1^T}$,

$$\therefore \max_{Q_{X_1^T}} \mathbb{E}_{Q_{X_1^T} P_{\zeta_1^T}^*}[\mathbb{1}\{X_1^T = g(\zeta_1^T)\}] \leq \alpha.$$

**Type-II error:**

$$1 - \mathbb{E}_{P_{X_1^T, \zeta_1^T}^*}[\mathbb{1}\{X_1^T = g(\zeta_1^T)\}]$$

$$= 1 - \sum_{x_1^T} \sum_{\zeta_1^T} P_{X_1^T, \zeta_1^T}^*(x_1^T, \zeta_1^T)\mathbb{1}\{x_1^T = g(\zeta_1^T)\}$$

$$= 1 - \sum_{x_1^T} P_{X_1^T}^*(x_1^T) \sum_{\zeta_1^T} P_{\zeta_1^T|X_1^T}^*(\zeta_1^T|x_1^T)\mathbb{1}\{x_1^T = g(\zeta_1^T)\}$$

$$= 1 - \sum_{x_1^T} \left(P_{X_1^T}^*(x_1^T) \wedge \alpha\right)$$

$$= \sum_{x_1^T : P_{X_1^T}^*(x_1^T) > \alpha} \left(P_{X_1^T}^*(x_1^T) - \alpha\right).$$

The optimality of $P_{X_1^T, \zeta_1^T}^*$ is thus proved. We note that (9) in (Opt-II) holds with equality under this optimal conditional distribution $P_{\zeta_1^T|X_1^T}^*$.

Compared to Huang et al. (2023, Theorem 3.2), their proposed detector is equivalent to $\gamma(X_1^T, \zeta_1^T) = \mathbb{1}\{X_1^T = \zeta_1^T\}$, where $\mathcal{Z}^T = \mathcal{V}^T \cup \{\tilde{\zeta}_1^T\}$ and $\tilde{\zeta}_1^T \notin \mathcal{V}^T$, meaning that it belongs to $\Gamma^*$. $\qquad\square$

# E    OPTIMAL WATERMARKING SCHEME WITH UNIFORM $P_{\zeta_1^T}$ FOR $\gamma \in \Gamma^*$

After enforcing the marginal distribution $P_{\zeta^T} = \text{Unif}(\mathcal{Z}^T)$ and using a shared key to sample $\zeta_1^T$ via a hash function, the alternative watermarking scheme optimal for $\gamma \in \Gamma^*$ when $g$ is an identity mapping is given in the following lemma. The scheme can be generalized to other functions $g$.

**Lemma 7** (Optimal watermarking scheme for $\gamma = \mathbb{1}\{X_1^T = \zeta_1^T\}$ when $P_{\zeta_t} = \text{Unif}(\mathcal{Z})$). *When* $\gamma = \mathbb{1}\{X_1^T = \zeta_1^T\}$, $P_{\zeta_t} = \text{Unif}(\mathcal{Z})$, *and* $\alpha \geq \frac{1}{|\mathcal{Z}|^T}$, *the minimum Type-II error is* $\min_{P_{X_1^T} : \mathsf{D}_{\mathsf{TV}}(P_{X_1^T} \| Q_{X_1^T}) \leq \epsilon} \sum_{x_1^T} \left(P_{X_1^T}(x_1^T) - \frac{1}{|\mathcal{Z}|^T}\right)_+$. *The optimal $\epsilon$-distorted watermarking scheme that achieves the minimum Type-II error is*

$$P_{X_1^T, \zeta_1^T}^*(x_1^T, \zeta_1^T) = \begin{cases} \min\{P_{X_1^T}^*(x_1^T), \frac{1}{|\mathcal{Z}|^T}\}, & \text{if } x_1^T = \zeta_1^T; \\ \dfrac{\left(P_{X_1^T}^*(x_1^T) - \frac{1}{|\mathcal{Z}|^T}\right)_+ \cdot \left(\frac{1}{|\mathcal{Z}|^T} - P_{X_1^T}^*(\zeta_1^T)\right)_+}{\mathsf{D}_{\mathsf{TV}}(P_{X_1^T}^*, \text{Unif}(\mathcal{Z}^T))}, & \text{otherwise,} \end{cases}$$

*where* $P_{X_1^T}^* = \arg\min_{P_{X_1^T} : \mathsf{D}_{\mathsf{TV}}(P_{X_1^T} \| Q_{X_1^T}) \leq \epsilon} \sum_{x_1^T} \left(P_{X_1^T}(x_1^T) - \frac{1}{|\mathcal{Z}|^T}\right)_+$.

The proof of Lemma 7 follows from the fact that $\mathsf{D}_{\mathsf{TV}}(\mu, \nu) = \inf_{\pi \in \Pi(\mu, \nu)} \pi(X \neq Y)$, where $X \sim \mu$, $Y \sim \nu$ and $\Pi(\mu, \nu)$ is the set of all couplings of Borel probability measures $\mu$ and $\nu$. Note that when $P_{\zeta_t} = \text{Unif}(\mathcal{Z})$, if $\alpha < \frac{1}{|\mathcal{Z}|^T}$, the feasible region of (Opt-I) becomes empty. With this watermarking scheme, the detector can fully recover $\zeta_1^T$ using a pseudorandom generator and shared key. However, the resulting minimum Type-II error is larger than $\beta_1^*(Q_{X_1^T}, \alpha, \epsilon)$ from Theorem 1, as $\alpha \geq \frac{1}{|\mathcal{Z}|^T}$. In practice, the gap is significant since $\frac{1}{|\mathcal{Z}|^T} = \mathcal{O}(10^{-4T})$ is much smaller than typical values of $\alpha$. This gap reflects the cost of pseudo-transmitting $\zeta_1^T$ using only the shared key. Nonetheless, if $T = 1$, it is possible to set false alarm constraint to $\alpha = \frac{1}{|\mathcal{Z}|}$ and mitigate the performance loss. Motivated by this, we move on to discuss the token-level optimal watermarking scheme.

*Proof.* Consider $\gamma(X_1^T, \zeta_1^T) = \mathbb{1}\{X_1^T = \zeta_1^T\}$ and $\mathcal{V}^T \subseteq \mathcal{Z}^T$, which is a model-agnostic detector. Let us first assume $\epsilon = 0$. The objective function (i.e. Type-II error) becomes $P_{X_1^T, \zeta_1^T}(X_1^T \neq \zeta_1^T)$, whose minimum is well-known as $\mathsf{D}_{\mathsf{TV}}(Q_{X_1^T}, P_{\zeta_1^T})$ and the minimizer is

$$P_{X_1^T, \zeta_1^T}^*(x_1^T, \zeta_1^T) = \begin{cases} \min\{Q_{X_1^T}(x_1^T), P_{\zeta_1^T}(\zeta_1^T)\}, & \text{if } x_1^T = \zeta_1^T; \\ \frac{(Q_{X_1^T}(x_1^T) - P_{\zeta_1^T}(x_1^T))_+ + (P_{\zeta_1^T}(\zeta_1^T) - Q_{X_1^T}(\zeta_1^T))_+}{\mathsf{D}_{\mathsf{TV}}(Q_{X_1^T}, P_{\zeta_1^T})}, & \text{otherwise.} \end{cases} \quad (11)$$

This holds for any given pair of $(Q_{X_1^T}, P_{\zeta_1^T})$. This watermarking scheme basically tries to force $X_1^T = \zeta_1^T$ as often as possible. However, we need to design $P_{\zeta_1^T}$ such that the Type-I error probability $\sup_{Q_{X_1^T}} \mathbb{E}_{Q_{X_1^T} P_{\zeta_1^T}}[\mathbb{1}\{X_1^T = \zeta_1^T\}] \leq \alpha$, i.e.,

$$P_{\zeta_1^T}^* := \underset{P_{\zeta_1^T}: \sup_{Q_{X_1^T}} \mathbb{E}_{Q_{X_1^T} \otimes P_{\zeta_1^T}}[\mathbb{1}\{X_1^T = \zeta_1^T\}] \leq \alpha}{\arg\min} \mathsf{D}_{\mathsf{TV}}(Q_{X_1^T}, P_{\zeta_1^T})$$

$$= \underset{P_{\zeta_1^T}: \sup_{Q_{X_1^T}} \langle Q_{X_1^T}, P_{\zeta_1^T} \rangle \leq \alpha}{\arg\min} \sum_{x_1^T \in \mathcal{V}^T} \left(Q_{x_1^T}(x_1^T) - P_{\zeta_1^T}(x_1^T)\right)_+.$$

To further consider cases where we allow distortion $\mathsf{D}(P_{X_1^T} \| Q_{X_1^T}) \leq \epsilon$ for some $\epsilon \geq 0$, we solve

$$(P_{X_1^T}^*, P_{\zeta_1^T}^*) := \underset{\substack{(P_{X_1^T}, P_{\zeta_1^T}): \\ \mathsf{D}_{\mathsf{TV}}(P_{X_1^T} \| Q_{X_1^T}) \leq \epsilon, \\ \sup_{Q_{X_1^T}} \langle Q_{X_1^T}, P_{\zeta_1^T} \rangle \leq \alpha}}{\arg\min} \mathsf{D}(P_{X_1^T}, P_{\zeta_1^T})$$

$$= \underset{\substack{(P_{X_1^T}, P_{\zeta_1^T}): \\ \mathsf{D}(P_{X_1^T} \| Q_{X_1^T}) \leq \epsilon, \\ \sup_{Q_{X_1^T}} \langle Q_{X_1^T}, P_{\zeta_1^T} \rangle \leq \alpha}}{\arg\min} \sum_{x_1^T \in \mathcal{V}^T} \left(P_{x_1^T}(x_1^T) - P_{\zeta_1^T}(x_1^T)\right)_+,$$

and plug them into (11).

**Special case ($\mathcal{V}^T \subseteq \mathcal{S} \subseteq \mathcal{Z}^T$ and $P_{\zeta_1^T} = \mathrm{Unif}(\mathcal{S})$).** For any $\zeta_1^T \in \mathcal{S}$, $P_{\zeta_1^T}(\zeta_1^T) = \frac{1}{|\mathcal{S}|}$. To ensure that the false alarm constraint is satisfied, we require $\alpha \geq \sup_{Q_{X_1^T}} \sum_{x_1^T} Q_{X_1^T}(x_1^T) \cdot \frac{1}{|\mathcal{S}|} = \frac{1}{|\mathcal{S}|}$. In other words, to enforce lower false alarm probability, we need to increase the size of $\mathcal{S}$. The minimum Type-II error probability is given by

$$\mathsf{D}_{\mathsf{TV}}(Q_{X_1^T}, \mathrm{Unif}(\mathcal{S})) = \sum_{x_1^T \in \mathcal{V}^T} \left(Q_{X_1^T}(x_1^T) - \frac{1}{|\mathcal{S}|}\right)_+.$$

If $|\mathcal{S}| = \frac{1}{\alpha}$, this minimum Type-II error is equal to the optimal result $\sum_{x_1^T \in \mathcal{V}^T}(Q_{X_1^T}(x_1^T) - \alpha)_+$. Otherwise, if $|\mathcal{S}| > \frac{1}{\alpha}$, this Type-II error is larger and the gap represents the price paid by using the uniform distribution $P_{\zeta_1^T}$, i.e., sending pseudorandom numbers.

$\square$

## F CONSTRUCTION OF TOKEN-LEVEL OPTIMAL WATERMARKING SCHEME

The toke-level optimal watermarking scheme is the optimal solution to the following optimization problem:

$$\inf_{P_{X_t, \zeta_t | X_1^{t-1}, \zeta_1^{t-1}}} \mathbb{E}_{P_{X_t, \zeta_t | X_1^{t-1}, \zeta_1^{t-1}}}[1 - \mathbb{1}\{X_t = g(\zeta_t)\}]$$

$$\text{s.t. } \sup_{Q_{X_t | X_1^{t-1}}} \mathbb{E}_{Q_{X_t | X_1^{t-1}} \otimes P_{\zeta_t | \zeta_1^{t-1}}}[\mathbb{1}\{X_t = g(\zeta_t)\}] \leq \eta, \ \mathsf{D}_{\mathsf{TV}}(P_{X_t | X_1^{t-1}}, Q_{X_t | X_1^{t-1}}) \leq \epsilon.$$

The optimal solution $P_{X_t, \zeta_t | X_1^{t-1}, \zeta_1^{t-1}}^*$ follows the similar rule as that of $P_{X_1^T, \zeta_1^T}^*$ in Theorem 2 with $(Q_{X_1^T}, P_{X_1^T}, \alpha)$ replaced by $(Q_{X_t | X_1^{t-1}}, P_{X_t | X_1^{t-1}}, \eta)$. We refer readers to Appendix D for further details.

## G   FORMAL STATEMENT OF LEMMA 4 AND ITS PROOF

Let $P^{\text{token}*}_{X_1^T, \zeta_1^T}$ and $P^{\text{token}*}_{\zeta_1^T}$ denote the joint distributions induced by the token-level optimal watermarking scheme.

**Lemma 4 (Formal)** (Token-level optimal watermarking detection errors)**.** *Let $\eta = (\alpha / \binom{T}{\lceil T\lambda \rceil})^{\frac{1}{\lceil T\lambda \rceil}}$. Under the detector $\gamma$ in (3) and the token-level optimal watermarking scheme $P^*_{X_t, \zeta_t | X_1^{t-1}, \zeta_1^{t-1}}$, the Type-I error is upper bounded by*

$$\sup_{Q_{X_1^T}} \beta_0(\gamma, Q_{X_1^T}, P^{\text{token}*}_{\zeta_1^T}) \leq \alpha.$$

*Assume that when $T$ and $n \leq T$ are both large enough, token $X_t$ is independent of $X_{t-i}$, i.e., $P_{X_t, X_{t-i}} = P_{X_t} \otimes P_{X_{t-i}}$, for all $i \geq n+1$ and $t \in [T]$. Let $\mathcal{I}_{T,n}(i) = ([i-n, i+n] \cap [T]) \backslash \{i\}$. By setting the detector threshold as $\lambda = \frac{a}{T} \sum_{t=1}^{T} \mathbb{E}_{X_t, \zeta_t}[\mathbb{1}\{X_t = g(\zeta_t)\}]$ for some $a \in [0,1]$, the Type-II error exponent is*

$$-\log \beta_1(\gamma, P^{\text{token}*}_{X_1^T, \zeta_1^T}) = \Omega\left(\frac{T}{n}\right).$$

The following is the proof of Lemma 4.

To choose $\lceil T\lambda \rceil$ indices out of $\{1, \ldots, T\}$, there are $\binom{T}{\lceil T\lambda \rceil}$ choices. Let $k = 1, \ldots, \binom{T}{\lceil T\lambda \rceil}$ and $S_k$ be the $k$-th set of the chosen indices. The Type-I error is upper bounded by

$$\beta_0(\gamma, Q_{X^{(T)}}, P^{\text{token}*}_{\zeta_1^T}) = \Pr\left(\frac{1}{T}\sum_{t=1}^{T} \mathbb{1}\{X_t = g(\zeta_t)\} \geq \lambda \mid H_0\right)$$

$$\leq \Pr\left(\bigcup_{k=1}^{\binom{T}{\lceil T\lambda \rceil}} \{\mathbb{1}\{X_t = g(\zeta_t)\} = 1, \forall t \in S_k\} \mid H_0\right)$$

$$\leq \sum_{k=1}^{\binom{T}{\lceil T\lambda \rceil}} \underbrace{\Pr\left(\{\mathbb{1}\{X_t = g(\zeta_t)\} = 1, \forall t \in S_k\} \mid H_0\right)}_{P_{\text{FA},k}}.$$

Without loss of generality, let $m = \lceil T\lambda \rceil$ and $S_k = \{1, 2, \ldots, m\}$. We can rewrite $P_{\text{FA},k}$ as

$$P_{\text{FA},k} = \mathbb{E}_{Q_{X^{(T)}} \otimes P_{\zeta^{(T)}}}[\{\mathbb{1}\{X_t = g(\zeta_t)\} = 1, \forall t \in S_k\}]$$

$$= \mathbb{E}_{Q_{X^{(T)}} \otimes P_{\zeta^{(T)}}}[\prod_{t \in S_k} \mathbb{1}\{X_t = g(\zeta_t)\}]$$

$$= \mathbb{E}_{Q_{X_1} \otimes P_{\zeta_1}}\left[\mathbb{1}\{X_1 = g(\zeta_1)\} \mathbb{E}_{Q_{X_2|X_1} \otimes P_{\zeta_2|\zeta_1}}\left[\mathbb{1}\{X_2 = g(\zeta_2)\} \cdots \right.\right.$$

$$\left.\left. \cdots \mathbb{E}_{Q_{X_m|X_1^{m-1}} \otimes P_{\zeta_m|\zeta_1^{m-1}}}[\mathbb{1}\{X_m = g(\zeta_m)\}] \cdots \right]\right]$$

$$\leq \eta^m, \quad \forall Q_{X_1^T}.$$

Then the Type-I error is finally upper bounded by

$$\sup_{Q_{X_1^T}} \beta_0(\gamma, Q_{X_1^T}, P^{\text{token}*}_{\zeta_1^T}) \leq \binom{T}{\lceil T\lambda \rceil} \eta^{\lceil T\lambda \rceil} \leq \alpha.$$

We prove the Type-II error bound by applying Janson (1998, Theorem 10).

**Theorem 8** (Theorem 10, Janson (1998))**.** *Let $\{I_i\}_{i \in \mathcal{I}}$ be a finite family of indicator random variables, defined on a common probability space. Let $G$ be a dependency graph of $\mathcal{I}$, i.e., a graph with vertex set $\mathcal{I}$ such that if $A$ and $B$ are disjoint subsets of $\mathcal{I}$, and $\Gamma$ contains no edge between $A$ and*

*B, then $\{I_i\}_{i \in A}$ and $\{I_i\}_{i \in B}$ are independent. We write $i \sim j$ if $i, j \in \mathcal{I}$ and $(i, j)$ is an edge in G. In particular, $i \nsim i$. Let $S = \sum_{i \in \mathcal{I}} I_i$ and $\Delta = \mathbb{E}[S]$. Let $\Psi = \max_{i \in \mathcal{I}} \sum_{j \in \mathcal{I}, j \sim i} \mathbb{E}[I_j]$ and $\Phi = \frac{1}{2} \sum_{i \in \mathcal{I}} \sum_{j \in \mathcal{I}, j \sim i} \mathbb{E}[I_i I_j]$. For any $0 \leq a \leq 1$,*

$$\Pr(S \leq a\Delta) \leq \exp\left\{ -\min\left\{ (1-a)^2 \frac{\Delta^2}{8\Phi + 2\Delta}, (1-a)\frac{\Delta}{6\Psi} \right\} \right\}. \tag{12}$$

Given any detector $\gamma$ that accepts the form in (3) and the corresponding optimal watermarking scheme, for some $a \in (0, 1)$, we first set the threshold in $\gamma$ as

$$T\lambda = a \sum_{t=1}^{T} \mathbb{E}_{X_t, \zeta_t}[\mathbb{1}\{X_t = g(\zeta_t)\}] = a \sum_{t=1}^{T} \mathbb{E}_{X_1^{t-1}}\left[ \sum_x \left( P^*_{X_t|X_1^{t-1}}(x|X_1^{t-1}) - \eta \right)_+ \right] =: a\Delta_T,$$

where $P^*_{X_t|X_1^{t-1}}$ is induced by $P^*_{X_t, \zeta_t|X_1^{t-1}, \zeta_1^{t-1}}$. The Type-II error is given by

$$\beta_1(\gamma, P^{\text{token}*}_{X_1^T, \zeta_1^T}) = P^{\text{token}*}_{X_1^T, \zeta_1^T}\left( \sum_{t=1}^{T} \mathbb{1}\{X_t = g(\zeta_t)\} < a\Delta_T \right)$$

which is exactly the left-hand side of (12).

Assume that when $T$ and $n \leq T$ are large enough, token $X_t$ is independent of all $X_{t-i}$ for all $i \geq n+1$ and $t \in [T]$, i.e., $P_{X_t, X_{t-i}} = P_{X_t} \otimes P_{X_{t-i}}$. Let $\mathcal{I}_{T,n}(i) = ([i-n, i+n] \cap [T])\backslash\{i\}$. The $\Psi$ and $\Phi$ on the right-hand side of (12) are given by:

$$\Psi := \max_{i \in [T]} \sum_{t \in [T], t \sim i} \mathbb{E}_{X_t, \zeta_t}[\mathbb{1}\{X_t = g(\zeta_t)\}] = \max_{i \in [T]} \sum_{t \in \mathcal{I}_{T,n}(i)} \mathbb{E}_{X_t, \zeta_t}[\mathbb{1}\{X_t = g(\zeta_t)\}] = \Theta(n),$$

$$\Phi := \frac{1}{2} \sum_{i \in [T]} \sum_{j \in [T], j \sim i} \mathbb{E}[\mathbb{1}\{X_i = g(\zeta_i)\}\mathbb{1}\{X_j = g(\zeta_j)\}]$$

$$= \frac{1}{2} \sum_{i \in [T]} \sum_{j \in \mathcal{I}_{T,n}(i)} \mathbb{E}[\mathbb{1}\{X_i = g(\zeta_i)\}\mathbb{1}\{X_j = g(\zeta_j)\}] = \Theta(Tn).$$

By plugging $\Delta_T$, $\Omega$ and $\Theta$ back into the right-hand side of (12), we have the upper bound

$$\beta_1(\gamma, P^{\text{token}*}_{X_1^T, \zeta_1^T}) \leq \exp\left\{ -\min\left\{ (1-a)^2 \frac{\Delta_T^2}{8\Phi + 2\Delta_T}, (1-a)\frac{\Delta_T}{6\Psi} \right\} \right\}$$

where $U_t = \mathbb{E}_{X_1^{t-1}}\left[ \sum_x \left( P^*_{X_t|X_1^{t-1}}(x|X_1^{t-1}) - \eta \right)_+ \right]$, $\Delta_T := \sum_{t=1}^{T} U_t$, $\Psi = \max_{i \in [T]} \sum_{t \in \mathcal{I}_{T,n}(i)} U_t$, and $\Phi = \frac{1}{2} \sum_{i \in [T]} \sum_{j \in \mathcal{I}_{T,n}(i)} \mathbb{E}[\mathbb{1}\{X_i = g(\zeta_i)\}\mathbb{1}\{X_j = g(\zeta_j)\}]$. This implies

$$-\log \beta_1(\gamma, P^{\text{token}*}_{X_1^T, \zeta_1^T}) \geq \min\left\{ (1-a)^2 \Theta\left(\frac{T}{n}\right), (1-a)\Theta\left(\frac{T}{n}\right) \right\}$$

$$\implies -\log \beta_1(\gamma, P^{\text{token}*}_{X_1^T, \zeta_1^T}) = \Omega\left(\frac{T}{n}\right).$$

# H DIAGRAM OF PRACTICAL WATERMARKING GENERATION AND DETECTION ALGORITHMS

In Figure 4, we show an illustration of how our designed algorithms work in practice. We leverage Gumbel-max trick and SLM to recover the auxiliary sequences $\zeta_1^T$ to ensure high detection accuracy.

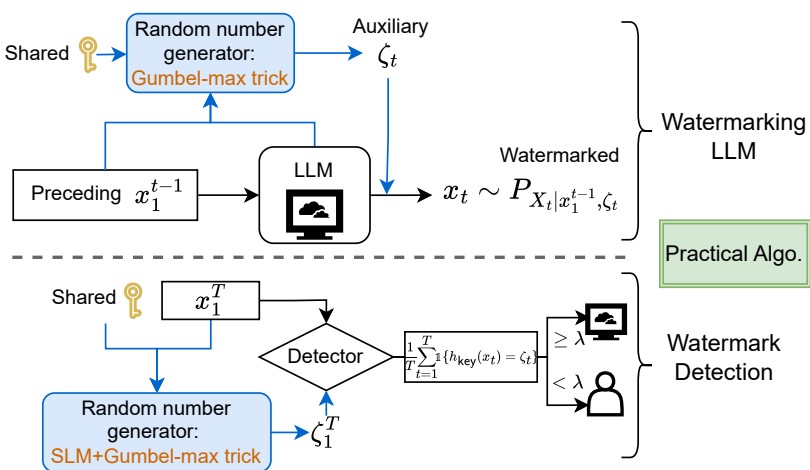

Figure 4: Diagram of practical algorithms for watermark generation and detection.

# I  PROOF OF THEOREM 6

According to the Type-I error constraint, we have $\forall x_1^T \in \mathcal{V}^T$,

$$
\alpha \geq \max_{Q_{X_1^T}} \mathbb{E}_{Q_{X_1^T} \otimes P_{\zeta_1^T}} \left[ \sup_{\tilde{x}_1^T \in \mathcal{B}_f(X_1^T)} \mathbb{1}\{\gamma(\tilde{x}_1^T, \zeta_1^T) = 1\} \right]
$$

$$
\geq \mathbb{E}_{\delta_{x_1^T} \otimes P_{\zeta_1^T}} \left[ \sup_{\tilde{x}_1^T \in \mathcal{B}_f(X_1^T)} \mathbb{1}\{\gamma(\tilde{x}_1^T, \zeta_1^T) = 1\} \right] = \mathbb{E}_{P_{\zeta_1^T}} \left[ \sup_{\tilde{x}_1^T \in \mathcal{B}_f(x_1^T)} \gamma(\tilde{x}_1^T, \zeta_1^T) \right]
$$

$$
= \sum_{\zeta_1^T} P_{\zeta_1^T}(\zeta_1^T) \sup_{\tilde{x}_1^T \in \mathcal{B}_f(x_1^T)} \gamma(\tilde{x}_1^T, \zeta_1^T).
$$

For brevity, let $\mathcal{B}(k) := \mathcal{B}_f(x_1^T)$ if $f(x_1^T) = k$. The $f$-robust Type-II error is equal to $1 - \mathbb{E}_{P_{X_1^T, \zeta_1^T}}[\inf_{\tilde{x}_1^T \in \mathcal{B}_f(X_1^T)} \gamma(\tilde{x}_1^T, \zeta_1^T)]$. We have

$$
\mathbb{E}_{P_{X_1^T, \zeta_1^T}} \left[ \inf_{\tilde{x}_1^T \in \mathcal{B}_f(X_1^T)} \gamma(\tilde{x}_1^T, \zeta_1^T) \right] \leq \mathbb{E}_{P_{X_1^T, \zeta_1^T}} \left[ \sup_{\tilde{x}_1^T \in \mathcal{B}_f(X_1^T)} \gamma(\tilde{x}_1^T, \zeta_1^T) \right]
$$

$$
= \sum_{k \in [K]} \underbrace{\sum_{x_1^T : f(x_1^T) = k} \sum_{\zeta_1^T} P_{X_1^T, \zeta_1^T}(x_1^T, \zeta_1^T) \sup_{\tilde{x}_1^T \in \mathcal{B}_f(x_1^T)} \gamma(\tilde{x}_1^T, \zeta_1^T)}_{C(k)},
$$

where according to the $f$-robust Type-I error constraint, for all $k \in [K]$,

$$
C(k) \leq \sum_{x_1^T : f(x_1^T) = k} P_{X_1^T}(x_1^T), \quad \text{and}
$$

$$
C(k) = \sum_{\zeta_1^T} P_{\zeta_1^T}(\zeta_1^T) \sum_{x_1^T : f(x_1^T) = k} P_{X_1^T | \zeta_1^T}(x_1^T | \zeta_1^T) \sup_{\tilde{x}_1^T \in \mathcal{B}(k)} \gamma(\tilde{x}_1^T, \zeta_1^T)
$$

$$
\leq \sum_{\zeta_1^T} P_{\zeta_1^T}(\zeta_1^T) \sup_{\tilde{x}_1^T \in \mathcal{B}(k)} \gamma(\tilde{x}_1^T, \zeta_1^T) \leq \alpha.
$$

Therefore,

$$\mathbb{E}_{P_{X_1^T,\zeta_1^T}}\left[\inf_{\tilde{x}_1^T\in\mathcal{B}(f(X_1^T))}\gamma(\tilde{x}_1^T,\zeta_1^T)\right]\leq\sum_{k\in[K]}C(k)$$

$$\leq\sum_{k\in[K]}\left(\left(\sum_{x_1^T:f(x_1^T)=k}P_{X_1^T}(x_1^T)\right)\wedge\alpha\right)=1-\sum_{k\in[K]}\left(\left(\sum_{x_1^T:f(x_1^T)=k}P_{X_1^T}(x_1^T)\right)-\alpha\right)_+ \quad (13)$$

where (13) is maximized by taking

$$P_{X_1^T}=P_{X_1^T}^{*,f}:=\operatorname*{arg\,min}_{P_{X_1^T}:\mathsf{D}(P_{X_1^T},Q_{X_1^T})\leq\epsilon}\sum_{k\in[K]}\left(\left(\sum_{x_1^T:f(x_1^T)=k}P_{X_1^T}(x_1^T)\right)-\alpha\right)_+.$$

For any $P_{X_1^T}$, the $f$-robust Type-II error is lower bounded by

$$\mathbb{E}_{P_{X_1^T,\zeta_1^T}}\left[\sup_{\tilde{x}_1^T\in\mathcal{B}_f(X_1^T)}\mathbb{1}\{\gamma(\tilde{x}_1^T,\zeta_1^T)=0\}\right]\geq\sum_{k\in[K]}\left(\left(\sum_{x_1^T:f(x_1^T)=k}P_{X_1^T}(x_1^T)\right)-\alpha\right)_+.$$

By plugging $P_{X_1^T}^{*,f}$ into the lower bound, we obtain the universal minimum $f$-robust Type-II error over all possible $\gamma$ and $P_{X_1^T,\zeta_1^T}$, denoted by

$$\beta_1^*(f,Q_{X_1^T},\epsilon,\alpha):=\min_{P_{X_1^T}:\mathsf{D}(P_{X_1^T},Q_{X_1^T})\leq\epsilon}\sum_{k\in[K]}\left(\left(\sum_{x_1^T:f(x_1^T)=k}P_{X_1^T}(x_1^T)\right)-\alpha\right)_+. \quad (14)$$

## J  OPTIMAL TYPE OF $f$-ROBUST DETECTORS AND WATERMARKING SCHEMES

**Theorem 9** (Optimal type of $f$-robust detectors and watermarking schemes). *Let $\Gamma_f^*$ be a collection of detectors that accept the form*

$$\gamma(X_1^T,\zeta_1^T)=\mathbb{1}\{X_1^T=g(\zeta_1^T)\text{ or }f(X_1^T)=g(\zeta_1^T)\}$$

*for some function $g:\mathcal{Z}^T\to\mathcal{S}$, $\mathcal{S}\cap([K]\cup\mathcal{V}^T)\neq\emptyset$ and $|\mathcal{S}|>K$. If an only if the detector $\gamma\in\Gamma_f^*$, the minimum Type-II error attained from (Opt-R) reaches $\beta_1^*(Q_{X_1^T},\epsilon,\alpha,f)$ in (14) for all text distribution $Q_{X_1^T}\in\mathcal{P}(\mathcal{V}^T)$ and distortion level $\epsilon\in\mathbb{R}_{\geq0}$.*

*After enlarging $\mathcal{Z}^T$ to include redundant auxiliary values, the $\epsilon$-distorted optimal $f$-robust watermarking scheme $P_{X_1^T,\zeta_1^T}^{*,f}(x_1^T,\zeta_1^T)$ is given as follows:*

$$P_{X_1^T}^{*,f}:=\operatorname*{arg\,min}_{P_{X_1^T}:\mathsf{D}_{\mathsf{TV}}(P_{X_1^T},Q_{X_1^T})\leq\epsilon}\sum_{k\in[K]}\left(\left(\sum_{x_1^T:f(x_1^T)=k}P_{X_1^T}(x_1^T)\right)-\alpha\right)_+,$$

*and for any $x_1^T\in\mathcal{V}^T$,*

*1) for all $\zeta_1^T$ s.t. $\sup_{\tilde{x}_1^T\in\mathcal{B}(f(x_1^T))}\gamma(\tilde{x}_1^T,\zeta_1^T)=1$: $P_{\zeta_1^T|X_1^T}^{*,f}(\zeta_1^T|x_1^T)$ satisfies*

$$\sum_{\tilde{x}_1^T\in\mathcal{B}_f(x_1^T)}P_{X_1^T}^{*,f}(\tilde{x}_1^T)\sum_{\zeta_1^T}P_{\zeta_1^T|X_1^T}^{*,f}(\zeta_1^T|\tilde{x}_1^T)\sup_{\tilde{x}_1^T\in\mathcal{B}_f(x_1^T)}\gamma(\tilde{x}_1^T,\zeta_1^T)=\left(\sum_{\tilde{x}_1^T\in\mathcal{B}_f(x_1^T)}P_{X_1^T}^{*,f}(\tilde{x}_1^T)\right)\wedge\alpha.$$

*2) $\forall\zeta_1^T$ s.t. $|\{x_1^T\in\mathcal{V}^T:\gamma(x_1^T,\zeta_1^T)=1\}|=0$: $P_{X_1^T,\zeta_1^T}^{*,f}(x_1^T,\zeta_1^T)$ satisfies*

$$\sum_{\tilde{x}_1^T\in\mathcal{B}_f(x_1^T)}P_{X_1^T}^{*,f}(x_1^T)\sum_{\zeta_1^T:|\{x_1^T:\gamma(x_1^T,\zeta_1^T)=1\}|=0}P_{\zeta_1^T|X_1^T}^{*,f}(\zeta_1^T|x_1^T)=\left(\left(\sum_{\tilde{x}_1^T\in\mathcal{B}_f(x_1^T)}P_{X_1^T}^{*,f}(\tilde{x}_1^T)\right)-\alpha\right)_+.$$

*3) all other cases of $\zeta_1^T$: $P_{X_1^T,\zeta_1^T}^{*,f}(x_1^T,\zeta_1^T)=0$.*

*Proof of Theorem 9.* When $f$ is an identity mapping, it is equivalent to Theorem 2. When $f : \mathcal{V}^T \to [K]$ is some other function, following from the proof of Theorem 2, we consider three cases.

- **Case 1:** $\mathcal{S} \cap ([K] \cup \mathcal{V}^T) \neq \emptyset$ but $|\mathcal{S}| < K$. It is impossible for the detector to detect all the watermarked text sequences. That is, there exist $\tilde{x}_1^T$ such that for all $\zeta_1^T$, $\gamma(\tilde{x}_1^T, \zeta_1^T) = 0$. Under this case, in Appendix I, $C(f(\tilde{x}_1^T)) = 0 \neq (\sum_{x_1^T : f(x_1^T) = f(\tilde{x}_1^T)} P_{X_1^T}(x_1^T)) \wedge \alpha$, which means the $f$-robust Type-II error cannot reach the lower bound.

- **Case 2:** $\mathcal{S} \cap ([K] \cup \mathcal{V}^T) \neq \emptyset$ but $|\mathcal{S}| = K$. Under this condition, the detector needs to accept the form $\gamma(X_1^T, \zeta_1^T) = \mathbb{1}\{f(X_1^T) = g(\zeta_1^T)\}$ so as to detect all possible watermarked text. Otherwise, it will degenerate to Case 1. We can see $f(X_1^T)$ as an input variable and rewrite the detector as $\gamma'(f(X_1^T), \zeta_1^T) = \gamma(X_1^T, \zeta_1^T) = \mathbb{1}\{f(X_1^T) = g(\zeta_1^T)\}$. Similar the proof technique of Theorem 2, it can be shown that $C(k)$ in Appendix I cannot equal $(\sum_{x_1^T : f(x_1^T) = k} P_{X_1^T}(x_1^T)) \wedge \alpha$ for all $k \in [K]$, while the worst-case $f$-robust Type-I error remains upper bounded by $\alpha$ for all $Q_{X_1^T}$ and $\epsilon$.

- **Case 3:** Let $\Xi_\gamma(x_1^T) := \{\zeta_1^T \in \mathcal{Z}^T : \gamma(x_1^T, \zeta_1^T) = 1\}$. $\exists x_1^T, y_1^T \in \mathcal{V}^T$, s.t. $f(x_1^T) \neq f(y_1^T)$ and $\Xi_\gamma(x_1^T) \cap \Xi_\gamma(y_1^T) \neq \emptyset$. For any detector $\gamma \notin \Gamma_f^*$ that does not belong to Cases 1 and 2, it belongs to Case 3. Let us start from a simple case where $T = 1$, $\mathcal{V} = \{x_1, x_2, x_3\}$, $K = 2$, $\mathcal{Z} = \{\zeta_1, \zeta_2, \zeta_3\}$, and $\mathcal{S} = [2]$. Consider the mapping $f$ and the detector as follows: $f(x_1) = f(x_2) = 1$, $f(x_3) = 2$, $\gamma(x_1, \zeta_1) = \gamma(x_1, \zeta_1) = 1$, $\gamma(x_3, \zeta_2) = 1$, and $\gamma(x, \zeta) = 0$ for all other pairs $(x, \zeta)$. When $C(k) = (\sum_{x_1^T : f(x_1^T) = k} P_{X_1^T}(x_1^T)) \wedge \alpha$ for all $k \in [K]$, i.e.,

$$P_{X,\zeta}(x_1, \zeta_1) + P_{X,\zeta}(x_1, \zeta_2) + P_{X,\zeta}(x_2, \zeta_1) + P_{X,\zeta}(x_2, \zeta_2) = (P_X(x_1) + P_X(x_2)) \wedge \alpha,$$

and $\quad P_{X,\zeta}(x_3, \zeta_2) = P_X(x_3) \wedge \alpha,$

then the worst-case $f$-robust Type-I error is lower bounded by

$$\max_{Q_{X_1^T}} \mathbb{E}_{Q_{X_1^T} \otimes P_{\zeta_1^T}} \left[ \sup_{\tilde{x}_1^T \in \mathcal{B}_f(X_1^T)} \mathbb{1}\{\gamma(\tilde{x}_1^T, \zeta_1^T) = 1\} \right]$$

$$\geq \mathbb{E}_{P_{\zeta_1^T}} \left[ \sup_{\tilde{x}_1^T \in \mathcal{B}(1)} \mathbb{1}\{\gamma(\tilde{x}_1^T, \zeta_1^T) = 1\} \right]$$

$$= (P_X(x_1) + P_X(x_2)) \wedge \alpha + P_X(x_3) \wedge \alpha$$

$$> \alpha, \quad \text{if } P_X(x_1) + P_X(x_2) > \alpha \text{ or } P_X(x_3) > \alpha.$$

Thus, for any $Q_X$ such that $\{P_X \in \mathcal{P}(\mathcal{V}) : \mathsf{D}_{\mathsf{TV}}(P_X, Q_X) \leq \epsilon\} \subseteq \{P_X \in \mathcal{P}(\mathcal{V}) : P_X(x_1) + P_X(x_2) > \alpha \text{ or } P_X(x_2) > \alpha\}$, the false-alarm constraint is violated when $C(k) = (\sum_{x_1^T : f(x_1^T) = k} P_{X_1^T}(x_1^T)) \wedge \alpha$ for all $k \in [K]$. The result can be generalized to larger $(T, \mathcal{V}, \mathcal{Z}, K, \mathcal{S})$, other functions $f$ and other detectors that belong to Case 3.

In conclusion, if and only if $\gamma \in \Gamma^*$, the minimum Type-II error attained from (Opt-R) reaches the universal minimum $f$-robust Type-II error $\beta_1^*(f, Q_{X_1^T}, \epsilon, \alpha)$ in (14) for all $Q_{X_1^T} \in \mathcal{P}(\mathcal{V}^T)$ and $\epsilon \in \mathbb{R}_{\geq 0}$.

Under the watermarking scheme $P_{X_1^T, \zeta_1^T}^{*,f}$, the $f$-robust Type-I and Type-II errors are given by:

**$f$-robust Type-I error:**

$$\because \forall y_1^T \in \mathcal{V}^T, \quad \mathbb{E}_{P_{\zeta_1^T}^{*,f}} \left[ \sup_{\tilde{x}_1^T \in \mathcal{B}_f(y_1^T)} \mathbb{1}\{\gamma(\tilde{x}_1^T, \zeta_1^T) = 1\} \right]$$

$$= \sum_{\zeta_1^T} \sum_{x_1^T} P_{X_1^T, \zeta_1^T}^{*,f}(x_1^T, \zeta_1^T) \sup_{\tilde{x}_1^T \in \mathcal{B}_f(y_1^T)} \mathbb{1}\{\gamma(\tilde{x}_1^T, \zeta_1^T) = 1\}$$

$$= \sum_{x_1^T \in \mathcal{B}_f(y_1^T)} P_{X_1^T}^{*,f}(x_1^T) \sum_{\zeta_1^T} P_{\zeta_1^T | X_1^T}^{*,f}(\zeta_1^T | x_1^T) \sup_{\tilde{x}_1^T \in \mathcal{B}_f(y_1^T)} \mathbb{1}\{\gamma(\tilde{x}_1^T, \zeta_1^T) = 1\}$$

$$= \left( \sum_{x_1^T \in \mathcal{B}_f(y_1^T)} P_{X_1^T}^{*,f}(x_1^T) \right) \wedge \alpha \leq \alpha,$$

and since any distribution $Q_{X_1^T}$ can be written as a linear combinations of $\delta_{y_1^T}$,

$$\therefore \sup_{Q_{X_1^T}} \mathbb{E}_{Q_{X_1^T} P_{\zeta_1^T}^{*,f}} \left[ \sup_{\tilde{x}_1^T \in \mathcal{B}_f(X_1^T)} \mathbb{1}\{\gamma(\tilde{x}_1^T, \zeta_1^T) = 1\} \right] \leq \alpha.$$

**$f$-robust Type-II error:**

$$1 - \mathbb{E}_{P_{X_1^T, \zeta_1^T}^{*,f}} \left[ \sup_{\tilde{x}_1^T \in \mathcal{B}_f(X_1^T)} \mathbb{1}\{\gamma(\tilde{x}_1^T, \zeta_1^T) = 1\} \right]$$

$$= 1 - \sum_{x_1^T} \sum_{\zeta_1^T} P_{X_1^T, \zeta_1^T}^{*,f}(x_1^T, \zeta_1^T) \sup_{\tilde{x}_1^T \in \mathcal{B}_f(x_1^T)} \mathbb{1}\{\gamma(\tilde{x}_1^T, \zeta_1^T) = 1\}$$

$$= 1 - \sum_{k \in [K]} \sum_{x_1^T \in \mathcal{B}(k)} P_{X_1^T}^{*,f}(x_1^T) \sum_{\zeta_1^T} P_{\zeta_1^T|X_1^T}^{*,f}(\zeta_1^T|x_1^T) \sup_{\tilde{x}_1^T \in \mathcal{B}(k)} \mathbb{1}\{\gamma(\tilde{x}_1^T, \zeta_1^T) = 1\}$$

$$= 1 - \sum_{k \in [K]} \left( \left( \sum_{x_1^T \in \mathcal{B}(k)} P_{X_1^T}^{*,f}(x_1^T) \right) \wedge \alpha \right)$$

$$= \sum_{k \in [K]} \left( \left( \sum_{x_1^T \in \mathcal{B}(k)} P_{X_1^T}^{*,f}(x_1^T) \right) - \alpha \right)_+.$$

The optimality of $P_{X_1^T, \zeta_1^T}^{*,f}$ is thus proved. □

Figure 5 compares the universally minimum Type-II errors with and without semantic-invariant text modification.

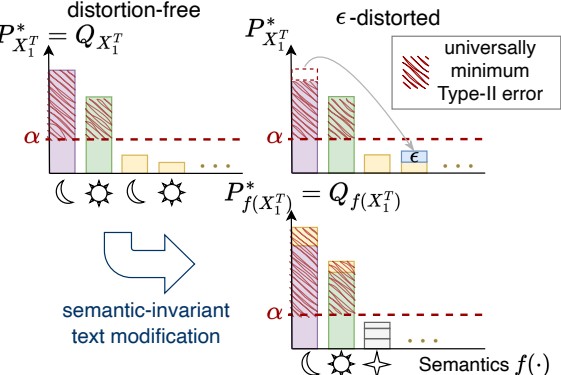

Figure 5: Universally minimum Type-II error w/o distortion and with semantic-invariant text modification.

## K  IMPLEMENTATION OF WATERMARKING SCHEME WITH UNIFORM $P_{\zeta_t}$

### K.1  ALGORITHM DESCRIPTION

Algorithm 3 describes the optimal watermarking scheme with uniform $P_{\zeta_t}$. We first uniformly sample $\zeta_t$ from $\mathcal{Z} = \{h_{\text{key}}(x)\}_{x \in \mathcal{V}}$. Then, with the sampled $\zeta_t$, we can derive the new NTP distribution such that $P_{X_t|x_1^{t-1},u}(x) = |\mathcal{V}| \min\{Q_{X_t|x_1^{t-1},u}(x), \frac{1}{|\mathcal{V}|}\}$ for $h_{\text{key}}(x) = \zeta_t$, while

$$P_{X_t|x_1^{t-1},u}(x) = \frac{|\mathcal{V}| \left( Q_{X_t|x_1^{t-1},u}(x) - \frac{1}{|\mathcal{V}|} \right)_+ \cdot \left( \frac{1}{|\mathcal{V}|} - Q_{X_t|x_1^{t-1},u}(h_{\text{key}}^{-1}(\zeta_t)) \right)_+}{\mathsf{D}_{\mathsf{TV}}(Q_{X_t|x_1^{t-1},u}, \mathrm{Unif}(\mathcal{V}))}$$ otherwise. Next token is then

sampled from obtained $P_{X_t|x_1^{t-1},u}(x)$.

Algorithm 4 outlines the corresponding detection process. For any given suspicious text, we analyze each token sequentially, mirroring the generation process. First, we uniformly sample $\zeta_t$ using previous tokens as a hash. Then, we compute the score as $\frac{1}{T}\sum_{t=1}^{T} \mathbb{1}\{h_{\texttt{key}}(x_t) = \zeta_t\}$. Any text with a score greater than a threshold $\lambda \in (0, 1)$, will be classified as watermarked.

---

**Algorithm 3** Watermarked Text Generation with Uniform $P_{\zeta_t}$

---

**Input:** Language Model $Q$, Vocabulary $\mathcal{V}$, Prompt $u$, Secret $\texttt{key}$, Token-level False alarm $\eta$

1: $\mathcal{Z} \leftarrow \{h_{\texttt{key}}(x)\}_{x \in \mathcal{V}}$

2: **for** $t = 1, \ldots, T$ **do**

3:     Compute a hash of previous $n$ tokens, and use it as a seed to uniformly sample $\zeta_t$ from $\mathcal{Z}$.

4:     $P_{X_t | x_1^{t-1}, u}(x) = \begin{cases} |\mathcal{V}| \min\{Q_{X_t|x_1^{t-1},u}(x), \frac{1}{|\mathcal{V}|}\}, & \text{if } h_{\texttt{key}}(x) = \zeta_t; \\ \dfrac{|\mathcal{V}|\left(Q_{X_t|x_1^{t-1},u}(x) - \frac{1}{|\mathcal{V}|}\right)_+ \cdot \left(\frac{1}{|\mathcal{V}|} - Q_{X_t|x_1^{t-1},u}(h_{\texttt{key}}^{-1}(\zeta_t))\right)_+}{\mathsf{D}_{\mathsf{TV}}(Q_{X_t|x_1^{t-1},u}, \mathrm{Unif}(\mathcal{V}))}, & \text{otherwise,} \end{cases}$

5:     Sample $x_t \sim P_{X_t|x_1^{t-1},u}$

6: **end for**

**Output:** Watermarked text $x_1^T = (x_1, \ldots, x_T)$.

---

**Algorithm 4** Watermarked Text Detection with Uniform $P_{\zeta_t}$

---

**Input:** Language Model $Q$, Vocabulary $\mathcal{V}$, Prompt $u$, Secret $\texttt{key}$, Token-level False alarm $\eta$

1: $\mathcal{Z} \leftarrow \{h_{\texttt{key}}(x)\}_{x \in \mathcal{V}}$

2: score $= 0$

3: **for** $t = 1, \ldots, T$ **do**

4:     Compute a hash of previous $n$ tokens, and use it as a seed to uniformly sample $\zeta_t$ from $\mathcal{V}$.

5:     score $=$ score $+ \mathbb{1}\{h_{\texttt{key}}(x_t) = \zeta_t\}$

6:     **if** score $> T\lambda$ **then**

7:         **return** 1              ▷ Input text is watermarked

8:     **else**

9:         **return** 0              ▷ Input text is unwatermarked

10:     **end if**

11: **end for**

---

