# OpenReview forum: "Universally Optimal Watermarking Schemes for LLMs: from Theory to Practice"
_ICLR.cc/2025/Conference — Submitted to ICLR 2025_

### Official Review · Reviewer_sTiD · 2024-10-23

**Soundness:** 3
**Presentation:** 3
**Contribution:** 3
**Rating:** 6
**Confidence:** 5

**Summary:**

This paper finds the optimal watermarking scheme for LLM-generated texts. This is not practical therefore they proposed suboptimal but more practical schemes.

**Strengths:**

**S1. Awesome theoretical derivations**

**S2. Quite a readable paper despite theoretical with high technicality**

**Weaknesses:**

**W1. Is this watermarking?**

The main part of the submission deals with the analysis of a problem that is not watermarking per se. Therefore, it is wrong to claim "*we characterize the entire class of detectors and watermarking schemes that are universally optimal*".

The root of the problem is the non-causality of the scheme which samples jointly the text and the `auxiliary data' for detecting it. Then this data is magically transferred to the detector. This is not practical, so this is not watermarking. The auxiliary should be generated first (from the key and maybe *previous* X), and then the watermarked text X is generated.

I am referring to:
- line 182: "*We assume the auxiliary sequence can be fully recovered from X and the key*".
- line 304: "*...the detector having full access to $\zeta$*". This is not the problem. The problem is that this $\zeta$ is generated along with $X$. $\zeta$ should be sampled first, shared with the detector, then the embedder samples $X$ given $\zeta$.
- line 355: "*if the watermark sequence $\zeta$ is shared between the LLM and the detector.*"
- line 366: "*A challenge in implementing the optimal watermarking scheme is transmitting $\zeta$ to the detector*".
- line 368: "*This alternative watermarking ... results in higher minimum Type-II error*". I am afraid this is not an alternative. This is the only **watermarking** scheme that holds in practice.
- line 810: Again a problem of causality: $P_{\zeta|X}$ reads as first generate $X$, then forge the key $\zeta$ that allows the detection of $X$.

- line 256: Let me summarize the construction:
1. From $P_X$, we construct $P^*_X$,
2. With the help of function $g$, we construct $P^*_{X,\zeta}$ such that the joint distribution is not null only when $g(X)=\zeta$.
3. We sample from this joint distribution one occurrence of $X$ and $\zeta$.
4. Auxiliary $\zeta$ is magically transmitted to the detector.
5. Later on, the detector receives either the corresponding text $X$ or a random text $Y$.
There is little chance that $g(Y) = \zeta$ (ie. less than $\alpha$) whereas $g(X) = \zeta$ by construction (unless $\zeta$ is redundant).

I do not see the use of that scheme and I doubt that this is watermarking.
For instance, suppose that we number all the text from 1 to $|V|^T$. Function $g$ is indeed this indexation function.
Basically, the embedder samples a $X$ and tells the detector "Hey, I have generated text number $\zeta=g(X)$!". When the detector sees that text number, it shouts "Detected!". I do not see any scenario where this functionality could be useful.

However, two novel and practical **watermarking** schemes are also proposed in this paper. It is a pity that their description/benchmarking takes so little space. There is no comparison between the 2 (uniform key and SLM side-informed detector). What about the impact of the approximate distribution from the surrogate SLM. Plenty of questions left unsolved.

As for Section 6 supposedly presenting a

**W2. Limitations**

The auxiliary data is a discrete random variable. Is this a limitation? Many schemes use a continuous rv.
There seems to be a limitation as well on the level $\alpha$ in the practical schemes which is not clearly stated.

**W3. Experimental setup**

- line 426: Surrogate models are not really SLMs. 7B parameters, this is big. Remove "*both*".
- line 428: Given $\eta$ and $T$, $\alpha$ is a function of $\lambda$. I thought it was the other way around: given $\alpha$ and $T$, $\eta$ is a function of $\lambda$. BTW, what is the value of $\lambda$?
- line 431: There are subtle differences between EXP-edit and Gumbel-Max. EXP-edit is nothing more than Gumbel-max but with a different seeding approach (no hash window) and an enhanced decoder based on edit distance. Gumbel-Max uses a hash window. Its size is not mentioned. In practice, since no token is added or removed in the attack, the 2 schemes should share similar performances unless Gumbel-Max uses a large hash window. This is not the case here so the size of the hash window matters a lot.
- line 443: ROC-AUC is not very informative. TPR@FPR is much better, however the FPR values (1% or even 10%) are quite large for practical application.
- line 458: "*Our watermarking method demonstrates superior performance ... This success stems from the design of our watermarking scheme*". Not only. It is also because your detector is side-informed by the SLM. Maybe the other schemes could benefit from the SLM as well. In other words, the comparison is not fair.
- line 493: "*Utilizing Alg. 1, we generate 8000 watermarked sentences*". I do not understand. You are empirically measuring the FPR, so you operate under H_0; there is no need for Alg. 1, and there is no need for watermarked sentences. This is very doubtful.

**W4. Minor comments**

- line 107: You may also cite
*Optimal Watermark Embedding and Detection Strategies Under Limited Detection Resources*, Merhav and Sabbag, IEEE Trans. Inf. Theory, Jan. 2008. *On causal and semicausal codes for joint information embedding and source coding,*, N. Merhav and E. Ordentlich,  IEEE Trans. Inform. Theory, Jan. 2006.
- line 136: It is confusing to denote both the original LLM distribution (used in hypothesis H_1) and the human-generated text (used in hypothesis H_0) by Q.  Under H_1, Q is given and the embedding crafts P. Under H_0, Q is unknown and the false alarm rate is guaranteed whatever Q (line 195).
- line 404: Union with singleton {$\{\tilde{\zeta}\}$}
- Alg.2 line 3,4: Replace $P$ by $\tilde{P}$, approximation of $P$
- line 809: $x_1^T$ are missing in the equation. Equals $\epsilon$ because the TV is defined with a factor 1/2 ... It took me ages to get it.
- line 818: (8) comes from (5)... It took me ages to get it.
- line 831, line 885: sometimes $\mathcal{S}$ is a subset of  $\mathcal{V}$, sometimes $\mathcal{S}$ is a subset of  $\mathcal{Z}$. Confusing.
- line 1040: In the special case, we are back to $\epsilon=0$, right?
- line 1056: sceheme
- line 1353: Why notation $\tilde{Q}_X$? why not $P_X$ as before?

**Questions:**

**Q1. Modify the first claim**
- Are you ready to modify the 1st claim to make clear that you study a problem that is simpler than watermarking, but yet interesting because it provides a lower bound on the Type II error, and some hints about the design of a real watermarking scheme?

**Q2. Efficient watermarking scheme**
- line 399, 406, 411: How can we compute $h^{-1}(\zeta)$ efficiently since a hash function is not reversible?

**Q3. Relation between parameters**
- line 337: Definition of $\eta$. The application usually states level $\alpha$ as a requirement. How to choose $\lambda$ then?
It is unclear how $\eta$, $\alpha$, $\lambda$ are set and in which order.

**Q4. Technical detail**
- line 777: I don't get the 1st inequality. Please, explain.

---

> ### Author Response · Authors · 2024-11-21
> **To-W1&Q1. Is this watermarking?**
>
> We thank the reviewer for the insightful comments. We have addressed all the concerns as follows and made corresponding edits (highlighted in blue) in the revised manuscript.
>
> **To-W1&Q1. Is this watermarking?**
>
> We would like to address this question from the following aspects. Additionally, there seems to be an incomplete sentence at the end of **W1**, and we kindly request the reviewer to provide some clarification.
>
> 1) **Problem setup:** Distinct from traditional post-process watermarking schemes that embed watermarks after text generation, we explore in-process watermarking that embeds watermarks during generation. To make full use of LLMs, this embedding process evolves from a functional operation to modifying the next-token prediction (NTP) distribution using auxiliary random variables (i.e., the watermark information). In this procedure, given a LLM (i.e., $Q_{X_1^T}$),  we have **degrees of freedom in designing** **(1)**  the joint distribution $P_{X_1^T, \zeta_1^T}$ (including $P_{X_1^T}$, $P_{\zeta_1^T}$ and $P_{X_1^T|\zeta_1^T}$), and **(2)** the detector. **This framework deviates from the classical settings where the embedded auxiliary sequence $\zeta_1^T$ is fixed or sampled from a fixed distribution.** It also differs from existing in-process watermarking schemes, which assume a fixed uniform distribution $P_{\zeta_1^T}$ and fail to fully leverage the capabilities of the LLM. By harnessing the flexibility of our framework, we derive watermarking schemes that adapt to the original LLM $Q_{X_1^T}$, optimizing the tradeoff between detection accuracy, text quality, and robustness.
>
>     To clarify that our framework does not break the causality between $X_1^T$ and $\zeta_1^T$, we modify Figure 1 to better illustrate the generic framework for LLM watermarking and detection: https://anonymous.4open.science/r/ICLR_rebuttal-5581/watermark_framework.pdf. At each position $t$, the auxiliary variable $\zeta_t$ is sampled from a **designed distribution**  using a shared secret $\mathsf{key}$, **which is usually but not necessarily dependent on the previous tokens and prompt**. The token $X_t$ is then sampled from the watermarked NTP distribution $P_{X_t|x_1^{t-1},\zeta_t}$. Thus, the watermarking process maintains the causal relationship between the watermark and the text. In the end, we obtain a pair of dependent sequences $(X_1^T,\zeta_1^T)$ that follows the joint distribution $P_{X_1^T,\zeta_1^T}$, which actually characterizes the entire watermark generation process. On the detector side, it only receives the text $X_1^T$ and the shared $\mathsf{key}$, which can be used to recover the auxiliary sequence $\zeta_1^T$. In theory, our goal is to jointly optimize $P_{X_1^T,\zeta_1^T}$ and the detector $\gamma$ that minimizes the Type-II error under the constraint of worst-case Type-I error.
>
>
>
> 2) **Practical implications:**  The main contribution of the theoretically optimal results is to provide valuable guidelines for designing near-optimal and implementable algorithms. Note that this is a more systematic approach compared to existing heuristic designs. In Theorem 2, we prove that the optimal watermarking scheme $P_{X_1^T, \zeta_1^T}^*$ should adapt to the LLM $Q$. This result explains why other approaches, such as green/red list watermarking and Gumbel-Max watermarking, which sample $\zeta_t$ from a fixed distribution, are suboptimal.  Building on this theoretical result, we design algorithms that address two practical concerns:  adapting to variable-length text generation and recovering the auxiliary sequences at the model-agnostic detector side. In Section 4, we propose a token-level optimal detector and watermarking scheme based on our sequence-level optimal result. This approach effectively adapts to variable-length text generation while maintaining strong performance guarantees. In Section 5, we propose decomposing the sampling procedure of the auxiliary sequence $\zeta_t$ using the Gumbel-max trick, enabling the Gumbel noise (generated from fixed distribution) to be shared with the detector via a secret key. For watermark detection, a surrogate language model (SLM) is used to approximate the NTP distribution $Q$, and combined with the Gumbel-max trick to extract $\zeta_1^T$ from the text (illustrated in https://anonymous.4open.science/r/ICLR_rebuttal-5581/watermark_practice.pdf).
> Guided by the theoretical framework, experimental results show that our algorithm achieves excellent detection performance and robustness, even with the approximated auxiliary sequence $\zeta_1^T$.
>
>     Therefore, we believe that the proposed theoretical framework and the derived optimality are useful and insightful in practice.

---

> > ### Comment · Reviewer_sTiD · 2024-11-23
> >
> > >To clarify that our framework does not break the causality between ...  we modify Figure 1 to better illustrate ...
> >
> > This figure depicts a scheme where tokens are generated one after another. This corresponds to Section 4 of your paper.
> > I am fine with this section.
> >
> > My comment was about Section 3 "*JOINTLY OPTIMIZE WATERMARKING SCHEME AND DETECTOR*". I do acknowledge that you succeeded in optimizing something, yet, that thing is not watermarking. It is a new scheme where the generated key can be transmitted to the detector. How shall we call it? A side-informed detection problem? Since this scheme is a relaxation of watermarking, it gives a lower bound. This is valid. But you cannot claim that you "*characterize the entire class of detectors and watermarking schemes universally optimal.*"

---

> ### Author Response · Authors · 2024-11-21
> **[Continued] To-W1&Q1. Is this watermarking?**
>
> 3) **Usage scenario of Theorem 2:** We address the problem of identifying whether a sentence is generated by an LLM using watermarking. Thus, there is **no restriction** on the information to be embedded into the text, as long as it enables reliable watermark detection. In our problem setup, we propose a generic framework that can be specialized to any of the existing in-process watermarking schemes. The optimal solution to the generic framework turns out to be seemingly "naive" where the embedded information, i.e., the auxiliary sequence, is either a token index or a null message. However, for the sole purpose of determining whether a text is watermarked, decoding this embedded information is sufficient when the shared key is available.
>
> 3) **Extension to robust watermarking against broad attacks:** In Section 6, we show that this framework can be further generalized to the scenarios with semantic-invariant attacks. We prove the theoretical optimal and robust design, which also provides valuable insights into designing practical and robust semantic-based watermarking algorithms.
>
> 4) **Clarification on the watermarking scheme with uniform $\zeta_1^T$:** In Line 368, since the uniform distribution is not the optimal marginal $P_{\zeta_1^T}^*$, there exists a theoretical detection performance loss, as we have discussed in Appendix E. Our experimental explorations confirm that the loss is non-negligible in practice, as demonstrated by a detection performance of only 0.602 TPR@1% FPR when using Llama2-13B—a result significantly lower than that achieved by our current algorithm. That is why we propose the practical token-level optimal watermarking scheme by allowing the sampling distribution of $\zeta_t$ to  adapt to NTP distributions in Sections 4 and 5.
>
> 5) **Impact of surrogate language model (SLM):** To investigate the impact of the approximate distribution from the surrogate SLM on our detection performance, we conducted experiments to compare detection performance with and without prompts attached to the watermarked text during the detection process. Moreover, we conducted experiments using both the original watermarking LLM and prompts during the detection process to evaluate detection performance under conditions where we could precisely reconstruct $P_{\zeta}$.
> These experiments were performed using LLama2-13B on the C4 dataset, both with and without adversarial attacks. The sampling temperature is set to $0.8$. The results are presented in the following tables. Notably, even when prompts are absent and the surrogate model cannot perfectly reconstruct the same $P_{\zeta}$​ as in the generation process, detection performance remains unaffected. This demonstrates the robustness of our watermarking method, regardless of whether a prompt is included during the detection phase. As discussed in our manuscript, being prompt- and model-agnostic is a highly practical and valuable property of a watermarking method. The observed detection performance loss of approximately $0.1$% to achieve this agnostic capability is minimal and represents a worthwhile trade-off for improved practicality.
>
> *Without attack:*
> | Language Model | Surrogate model | Setting        | ROC-AUC | TPR@1% FPR | TPR@10% FPR |
> |----------------|-----------------|----------------|---------|------------|-------------|
> | Llama2-13B     | Llama2-7B       | Without Prompt | 0.997   | 0.983      | 0.995       |
> | Llama2-13B     | Llama2-7B       | With Prompt    | 0.998   | 0.989      | 0.996       |
> | Llama2-13B     | Llama2-13B      | With Prompt    | 0.998   | 0.990      | 0.996       |
>
> *With token replacement attack:*
> | Language Model | Surrogate model | Setting        | ROC-AUC | TPR@1% FPR | TPR@10% FPR |
> |----------------|-----------------|----------------|---------|------------|-------------|
> | Llama2-13B     | Llama2-7B       | Without Prompt | 0.977   |  0.818     | 0.953       |
> | Llama2-13B     | Llama2-7B       | With Prompt    | 0.979   |  0.816     | 0.960       |
> | Llama2-13B     | Llama2-13B      | With Prompt    | 0.980   |  0.817     | 0.962       |

---

> > ### Author Response · Authors · 2024-11-21
> > **To-W2**
> >
> > **To-W2. Limitations (1)**
> > 1) In the problem setup, we state that the auxiliary random variable $\zeta_t$ takes values in a space $\mathcal{Z}$, which can be **either discrete or continuous**. In Theorem 1, the universally minimum Type-II error is shown to be independent of $\mathcal{Z}$, and the proof holds for both discrete and continuous cases. We have clarified this point in Section 2 "Watermarking LLM", and  Appendices B, C and D of the revised manuscript.
> >
> >    Additionally, in Theorem 2, we demonstrate that the optimal class of detectors can only take the form $\gamma(X_1^T,\zeta_1^T)=\mathbf{1}[X_1^T=g(\zeta_1^T)]$for some surjective function $g$. Given that the token vocabulary is discrete, it follows that it is sufficient to consider $\mathcal{Z}$ as a discrete space for the auxiliary variable $\zeta_t$ when deriving the corresponding optimal watermarking scheme.
> >
> >    For instance, in Figure 3, one may let $\mathcal{Z}$ be a continuous space and interpret $\{\zeta^{(1)},\ldots,\zeta^{(4)},\tilde{\zeta}\}$ as the collection of all disjoint subsets of $\mathcal{Z}$. This interpretation is, in fact, equivalent to quantization and considering a discrete auxiliary space $\mathcal{Z}$.
> >
> >
> > **To-W2. Limitations (2)& W3 “line 428, line 493” & Q3. Relation between parameters:**
> >
> > The sequence-level false alarm $\alpha$ is dependent on the chosen token-level false alarm $\eta$, the generated sequence length $T$ and the detection threshold $\lambda$. The relationship is stated in Lemma 3: if $\eta\in(0, \min\{ 1, (\alpha/\binom{T}{\lceil T\lambda \rceil})^{\frac{1}{\lceil T\lambda \rceil}}\}]$, the worst-case false alarm for a length-$T$ sequence is upper bounded by $\alpha$. In general, given $T$ and $\lambda$, a smaller $\eta$ results in a smaler $\alpha$; given $T$ and $\eta$, a higher $\lambda$ results in a smaller $\alpha$.
> >
> >   In the practical scheme, the algorithms take input $\eta$ and $T$, where we set $\eta=0.2$ and $T=200$. The detection threshold $\lambda$ is determined automatically using the roc_auc_score function, which empirically selects the optimal threshold based on the provided 500 unwatermarked and 500 watermarked sequences. The roc_auc_score function varies $\lambda$ to achieve different values of $\alpha$, ranging from 0 to 1, and reports the corresponding TPR in Tables 1 and 2. However, in Table 3, we first set a theoretical FPR $\alpha$ and obtain $\lambda$ by solving $\alpha=\binom{T}{\lceil T \lambda\rceil} \eta^{\lceil T \lambda\rceil}$. Then the empirical FPR is calculated using the corresponding $\lambda$ based on 8000 unwatermarked sentences. Table 3 shows that our theoretical guarantee can effectively control the empirical FPR.
> >
> >    We have corrected the typos and added a clarification in Section 5 “Experiment Settings.”, and below Table 3.

---

> > > ### Author Response · Authors · 2024-11-21
> > > **To-W3**
> > >
> > > **To-W3. Experimental setup**
> > >
> > > **- line 426:** Thank you for pointing out this typo, and we have removed ‘both’ from our manuscript. To solve your concern about the SLM being too large, we conduct experiments on a smaller SLM to investigate it’s impact on our detection performance. To illustrate our approach, we apply our watermarking algorithm to GPT-J-6B (a model with 6 billion parameters) and use GPT-2 Large (774 million parameters) as the surrogate model. We conduct experiments using the C4 dataset, and the results are presented in the tables below. The results demonstrate the effectiveness of our proposed watermarking method with or without attack even when using a smaller surrogate model. Notably, the surrogate model, despite having fewer parameters and lower overall capability compared to the watermarking language model, does not compromise the watermarking performance.
> > >
> > > *Without attack:*
> > > | Language Model | surrogate model | ROC-AUC | TPR@1% FPR | TPR@10% FPR |
> > > |----------------|-----------------|---------|------------|-------------|
> > > | Llama2-13B     | Llama2-7B       | 0.999   | 0.998      | 1.000       |
> > > | Mistral-8 × 7B | Mistral-7B      | 0.999   | 0.998      | 1.000       |
> > > | GPT-J-6B       | GPT-2 large     | 0.997   | 0.990      | 0.997       |
> > >
> > > *With token replacement attack:*
> > > | Language Model | surrogate model | ROC-AUC | TPR@1% FPR | TPR@10% FPR |
> > > |----------------|-----------------|---------|------------|-------------|
> > > | Llama1-13B     | Llama2-7B       | 0.989   | 0.860      | 0.976       |
> > > | Mistral-8 × 7B | Mistral-7B      | 0.990   | 0.881      | 0.966       |
> > > | GPT-J-6B       | GPT-2 large     | 0.987   | 0.892      | 0.962       |
> > >
> > >
> > >
> > > **- line 431:** We thank the reviewer for pointing out this confusion. In our experiment for the Gumbel-Max watermark, we hash the time step $t$ instead of a hash window during the generation and detection process. To align with the description in Example 3, we conduct experiments on the Gumbel-Max watermark and use a hash window during the generation and detection process. The updated robustness results for the Gumbel-Max watermark on the C4 dataset are presented in the following table. We will update our manuscript correspondingly.
> > >
> > > | Language Model |  Methods   | ROC-AUC | TPR@1% FPR | TPR@10% FPR |
> > > |----------------|------------|---------|------------|-------------|
> > > | Llama2-13B     | Gumbel-Max |  0.968  |  0.858     |  0.970      |
> > >
> > > **- line 443:** Thank you for your good suggestion. In our experiments, we generated 500 unwatermarked texts and 500 watermarked texts separately. As a result, we did not observe an extremely low False Positive Rate (FPR) under this setting. However, empirically, we change the threshold $\lambda$ to investigate the TPR and corresponding FPR. We find that when $\lambda>=0.246$, the FPR drops to $0$, while the corresponding highest achievable TPR is $0.981$. We compare the highest achievable TPR@0%FPR across different watermarking methods, and the results are presented in the following table. Our method achieves comparable performance with other baselines.
> > >
> > > | Watermarking Method          | KGW-1 | EXP-edit | Gumble-Max | Ours  |
> > > |------------------------------|-------|----------|------------|-------|
> > > | Highest Achievable TPR@0%FPR | 0.975 | 0.965    | 0.980      | 0.981 |
> > >
> > > **- line 458:** The use of SLMs is an integral part of our algorithmic design, setting it apart from other baseline watermarking algorithms. This distinction arises from the fact that their watermarking schemes and detectors are independent of the NTP, leaving no straightforward way to enhance their algorithms using this approach. We believe that designing watermarking schemes based on the NTP and leveraging the capabilities of SLMs represents a novel contribution to our work, and we look forward to future research building upon this foundation. In addition, in response to **- line 426:**, we present a comparison with smaller and different SLMs to demonstrate the effectiveness of our method.

---

> > > > ### Comment · Reviewer_sTiD · 2024-11-23
> > > >
> > > > Thanks for these new results.
> > > >
> > > > Again, TPR@FPR=10^-2 is not meaningful. Some recent papers now provide decent TPR@FPR=10^-6, like for instance "*WaterMax: breaking the LLM watermark detectability-robustness-quality trade-off*".
> > > >
> > > > Since you were still ignoring our concern about high FPR, I implemented your scheme and I can tell that it works nicely at high FPR (as you reported) but it just does not work at low FPR (below 10^-4). This reveals that your scheme is not practical.
> > > > Disclaimer: I am aware that my implementation is based on my understanding of your scheme in Section 5, which I hope is correct.
> > > >
> > > > I did not investigate further but I suppose that the inequality coupling $\eta$ with $T, \alpha,\lambda$ becomes un-tight at low $\alpha$. Then, $\eta$ becomes small, and the generator keeps on sampling the same useless $\tilde{\zeta}$.

---

> > > > ### Comment · Reviewer_sTiD · 2024-11-23
> > > >
> > > > >We find that when ... the FPR drops to 0
> > > >
> > > > No, the empirical FPR drops to 0, but since you measure it with only 500 samples, it means that the true FPR is between 0 and 10^-2 (roughly speaking).
> > > >
> > > > I observe here that your experimental setup is unable to operate at low FPR.
> > > >
> > > > By the way, I see that you have modified line 493. You are now testing on 8000 unwatermarked sentences. I believe that these are texts generated by the original LLM (without watermark). I advise you to consider real human writings like wikipedia which are more diverse than generated texts.

---

> > > > > ### Author Response · Authors · 2024-11-25
> > > > >
> > > > > **To new Q3**
> > > > >
> > > > > To clarify, we did not overlook your concern regarding the low FPR.  As we have shown in the new results on 0%FPR for 500 sentences, our scheme still achieves higher TPR than KGW-1, EXP-edit and Gumble-Max. In Table 3 in our manuscript, we calculate the empirical FPR on texts from **C4 dataset**, which is not generated by the unwatermarked LLM.
> > > > >
> > > > > In the following, we acknowledge the reviewer’s good suggestion on trying out the Wikipedia dataset.  Based on your suggestion, we compute the FPR on **100k sentences from the Wikipedia dataset**. The corresponding theoretical and empirical FPR is presented in the following table.
> > > > >
> > > > > | Theoretical FPR | 9e-3  | 2e-3  | 5e-4  | 9e-5  |
> > > > > |-------------|-------|-------|-------|-------|
> > > > > | Emperical FPR  | 1e-04 | 4e-05 | 2e-05 | 2e-05 |
> > > > >
> > > > > This new table shows that our theoretical upper bound effectively controls the empirical FPR even when it is $\leq 10^{-4}$. To further clarify, our theoretical result presents an upper bound on the true FPR (cf. Lemma 3), which we acknowledge is not tight. However, by suitably setting the token-level false alarm $\eta$, we can effectively control this upper bound, which consequently yields an even smaller true FPR on the sentence. Both the new result and Table 3 verify that our approach has good control.
> > > > >
> > > > > In summary, our scheme still works well in a low FPR regime from the evidence.

---

> > > > > > ### Comment · Reviewer_sTiD · 2024-11-25
> > > > > >
> > > > > > >We kindly request your clarification or result about ``it just does not work at low FPR (below 10^-4)”.
> > > > > >
> > > > > > I was not referring to Hypothesis $H_0$ (Table 3), but to hypothesis $H_1$. Your scheme is unique because the total FPR is a parameter not only of the watermark detection but also of the watermark embedding. When the FPR is small, my implementation keeps on sampling $\tilde{\zeta}$.
> > > > > >
> > > > > > My concern is not about whether the empirical FPR matched the theoretical PFR. My concern is about the TPR @ low FPR.

---

> > > > > > > ### Author Response · Authors · 2024-11-29
> > > > > > >
> > > > > > > > My concern is about the TPR @ low FPR.
> > > > > > >
> > > > > > > We conduct extra experiments on **100k sentences from the Wikipedia dataset**. The following table shows that our watermarking scheme achieves higher TPR even when FPR $\leq 10^{-4}$ compared to the baselines KGW-1 and Gumble-Max. We hope that this result resolves your concern.
> > > > > > >
> > > > > > > |            | TPR@1e-05FPR | TPR@1e-04FPR | TPR@1e-03FPR | TPR@1e-02FPR | TPR@1e-01FPR |
> > > > > > > |------------|--------------|--------------|--------------|--------------|--------------|
> > > > > > > | KGW-1      | 0.682        | 0.916        | 0.976        | 0.991        | 0.997        |
> > > > > > > | Gumble-Max | 0.515        | 0.844        | 0.955        | 0.984        | 0.994        |
> > > > > > > | Ours       | 0.882        | 0.951        | 0.992        | 0.997        | 0.999        |

---

> ### Author Response · Authors · 2024-11-21
> **To the rest of the questions**
>
> **W4. Minor comments**
> - line 107: We thank the reviewer for guiding us to the references. We have added the citations in "Other Related Literature". Merhav and Sabbag (2008) consider the text sequence $X_1^T$ is generated i.i.d. from a known distribution $P_X$ and the watermark embedder (i.e., watermarking scheme) is a function of $X_1^T$ and $\zeta_1^T$. They consider the cases where $\zeta_1^T$ is fixed or independently generated from a fixed distribution. By considering a given set of test statistics, they derive the asymptotically optimal detector and watermark embedder. Merhav and Ordentlich (2006) consider adopting causal codes for joint lossy source coding and watermark embedding. The encoder takes input as the previously generated symbols and watermarks and varies with time. The compressed watermark variable is uniformly distributed on $\{0,1\}^r$ for some constant $r$. They explore the tradeoffs among the distortion between the watermarked text and the original text, the text compression rate, and the watermark compression rate.
>
>   Both papers offer valuable insights that can inform future extensions of our work.
>
> - line 136: Since the unwatermarked-LLM-generated text is barely distinguishable from the human-generated one. In Section 2 above Figure 1, we state that we assume that the unwatermarked LLM distribution $Q$ is identical to that of humans. In the detection phase, the detector is model-agnostic, and $Q$ is unknown to the detector under both hypotheses. The unwatermarked LLM distribution $Q$ is only available in the watermark generation phase.
>
> - line 404: Thank you. We have corrected the typo.
> - Alg 2. line 3,4:  Thank you. We have corrected the notation.
> - line 809: We thank you for your meticulous reading. We have corrected the typo. Yes the TV is defined as $\mathsf{D}_\mathsf{TV}(\mu,\nu)\coloneqq\int \frac{1}{2}| \frac{d  \mu}{ d  \nu}-1|   d \nu$
> - line 831, line 885: Sorry for some typos in the proof. In line 831, this is part of the proof for Example 2, where we aim to show an example of suboptimal detector $\gamma(X_1^T,\zeta_1^T)= \mathbf{1}\lbrace f(X_1^T)=\zeta_1^T \rbrace$, for some surjective function $f:\mathcal{V}^T \to \mathcal{S}$ and $\mathcal{S}\subseteq \mathcal{Z}^T$. However, in line 885, this is part of the proof for Theorem 2 which presents the optimal type detectors and watermarking schemes. The optimal detector takes the form $\gamma(X_1^T,\zeta_1^T)= \mathbf{1} \lbrace X_1^T=g(\zeta_1^T) \rbrace$ for some surjective function $g: \mathcal{Z}^T \to \mathcal{S}$ and $\mathcal{S}\supset  \mathcal{V}^T$. $\mathcal{S}$ is used for different purposes in the two proofs.
> - line 1040: Right. The special case considers $\epsilon=0$.
> - line 1056: Thank you. We have corrected the typo.
> - line 1353: Thank you. Indeed, it is more consistent to use $P$ not $\tilde{Q}$. We have corrected the notation.
>
>
>
> **Q2. Efficient watermarking scheme**
> - line 399, 406, 411: Thank you for pointing out this confusion. We do not need to compute $h^{-1}$ in practice. Since the vocabulary is fixed, given a key and a $\zeta$, what we do is to search the vocabulary table to find the corresponding $x$ such that $h_{key}(x)=\zeta$.
>
>   To demonstrate the efficiency of our watermarking method, we conducted experiments measuring the average generation time for both un-watermarked and watermarked text. For these experiments, we employed LLaMA2-13B and the C4 dataset, setting the text length to 200 tokens in both scenarios. The results, presented in the following table, indicate that the difference in generation time between un-watermarked and watermarked text is less than $0.5$ seconds. This minimal difference confirms that our watermarking method has a negligible impact on generation speed, ensuring practical applicability.
>
>   | Language Model | Watermark      | Avg Generation Time/s |
>   |----------------|----------------|-----------------------|
>   | Llama2-13B     | Un-watermarked | 9.110s                |
>   | Llama2-13B     | Watermarked    | 9.386s                |
>
>
>
>
> **Q4. Technical detail**
>
> - line 777: Thanks for the question. We have added a detailed proof in the Appendix B:
>
> $\sum_{x_1^T:P^\*_{X_1^T}(x_1^T)\geq \alpha}\epsilon^\*(x_1^T)$
>
>   $= \sum_{x_1^T:P^\*_{X_1^T}(x_1^T)\geq \alpha, \epsilon^\*(x_1^T) \geq 0}\epsilon^*(x_1^T)$
>
>   $+\underbrace{\sum_{x_1^T:P^*_{X_1^T}(x_1^T)\geq \alpha, \epsilon^*(x_1^T) \leq 0}\epsilon^\*(x_1^T)}_{\leq 0}$
>
> $\leq \sum_{x_1^T:P^\*_{X_1^T}(x_1^T)\geq \alpha, \epsilon^*(x_1^T) \geq 0}\epsilon^\*(x_1^T)$
>
> $=\sum_{x_1^T:P^*_{X_1^T}(x_1^T)\geq \alpha, Q_{X_1^T}(x_1^T)\geq P^*_{X_1^T}(x_1^T) }\epsilon^\*(x_1^T)$
>
> $\leq \sum_{x_1^T:Q_{X_1^T}(x_1^T)\geq P^\*_{X_1^T}(x_1^T)}\epsilon^\*(x_1^T)\leq\epsilon$

---

> ### Author Response · Authors · 2024-11-25
>
> **To new Q1**
>
> We acknowledge that our proposed framework differs from the classical watermarking paradigm, which relies on fixed watermarks or a fixed watermark distribution. However, we view this as a generalized watermarking framework—a “relaxation,” as you mentioned. This approach, particularly well-suited for large language models (LLMs), allows the watermark distribution to adapt dynamically to the LLM and enables detection through innovative methods, such as leveraging a surrogate language model.
>
> From a theoretical perspective, we believe it is reasonable to assume that watermark variables can be recovered by the detector, as this assumption facilitates a focused exploration of the fundamental limits of watermarking. The primary challenge lies in the practical realization of such recovery, which we recognize as an important consideration but not a theoretical limitation.
>
> While we respect and understand the reviewers’ perspective, we hope to keep our broader definition of watermarking, as it aligns with the goals and nuances of our framework.

---

> > ### Comment · Reviewer_sTiD · 2024-11-25
> >
> > >We acknowledge that our proposed framework differs from the classical watermarking paradigm, which relies on fixed watermarks or a fixed watermark distribution.
> >
> > Ok. I will raise my score. How will it be implemented in the revised paper?

---

> ### Author Response · Authors · 2024-11-25
>
> **To new Q2**
>
> We thank the reviewer for trying out our scheme. We kindly request your clarification or result about ``it just does not work at low FPR (below 10^-4)”.
>
> We also thank you for referring us to the WaterMax paper, which we will add into the related works and consider comparisons with it in the future.
>
> Our theoretical result presents an upper bound on the true FPR (cf. Lemma 3), which we acknowledge is not tight. However, by suitably setting the token-level false alarm $\eta$, we can effectively control this upper bound, which should consequently yield an even smaller true FPR on the sentence.

---

> ### Author Response · Authors · 2024-11-29
> **Thank you for the enhanced rating and your valuable suggestions**
>
> > Ok. I will raise my score. How will it be implemented in the revised paper?
>
> We sincerely appreciate the reviewer’s insightful suggestions and the upgraded score. We will add the following clarifications in the revised manuscript:
>
> - After line 60: “...jointly optimize both the watermarking scheme and the detector, achieving universal optimality. **In contrast to the classical watermarking paradigm with fixed watermark messages or distributions, this study explores a broader definition of watermarking where watermarks can be dynamically adapted to the LLM and its output texts.**...”
> - In the first point of our contribution summary: “...More importantly, **within our framework, we characterize the complete class of universally optimal detectors and watermarking schemes, meaning no other type can achieve the same performance.**”

---

> > ### Comment · Reviewer_sTiD · 2024-11-29
> >
> > On the one hand, you have shown (at last!) some TPR@low FPR. I am very eager to run your code and see where it differentiates
> > from my implementation. I will increase my score.
> >
> > On the other hand, you are refusing to rewrite the first technical section to present it as the optimization of a new functionality *which is not watermarking, but a relaxation of watermarking*, yet whose results will be very useful as it gives an upper bound of the performance of any watermarking scheme. I will lower my score.
> >
> > >After line 60: “...jointly optimize both the watermarking scheme and the detector, achieving universal optimality.
> >
> > > “...More importantly, within our framework, we characterize the complete class of universally optimal detectors and watermarking schemes,
> >
> > Conclusion: my score remains unchanged. And this is my last move. This discussion has lasted too long. I am very sorry that we could not find an agreement.

---

### Official Review · Reviewer_3q3R · 2024-10-30

**Soundness:** 2
**Presentation:** 2
**Contribution:** 2
**Rating:** 3
**Confidence:** 5

**Summary:**

In this paper, the authors introduce a watermarking and detection framework aimed at achieving universal optimality by minimizing Type-II errors while controlling for Type-I errors and distortion. They propose a token-level watermarking scheme that is model-agnostic, utilizing a surrogate model and the Gumbel-Max trick. Their experiments on Llama-13B and Mistral-8×7B demonstrate the scheme's efficacy and robustness.

**Strengths:**

This paper provides detailed theoretical analyses of watermarking schemes.

**Weaknesses:**

1. This method is a modification based on Gumbel-Max, and employs a surrogate model during detection. However, its effectiveness may heavily depend on the similarity between the surrogate model and the original watermarked model. Notably, the authors only report results for Llama2-7B as a surrogate for Llama2-13B, and Mistral-7B for Mistral-8×7B, which are very similar pairs. This limits the algorithm’s applicability, as suitable surrogate models may not always be available.
2. During detection, the detector has no access to the user prompt $u$, which affects the reconstruction of $P_{\zeta_t | x_1^{t-1},u}(\zeta)$ and may degrade the detection performance.
3. I assume the authors apply the detection method for KGW-1 using the algorithm in lines 313–317, which is suboptimal compared to their original z-score-based detection algorithm.
4. The experiments do not report any metrics on generated text quality, and it is easy to trade text quality for improved detection ability.
5. The 1% FPR is generally too high for real-world applications. Could you provide results for TPR at much smaller FPR, such as 0.1%, 0.01% or even 0.001%?

**Questions:**

1. How do you determine the false positive rates? Are they calculated through a theoretical bound or by testing on unwatermarked texts?
2. Could you provide baseline results using a detector similar to the Gumbel-Max watermark detector in lines 318–320, instead of Eq. (4), to confirm the optimality of your approach?

---

> ### Author Response · Authors · 2024-11-21
> **To W1**
>
> We thank the reviewer for thoroughly reading our manuscript and providing constructive feedback. We have carefully considered the reviewer’s comments and provided the following responses.
>
> **W1. This method is a modification based on Gumbel-Max. The choice of the surrogate model may affect the effectiveness of the proposed watermarking method, which may limit the algorithm’s applicability.**
>
> 1) Our watermarking method is fundamentally different from the Gumbel-Max watermarking approach. What we use is the Gumbel-Max trick, a method to draw a sample from a categorical distribution. As detailed in Algorithms 1 and 2, one important component of our optimal watermarking method is the construction of sampling distribution $P_{\zeta_t|x_1^{t-1}}$ of auxiliary $\zeta_t$ based on the original Next Token Prediction (NTP) distribution. After constructing $P_{\zeta_t|x_1^{t-1}}$, we sample  $\zeta$ from it​ to watermark the next token. However, directly sending $\zeta$ from the generator to the detector is impractical due to its dependence on the NTP distribution. To address this challenge, we utilize a surrogate model and employ the Gumbel-Max trick to sample $\zeta$ from the approximated $P_{\zeta_t|x_1^{t-1}}$. Therefore, the Gumbel-Max trick is a way to help us sample the $\zeta_t$ during the detection process. In contrast, the Gumbel-Max watermark method directly samples the next token from the original NTP distribution using a Gumbel variable drawn from a uniform distribution. Furthermore, compared to the Gumbel-Max watermarking method, our approach is adaptive, leveraging the information from the original NTP distribution. We want to emphasize that the design of such an adaptive watermarking scheme is driven by our theoretical analysis of universally optimal watermarking, which is different from the existing heuristic watermarking designs.
>
>
> 2) The selection of the surrogate model is primarily based on its vocabulary or tokenizer rather than the specific language model within the same family. This choice is critical because, during detection, the text must be tokenized exactly using the same tokenizer as the watermarking model to ensure accurate token recovery. As a result, any language model that employs the same tokenizer can function effectively as the surrogate model.
>
>     To validate our approach, we add a **new experiment** by applying our watermarking algorithm to GPT-J-6B (a model with 6 billion parameters) and using GPT-2 Large (774 million parameters) as the surrogate model. Despite differences in developers, training data, architecture, and training methods, these two models share the same tokenizer, making them compatible for this task. We conduct experiments using the C4 dataset, and the results are presented in the tables below. The results demonstrate the effectiveness of our proposed watermarking method with or without attack even when using a surrogate model from a different family than the watermarking language model. Notably, the surrogate model, despite having fewer parameters and lower overall capability compared to the watermarking language model, does not compromise the watermarking performance.
>
>     Moreover, we believe that training a small surrogate model with fewer parameters is neither a particularly challenging nor a resource-intensive task for LLM developers. Techniques such as retraining or knowledge distillation can be employed to efficiently create such a model, making it a practical solution.
>
> **Without attack:**
>
> | Language Model | surrogate model | ROC-AUC | TPR@1% FPR | TPR@10% FPR |
> |----------------|-----------------|---------|------------|-------------|
> | Llama2-13B     | Llama2-7B       | 0.999   | 0.998      | 1.000       |
> | Mistral-8 × 7B | Mistral-7B      | 0.999   | 0.998      | 1.000       |
> | GPT-J-6B       | GPT-2 large     | 0.997   | 0.990      | 0.997       |
>
> **With token replacement attack:**
>
> | Language Model | surrogate model | ROC-AUC | TPR@1% FPR | TPR@10% FPR |
> |----------------|-----------------|---------|------------|-------------|
> | Llama1-13B     | Llama2-7B       | 0.989   | 0.860      | 0.976       |
> | Mistral-8 × 7B | Mistral-7B      | 0.990   | 0.881      | 0.966       |
> | GPT-J-6B       | GPT-2 large     | 0.987   | 0.892      | 0.962       |

---

> > ### Author Response · Authors · 2024-11-21
> > **To W2, W3, W4**
> >
> > **W2. During detection, the detector has no access to the user prompt $u$, which affects the reconstruction of $P_{\zeta_t|x_1^{t-1},u}$ and may degrade the detection performance.**
> >
> > To evaluate the impact of prompts on watermark detection performance, we conducted experiments to compare detection performance with and without prompts attached to the watermarked text during the detection process. Moreover, we conducted experiments using both the original watermarking LLM and prompts during the detection process to evaluate detection performance under conditions where we could precisely reconstruct $P_{\zeta_t|x_1^{t-1},u}$.
> >
> > These experiments were performed using LLama2-13B on the C4 dataset, both with and without adversarial attacks. The sampling temperature is set to $0.8$. The results are presented in the following tables. Notably, even when prompts are absent and the surrogate model cannot perfectly reconstruct the same $P_{\zeta_t|x_1^{t-1},u}$​ as in the generation process, detection performance remains unaffected. This demonstrates the robustness of our watermarking method, regardless of whether a prompt is included during the detection phase. As discussed in our manuscript, being prompt- and model-agnostic is a highly practical and valuable property of a watermarking method. The observed detection performance loss of approximately $0.1$% to achieve this agnostic capability is minimal and represents a worthwhile trade-off for improved practicality.
> >
> > *Without attack*
> > | Language Model | Surrogate model | Setting        | ROC-AUC | TPR@1% FPR | TPR@10% FPR |
> > |----------------|-----------------|----------------|---------|------------|-------------|
> > | Llama2-13B     | Llama2-7B       | Without Prompt | 0.997   | 0.983      | 0.995       |
> > | Llama2-13B     | Llama2-7B       | With Prompt    | 0.998   | 0.989      | 0.996       |
> > | Llama2-13B     | Llama2-13B      | With Prompt    | 0.998   | 0.990      | 0.996       |
> >
> >
> > *With token replacement attack*
> > | Language Model | Surrogate model | Setting        | ROC-AUC | TPR@1% FPR | TPR@10% FPR |
> > |----------------|-----------------|----------------|---------|------------|-------------|
> > | Llama2-13B     | Llama2-7B       | Without Prompt | 0.977   |  0.818     | 0.953       |
> > | Llama2-13B     | Llama2-7B       | With Prompt    | 0.979   |  0.816     | 0.960       |
> > | Llama2-13B     | Llama2-13B      | With Prompt    | 0.980   |  0.817     | 0.962       |
> >
> >
> > ---
> > ---
> >
> > **W3. I assume the authors apply the detection method for KGW-1 using the algorithm in lines 313–317, which is suboptimal compared to their original z-score-based detection algorithm.**
> >
> > Thank you for pointing out this potential source of confusion. In lines 313-317 (Example 3), we wrote a simplified version of the z-score-based detector.  Nonetheless, in our experiments with the KGW-1 watermark, we employed the z-score-based detection algorithm, consistent with the original paper. To address this issue and avoid further misunderstanding,  in Example 3 of the revised manuscript, we rewrite the detector for KGW-1 to be exactly the same as the z-score-based detector. This ensures alignment with the actual approach used in our experiments.
> >
> > *The revised version:*
> >
> >   Green-Red List watermark detector \citep{kirchenbauer2023watermark}: $\gamma(X_1^T,\zeta_1^T)=\mathbb{1}\lbrace \frac{2}{\sqrt{T}} (\sum_{t=1}^T \mathbb{1}\lbrace \zeta_t(X_t)=1\rbrace -\frac{T}{2})\rbrace$ where $\lambda>0$, $\zeta_t=(\zeta_t(x))_{x\in\mathcal{V}}$ is uniformly sampled from $  \lbrace \zeta\in\lbrace 0,1\rbrace^{|\mathcal{V}|}: \|\zeta\|_1=\rho|\mathcal{V}|\rbrace  $,
> >
> >   with the seed    $  h(X_{t-1},key)  $,   $  \rho\in(0,1)  $ is the green list proportion.
> >
> >
> > ---
> > ---
> > **W4. Report the metrics on watermarked text quality.**
> >
> > To evaluate the quality of watermarked text generated by our watermarking methods, we computed the perplexity of the watermarked text using GPT-3. The results are presented in the table below. Compared to unwatermarked text and text generated using other watermarking methods, our method achieves the lowest perplexity. This demonstrates that our watermarking approach has minimal impact on text quality, preserving its naturalness and coherence. Our watermarking scheme employs an NTP distribution-adaptive approach, which differs significantly from traditional uniform sampling methods. This design minimizes the impact on text quality, ensuring a less intrusive watermarking process.
> >
> > | Watermarking Method | Un-watermarked | KGW-1  | EXP-edit | Gumble-Max | Ours  |
> > |---------------------|----------------|--------|----------|------------|-------|
> > | Average Perplexity  | 8.846          | 14.327 | 12.186   | 11.732     | 6.495 |

---

> > > ### Author Response · Authors · 2024-11-21
> > > **To the rest of the questions**
> > >
> > > **W5. Provide results for TPR at much smaller FPR, such as 0.1%, 0.01% or even 0.001%.**
> > >
> > > Thank you for your good suggestion. In our experiments, we generated 500 unwatermarked texts and 500 watermarked texts separately. As a result, we did not observe an extremely low False Positive Rate (FPR) under this setting. However, empirically, we change the threshold $\lambda$ to investigate the TPR and corresponding FPR. We find that when $\lambda>=0.246$, the FPR drops to $0$, while the corresponding highest achievable TPR is $0.981$. We compare the highest achievable TPR@0%FPR across different watermarking methods, and the results are presented in the following table. Our method achieves comparable performance with other baselines.
> > >
> > > | Watermarking Method          | KGW-1 | EXP-edit | Gumble-Max | Ours  |
> > > |------------------------------|-------|----------|------------|-------|
> > > | Highest Achievable TPR@0%FPR | 0.975 | 0.965    | 0.980      | 0.981 |
> > >
> > > ---
> > > ---
> > > **Q1. How do you determine the false positive rates? Are they calculated through a theoretical bound or by testing on unwatermarked texts?**
> > >
> > > We calculate the False Positive Rate (FPR) using both theoretical and empirical approaches.
> > >
> > > The sequence-level false alarm $\alpha$ is dependent on the chosen token-level false alarm $\eta$, the generated sequence length $T$ and the detection threshold $\lambda$. The relationship is stated in Lemma 3: if $\eta\in(0, \min\{ 1, (\alpha/\binom{T}{\lceil T\lambda \rceil})^{\frac{1}{\lceil T\lambda \rceil}}\}]$, the worst-case false alarm for a length-$T$ sequence is upper bounded by $\alpha$. In general, given $T$ and $\lambda$, a smaller $\eta$ results in a smaler $\alpha$; given $T$ and $\eta$, a higher $\lambda$ results in a smaller $\alpha$.
> > >
> > > For the results presented in Table 1 and Table 2 of our manuscript, we report True Positive Rate (TPR) and FPR using ROC-AUC score function based on the generated 500 watermarked texts and 500 unwatermarked texts.  The ROC-AUC score function varies the detection threshold $\lambda$ to achieve different values of FPR, ranging from 0 to 1, and reports the corresponding TPR. For the results presented in Table 3 of our manuscript, with the given sequence length $T$ and token-level FPR $\eta$, we set several theoretical FPRs $\alpha$ and obtain $\lambda$ by solving $\alpha=\binom{T}{\lceil T \lambda\rceil} \eta^{\lceil T \lambda\rceil}$. Then, to evaluate the consistency between theoretical and empirical results, we use the detection threshold $\lambda$ derived from the theoretical FPR to compute the empirical FPR based on 8000 unwatermarked sentences.
> > >
> > > ---
> > > ---
> > >
> > > **Q2. Provide baseline results using a detector similar to the Gumbel-Max watermark detector in lines 318–320, instead of Eq. (4), to confirm the optimality of the proposed approach.**
> > >
> > >
> > > Thanks for pointing this out and we are sorry for any confusion. We would like to address this point from the following aspects:
> > > The Gumbel-max watermark detector takes input auxiliary $\zeta_t$ that are independently and uniformly sampled from $[0,1]$, and can only work well for the Gumbel-max watermarking scheme. It is not compatible with our watermarking scheme, where the auxiliary $\zeta_t$ is either not from $[0,1]$ or not uniformly sampled.
> > >
> > > We would like to clarify our problem setup and theoretical contributions. Within our proposed generic watermark generation and detection framework, we aim to jointly optimize the watermarking scheme, denoted by the joint distribution $P_{X_1^T, \zeta_1^T}$, and the model-agnostic detector $\gamma$ that minimizes the Type-II error (false negative) under the $\alpha$-constrained worst-case Type-I error (false positive).
> > >
> > > We formulate the optimization problem in (OPT-O). The optimal objective value is thus the **universally minimum** Type-II error, as the Type-I error is under control for all possible text distributions (Theorem 1). The optimizer identifies the **optimal class of detector-watermarking scheme pairs**. Additionally, we prove that detectors within this optimal class, denoted as $\Gamma^*$, are **universally optimal**. Specifically, for any detector $\gamma \notin \Gamma^*$, there always exists a text distribution $Q$ and a distortion level $\epsilon$ such that no watermarking scheme can achieve the universally minimum Type-II error.  It suggests that to guarantee the construction of an optimal watermarking scheme for any arbitrary LLM, the detector must be selected from the set $\Gamma^*$.
> > >
> > > Therefore, our designed practical watermarking scheme corresponds with the detector in our Algorithms 1 and 2.

---

> ### Author Response · Authors · 2024-11-25
> **Follow-up discussion**
>
> Dear reviewer 3q3R,
>
> We sincerely appreciate your time and effort in reviewing our submission and providing valuable suggestions. While we hope to have addressed your concerns adequately, we understand there may still be areas requiring further clarification or discussion. We are fully prepared to address your outstanding issues. Should our responses have successfully addressed all your questions, we would be deeply grateful if you could consider enhancing the score to further support our submission. Thank you very much for your thoughtful review.
>
> Best Regards,
>
> Paper12990 Authors

---

> > ### Comment · Reviewer_3q3R · 2024-11-25
> >
> > Thank you for your detailed response. However, I still have a few questions:
> >
> > 1. Do you have the prompt information for the results reported in Table 1 and Table 2?
> >
> > 2. I greatly doubt the correctness of your results on the average perplexity. As EXP-edit and Gumbel-Max are distortion-free watermarking schemes, their performance should closely match the un-watermarked baseline, as confirmed by many prior studies. Moreover, it is unusual that your results greatly outperform the un-watermarked model. Could you clarify how these results were obtained?
> >
> > 3. Could you provide additional experimental results on other tasks, such as machine translation or question answering, and report metrics like BLEU and ROUGE-L to validate this phenomenon?
> >
> > 4. For Table 1 and Table 2, could you report the TPR@FPR under the theoretical FPR? Then you can show the results with FPR=0.1%, 0.01% and 0.001%.

---

> ### Author Response · Authors · 2024-11-29
>
> We thank the reviewer for providing additional constructive feedback. We have carefully considered the reviewer’s comments and provided the following responses. We are trying our best to solve the reviewer's problem, and we would greatly appreciate it if the reviewer could kindly reconsider the score to further support our work.
>
>
> **Q1. Do you have the prompt information for the results reported in Table 1 and Table 2?**
>
> During the watermark generation process, we use the first two sentences of each text in the dataset as a prompt. It is also stated in the “Experiment settings”.
>
> During the watermark detection process, we don’t have access to the prompts.
>
>
> **Q2. Clarify how to get the perplexity results.**
>
> For the experiment setting, we conduct experiments on Llama2-13B. We set the temperature to be 0.8 for all methods. The perplexity is computed using GPT-3. Moreover, some prior works have shown that both Gumbel-Max and EXP-Edit exhibit higher perplexity compared to unwatermarked text, e.g. as shown in Table 4 of [1]. We kindly request the reviewer to share the references mentioned that present differing results.
>
> [1] Leyi Pan, Aiwei Liu, Zhiwei He, Zitian Gao, Xuandong Zhao, Yijian Lu, Binglin Zhou, Shuliang Liu, Xuming Hu, Lijie Wen, et al. Markllm: An open-source toolkit for llm watermarking. arXiv preprint arXiv:2405.10051, 2024a.
>
>
> **Q3. BLEU score on machine translation task.**
>
> We conduct new experiments on the machine translation task. We use the WMT19 dataset and MBART Model. It can be observed that our scheme achieves a higher BLEU score than the baseline schemes and is close to the unwatermarked BLEU score.
>
> | Methods    | Un-watermarked | KGW-1 | EXP-Edit | Gumble-Max | Ours  |
> |------------|----------------|-------|----------|------------|-------|
> | BLEU Score | 0.219          | 0.158 | 0.203    | 0.210      | 0.214 |
>
>
>
> **Q4. Show the results with FPR=0.1%, 0.01% and 0.001%**
>
> We conduct new experiments on **100k sentences from the Wikipedia dataset**. The following table shows that our watermarking scheme achieves higher TPR even when FPR $= 10^{-5}$.
>
> |            | TPR@1e-05FPR | TPR@1e-04FPR | TPR@1e-03FPR | TPR@1e-02FPR | TPR@1e-01FPR |
> |------------|--------------|--------------|--------------|--------------|--------------|
> | KGW-1      | 0.682        | 0.916        | 0.976        | 0.991        | 0.997        |
> | Gumble-Max | 0.515        | 0.844        | 0.955        | 0.984        | 0.994        |
> | Ours       | 0.882        | 0.951        | 0.992        | 0.997        | 0.999        |
>
>
> Furthermore, we also compare the empirical FPR with the corresponding theoretical, which is presented in the following table.
>
> | Theoretical FPR | 9e-3  | 2e-3  | 5e-4  | 9e-5  |
> |-------------|-------|-------|-------|-------|
> | Emperical FPR  | 1e-04 | 4e-05 | 2e-05 | 2e-05 |
>
> This table above shows that our theoretical upper bound effectively controls the empirical FPR even when it is $<=10^{-4}$.

---

> ### Author Response · Authors · 2024-12-02
> **Follow-up discussion**
>
> Dear reviewer 3q3R,
>
> We sincerely appreciate your time and constructive feedback, which helps us further improve the quality of our submission. We have made every effort to answer your insightful questions by modifying the manuscript and writing rebuttals. Considering today (Dec 2) is the deadline for the author's response, we hope our responses have addressed the reviewer's concerns, but if not, we are available to address any outstanding issues. In case we have addressed all your questions, we truly appreciate your feedback and reconsider your score for further support.
>
> Best Regards,
>
> Paper12990 Authors

---

### Official Review · Reviewer_BdQu · 2024-11-03

**Soundness:** 3
**Presentation:** 2
**Contribution:** 2
**Rating:** 3
**Confidence:** 4

**Summary:**

This paper optimizes watermarking and detection to improve performance while managing errors and distortion. It establishes key trade-offs and identifies optimal schemes. An efficient watermarking algorithm is introduced, demonstrating effectiveness on several models and datasets. The paper also explores enhancing robustness against adversarial attacks in future systems.

**Strengths:**

The conclusions in the paper are well explained and proved with sufficient math notations. Furthermore, the paper introduces a optimized algorithm based on previous conclusions.

**Weaknesses:**

Although the proposed watermarking scheme can achieve the best performance among baselines, the numerical difference is not so significant. That is to say, for the detection accuracy evaluation, the performance of baseline method is already good enough, for both high-entropy and low-entropy texts. Then with textual attacks, although the difference is more obvious, there lacks some up-to-date robustness-enhanced algorithms such as SIR [1].
In sum, the significance of the proposed method is not sufficiently demonstrated in the paper.

1. A Semantic Invariant Robust Watermark for Large Language Models

**Questions:**

As explained in Weaknesses.

---

> ### Author Response · Authors · 2024-11-21
>
> We thank the reviewer for thoroughly reading our manuscript and providing constructive feedback. We have carefully considered the reviewer’s comments and provided the following responses.
>
> **Q1. The significance of the proposed method is not sufficiently demonstrated in the paper.**
>
> 1) **Theoretical Contribution**
> First of all, to better illustrate our proposed generic framework of watermarking LLM and watermark detection, which can be specialized to any existing in-process watermarking schemes, we improve our Figure 1 as follows: https://anonymous.4open.science/r/ICLR_rebuttal-5581/watermark_framework.pdf.
> This shows a more detailed and readable process of watermark generation and detection. Specifically, the LLM watermarking process can be exactly characterized by the joint distribution $P_{X_1^T, \zeta_1^T}$. Second, given this watermarking process, we observe that the detection problem boils down to the independence testing problem between the received text $X_1^T$ and the recovered auxiliary sequence $\zeta_1^T$ (i.e., watermark information). Within the hypothesis testing framework, we aim to jointly optimize the watermarking scheme, denoted by the joint distribution $P_{X_1^T, \zeta_1^T}$, and the model-agnostic detector $\gamma$ that minimizes the Type-II error (false negative) under the $\alpha$-constrained worst-case Type-I error (false positive).
> We formulate the optimization problem in (OPT-O). The optimal objective value is thus the **universally minimum** Type-II error, as the Type-I error is under control for all possible text distributions (Theorem 1). The optimizer identifies the **optimal class of detector-watermarking scheme pairs**. Additionally, we prove that detectors within this optimal class, denoted as $\Gamma^*$, are **universally optimal**. Specifically, for any detector $\gamma \notin \Gamma^*$, there always exists a text distribution $Q$ and a distortion level $\epsilon$ such that no watermarking scheme can achieve the universally minimum Type-II error.  It suggests that to guarantee the construction of an optimal watermarking scheme for any arbitrary LLM, the detector must be selected from the set $\Gamma^*$.
>
> 2) **Practical implications:**  The main contribution of the theoretically optimal results is to provide valuable guidelines for designing near-optimal and implementable algorithms. Note that this is a more systematic approach compared to existing heuristic designs. In Theorem 2, we prove that the optimal watermarking scheme $P_{X_1^T, \zeta_1^T}^*$ should adapt to the LLM $Q$. This result explains why other approaches, such as green/red list watermarking and Gumbel-Max watermarking, which sample $\zeta_t$ from a fixed distribution, are suboptimal.  Building on this theoretical result, we design algorithms that address two practical concerns:  adapting to variable-length text generation and recovering the auxiliary sequences at the model-agnostic detector side. In Section 4, we propose a token-level optimal detector and watermarking scheme based on our sequence-level optimal result. This approach effectively adapts to variable-length text generation while maintaining strong performance guarantees. In Section 5, we propose decomposing the sampling procedure of the auxiliary sequence $\zeta_t$ using the Gumbel-max trick, enabling the Gumbel noise (generated from fixed distribution) to be shared with the detector via a secret key. For watermark detection, a surrogate language model (SLM) is used to approximate the NTP distribution $Q$, and combined with the Gumbel-max trick to extract $\zeta_1^T$ from the text (illustrated in https://anonymous.4open.science/r/ICLR_rebuttal-5581/watermark_practice.pdf).
> Guided by the theoretical framework, experimental results show that our algorithm achieves excellent detection performance and robustness, even with the approximated auxiliary sequence $\zeta_1^T$.
>
> Therefore, we believe that the proposed theoretical framework and the derived optimality are useful and insightful in practice.
>
> 3) **Extension to robust watermarking against broad attacks:** In Section 6, we show that this framework can be further generalized to the scenarios with semantic-invariant attacks. We prove the theoretical optimal and robust design, which also provides valuable insights into designing practical and robust semantic-based watermarking algorithms.

---

> ### Author Response · Authors · 2024-11-21
>
> **Q2. Compare the proposed method with robustness-enhanced algorithms such as SIR.**
>
> We conducted experiments using the SIR watermark and compared its detection performance and perplexity with our watermarking method, both with and without attack. The results of these comparisons are presented in the following tables. For detection performance without attack, our watermarking method is comparable to SIR. Under token replacement attacks, SIR demonstrates better robustness due to its design as a semantic-invariant watermark. Despite this advantage, SIR introduces biases in the next token prediction distributions, resulting in a relatively high perplexity. In contrast, our watermarking method is distortion-free, achieving significantly lower perplexity, which minimizes its impact on the generated text quality of the LLM.
>
> **Without attack**
>
> | Language Model | Methods | ROC-AUC | TPR@1% FPR | TPR@10% FPR | Perplexity |
> |----------------|---------|---------|------------|-------------|------------|
> | Llama2-13B     | SIR     | 0.997   | 0.987      | 0.990       | 14.537     |
> | Llama2-13B     | Ours    | 0.997   | 0.983      | 0.995       | 6.495      |
>
> **With token replacement attack**
>
> | Language Model | Methods | ROC-AUC | TPR@1% FPR | TPR@10% FPR |
> |----------------|---------|---------|------------|-------------|
> | Llama2-13B     | SIR     | 0.989   | 0.957      | 0.982       |
> | Llama2-13B     | Ours    | 0.977   | 0.818      | 0.953       |

---

> ### Author Response · Authors · 2024-11-25
> **Follow-up discussion**
>
> Dear reviewer BdQu,
>
> We sincerely appreciate your time and effort in reviewing our submission and providing valuable suggestions. While we hope to have addressed your concerns adequately, we understand there may still be areas requiring further clarification or discussion. We are fully prepared to address your outstanding issues. Should our responses have successfully addressed all your questions, we would be deeply grateful if you could consider enhancing the score to further support our submission. Thank you very much for your thoughtful review.
>
> Best Regards,
>
> Paper12990 Authors

---

> ### Author Response · Authors · 2024-12-02
>
> Dear reviewer BdQu,
>
> We sincerely appreciate your time and constructive feedback, which helps us further improve the quality of our submission. We have made every effort to answer your insightful questions by modifying the manuscript and writing rebuttals. Considering today (Dec 2) is the deadline for the author's response, we hope our responses have addressed the reviewer's concerns, but if not, we are available to address any outstanding issues. In case we have addressed all your questions, we truly appreciate your feedback and reconsider your score for further support.
>
> Best Regards,
>
> Paper12990 Authors

---

### Official Review · Reviewer_mxQq · 2024-11-04

**Soundness:** 3
**Presentation:** 1
**Contribution:** 2
**Rating:** 6
**Confidence:** 4

**Summary:**

The paper considers the question: For a given amount of allowable distortion and a given false positive rate, how can we design an LLM watermarking scheme to have the lowest false negative rate?
For a certain class of watermarking schemes (which includes all existing schemes), they derive an expression for this optimal false negative rate in terms of the LLM distribution.
For the same class of schemes, they show that all optimal detectors have a simple form.
This form is not taken by most existing watermarking schemes, suggesting a possibility for improvement.

They experimentally demonstrate their ideas, showing that they are competitive with the schemes they compared against.

**Strengths:**

It is useful to try to understand the optimal trade-offs in text watermarking.
The typical approach is to simply suggest a scheme that addresses some perceived issue, and this work stands out in that it takes a more principled approach to the problem.

I appreciated the abstract treatment of semantic watermarks, which could be useful in the future.

**Weaknesses:**

It is very difficult to ascertain the extent of your optimality results. They are buried in equations and technical writing.
It would be very helpful if you described formally, but without any equations, what precisely you are claiming about optimality.
As I understand it, you prove that for watermarking schemes that fall into the formulation you describe in "Watermarking LLMs" and for fixed-length generations with no errors, you present a scheme that achieves the lowest-possible false negative rate for any given distortion level, false negative rate, and LLM generation distribution (which must be known ahead of time).
As it is, though, one has to parse a bunch of symbols and a complicated theorem (Theorem 2, which states additional results as well) to understand this.
This problem of technicality without explanation pervades the paper.

You proved the optimality of a certain form of detectors for the case of fixed-length generations, but you didn't get any optimality results for variable-length generations as far as I can tell. This is of course the practically-relevant case, so it would be better if you could say something about this case.

**Questions:**

- It appears that the optimal watermarking scheme you construct requires knowledge of the LLM distribution. Is this correct? If so, can the parameters of the scheme be computed efficiently (practically, or in polynomial time)?
- Can you say anything about the optimality of your detector for generations that are not of a fixed length?
- Can you describe this set of optimal detectors \Gamma^* in simpler terms? It appears that you're saying that, for any fixed watermarked text generation algorithm of the form you consider, the optimal detector simply tests whether the entire generation is exactly equal to some string computed as a function of \zeta. However, this is clearly false: If \zeta is just a collection of red/green lists for each token position, then the detector should accept many possible texts. I think that what you're trying to say is that there exists a watermarking algorithm such that this detector is optimal, but this is not what appears to be stated in Theorem 2.
- Your description of the Kuditipudi et al. watermarking scheme appears to be completely unrelated to the scheme presented in their paper. Their scheme is not hash-based at all.

---

> ### Author Response · Authors · 2024-11-21
>
> We thank the reviewer for thoroughly reading our manuscript and providing constructive feedback. We have carefully considered the reviewer’s comments and provided the following responses.
>
> **Q1.  Described formally, but without any equations, what precisely you are claiming about optimality.**
>
> First of all, to better illustrate our proposed generic framework of watermarking LLM and watermark detection, which can be specialized to any existing in-process watermarking schemes, we improve our Figure 1 as follows: https://anonymous.4open.science/r/ICLR_rebuttal-5581/watermark_framework.pdf.
> This shows a more detailed and readable process of watermark generation and detection. Specifically, the LLM watermarking process can be exactly characterized by the joint distribution $P_{X_1^T, \zeta_1^T}$.
>
> Second, given this watermarking process, we observe that the detection problem boils down to the independence testing problem between the received text $X_1^T$ and the recovered auxiliary sequence $\zeta_1^T$ (i.e., watermark information). Within the hypothesis testing framework, we aim to jointly optimize the watermarking scheme, denoted by the joint distribution $P_{X_1^T, \zeta_1^T}$, and the model-agnostic detector $\gamma$ that minimizes the Type-II error (false negative) under the $\alpha$-constrained worst-case Type-I error (false positive).
> We formulate the optimization problem in (OPT-O). The optimal objective value is thus the **universally minimum** Type-II error, as the Type-I error is under control for all possible text distributions (Theorem 1). The optimizer identifies the **optimal class of detector-watermarking scheme pairs**. Additionally, we prove that detectors within this optimal class, denoted as $\Gamma^*$, are **universally optimal**. Specifically, for any detector $\gamma \notin \Gamma^*$, there always exists a text distribution $Q$ and a distortion level $\epsilon$ such that no watermarking scheme can achieve the universally minimum Type-II error.  It suggests that to guarantee the construction of an optimal watermarking scheme for any arbitrary LLM, the detector must be selected from the set $\Gamma^*$.
>  For clarity, we restate Theorem 2 as follows:
>
> **Optimal type of detectors and watermarking schemes.** Since we have established the universally minimum Type-II error, a natural question arises: what is the \emph{optimal type of detectors and watermarking schemes} that achieve this universal minimum (for all $Q_{X_1^T}$ and $\epsilon$)? Let  $\Pi^*(Q_{X_1^T},\alpha,\epsilon)$ denote the set of all solutions $(\gamma^*,P_{X_1^T,\zeta_1^T}^*)$ that achieve  $\beta_1^*(Q_{X_1^T},\alpha, \epsilon)$.
>
>
> **Theorem 2 ((Informal Statement) Optimal type of detectors and watermarking schemes)**
> The optimal type of detector is given by
> $$\Gamma^*\coloneqq \lbrace \gamma\mid \gamma(X_1^T,\zeta_1^T)= \mathbb{1}\lbrace X_1^T=g(\zeta_1^T)\rbrace, \text{ for some surjective } g:\mathcal{Z}^T\to \mathcal{S} \supset \mathcal{V}^T \rbrace.$$
>  For any  $\gamma^* \in \Gamma^*$ and any $(Q_{X_1^T},\epsilon)$, the corresponding optimal $\epsilon$-distorted watermarking scheme $P_{X_1^T,\zeta_1^T}^*$ is provided  in Appendix D, i.e., $(\gamma^*,P_{X_1^T,\zeta_1^T}^*)\in \Pi^*(Q_{X_1^T},\alpha,\epsilon)$.
>
> **Corollary 3 (Universal optimality of detectors $\Gamma^\*$)**
>     For any $\gamma \notin \Gamma^\*$, there exists $(\tilde{Q}_{X_1 ^T}$, $\tilde{\epsilon})$ such that no $\tilde{\epsilon}$ distorted watermarking scheme $P _{X_1^T, \zeta_1^T}$ satisfies $(\gamma$, $P _{X_1^T,\zeta_1^T})$ $\in$ $\Pi^*(\tilde{Q} _{X_1^T},\alpha,\tilde{\epsilon})$.
>
> Theorem 2 and Corollary 3 suggest that to guarantee the construction of an optimal watermarking scheme for any arbitrary LLM, the detector must be selected from the set $\Gamma^*$.
>
> ---
> **Q2. Shown results for variable-length generation.**
>
> Under our framework, we consider the detection performance given a generated sequence. Thus, our Theorems 1 and 2 hold for fixed-length generation cases. The theoretical analyses for variable-length generation cases would be an interesting future direction. However, although we do not provide theoretical optimality for variable-length generation, we provide a practical scheme that considers variable-length watermark generation and detection in Sections 4 and 5.  In Section 4, we propose a token-level optimal detector and watermarking scheme based on our sequence-level optimal result. This approach effectively adapts to variable-length text generation while maintaining strong performance guarantees, as stated in Lemma 4 in the revised manuscript. In Section 5, experimental results show that our algorithm achieves excellent detection performance and robustness.

---

> ### Author Response · Authors · 2024-11-21
>
> **Q3. Whether constructed optimal watermarking scheme requires knowledge of LLM distributions. If so, is it possible to compute the scheme's parameters efficiently, either practically or within polynomial time?**
>
> Yes, as outlined in Algorithm 1 of our manuscript, our optimal watermarking scheme depends on the Next Token Prediction (NTP) distribution of the LLM. However, the computation of our watermarking scheme remains highly time-efficient, with complexity scaling linearly with the vocabulary size.
> To demonstrate the efficiency of our watermarking method, we conducted experiments measuring the average generation time for both un-watermarked and watermarked text. For these experiments, we employed LLaMA2-13B and the C4 dataset, setting the text length to 200 tokens in both scenarios. The results, presented in the following table, indicate that the difference in generation time between un-watermarked and watermarked text is less than $0.5$ seconds. This minimal difference confirms that our watermarking method has a negligible impact on generation speed, ensuring practical applicability.
> | Language Model | Watermark      | Avg Generation Time/s |
> |----------------|----------------|-----------------------|
> | Llama2-13B     | Un-watermarked | 9.110s                |
> | Llama2-13B     | Watermarked    | 9.386s                |
>
> **Q4. Can you say anything about the optimality of your detector for generations that are not of a fixed length?**
>
> In our response to Q1, we clarify that the optimal type of detectors and watermarking schemes are jointly optimal, meaning that $(\gamma^*, P_{X_1^T,\zeta_1^T}^*)$ is optimal in pair. Therefore, given a $\gamma^*$ within the optimal class, it is not necessarily optimal for an arbitrary watermarking scheme. However, building on our theoretically optimal result, we propose a token-level optimal detector and watermarking scheme in Section 4, which adapts to variable-length generation. This approach ensures optimality at each token while maintaining strong detection performance across the entire sequence.
>
>
> **Q5. Describe the set of optimal detectors $\Gamma^\*$ in simpler terms. (It appears that you're saying that, for any fixed watermarked text generation algorithm of the form you consider, the optimal detector simply tests whether the entire generation is exactly equal to some string computed as a function of \zeta. However, this is clearly false: If \zeta is just a collection of red/green lists for each token position, then the detector should accept many possible texts. I think that what you're trying to say is that there exists a watermarking algorithm such that this detector is optimal, but this is not what appears to be stated in Theorem 2.)**
>
> We are sorry for the confusion. In our response to Q1, we restated Theorem 2 for clarity. The optimal solution to our optimization problem (OPT-O) identifies the **optimal class of detector-watermarking scheme pairs**. Additionally, we prove that detectors within this optimal class, denoted as $\Gamma^*$, are **universally optimal**. Specifically, for any detector $\gamma \notin \Gamma^\*$, there always exists a text distribution $Q$ and a distortion level $\epsilon$ such that no watermarking scheme can achieve the universal minimum Type-II error.  It suggests that to guarantee the construction of an optimal watermarking scheme for any arbitrary LLM, the detector must be selected from the set $\Gamma^*$. Therefore, for any detector in the optimal set  $\Gamma^*$, it only achieves the optimal performance when paired with its corresponding optimal watermarking scheme (construction detail in Appendix D)。

---

> ### Author Response · Authors · 2024-11-21
>
> **Q6. The description of the Kuditipudi et al. watermarking scheme appears to be completely unrelated to the scheme presented in their paper. Their scheme is not hash-based at all.**
>
> Our proposed framework in Section 2 is generic for any in-process watermarking scheme. In the problem formulation, we do not assume that the hash function will be used during watermark generation and detection. The auxiliary sequence $\zeta_1^T$ can be sampled using arbitrary methods. Example 3 presents some existing watermarking schemes, which, to the best of our knowledge, may all be seen as special cases of this formulation. In Appendix A, we are sorry for the description error about Kuditipudi et al. watermarking scheme. We have corrected the error, and it still can be seen as a special case of this formulation.
>
> The corrected version:
> - In the **inverse transform watermarking scheme** \citep{kuditipudi2023robust}, the vocabulary $\mathcal{V}$ is considered as $[|\mathcal{V}|]$ and the combination of the uniform random variable and the randomly permuted index vector is the auxiliary variable $\zeta_t$.
>
> - Use key as a seed to uniformly and independently sample $\lbrace U_t \rbrace_{t=1}^T$ from $[0,1]$, and $\lbrace \pi_t \rbrace_{t=1}^T$ from the space of permutations over $[|\mathcal{V}|]$. Let the auxiliary variable $\zeta_t=(U_t,\pi_t)$, for $t=1,2,\ldots,T$.
>
> - Sample $X_t$ as follows
>
> $X_t=\pi_t^{-1}(\min\{i\in[|\mathcal{V}|]: \sum_{x\in[|\mathcal{V}|]}(Q_{X_t|x_1^{t-1}}(x)\mathbb{1}\lbrace \pi_t(x)\leq i \rbrace \big)  \geq U_t \}),$
>
> where $\pi_t^{-1}$ denotes the inverse permutation.

---

> ### Author Response · Authors · 2024-11-25
> **Follow-up discussion**
>
> Dear reviewer mxQq,
>
> We sincerely appreciate your time and effort in reviewing our submission and providing valuable suggestions. While we hope to have addressed your concerns adequately, we understand there may still be areas requiring further clarification or discussion. We are fully prepared to address your outstanding issues. Should our responses have successfully addressed all your questions, we would be deeply grateful if you could consider enhancing the score to further support our submission. Thank you very much for your thoughtful review.
>
> Best Regards,
>
> Paper12990 Authors

---

> ### Author Response · Authors · 2024-12-02
>
> Dear reviewer mxQq,
>
> We sincerely appreciate your time and constructive feedback, which helps us further improve the quality of our submission. We have made every effort to answer your insightful questions by modifying the manuscript and writing rebuttals. Considering today (Dec 2) is the deadline for the author's response, we hope our responses have addressed the reviewer's concerns, but if not, we are available to address any outstanding issues. In case we have addressed all your questions, we truly appreciate your feedback and reconsider your score for further support.
>
> Best Regards,
>
> Paper12990 Authors

---

> > ### Comment · Reviewer_mxQq · 2024-12-02
> >
> > Thank you for your responses. However, after revisiting the paper I have decided not to update my score.

---

### Meta-Review · Area_Chair_m77K · 2024-12-15

**Metareview:**

This paper addresses designing optimal LLM watermarking schemes to minimize false negative rates under constraints on distortion and false positive rates. For a class of schemes encompassing existing methods, the authors derive an expression for the optimal false negative rate based on the LLM distribution and identify a simple form for optimal detectors, highlighting potential improvements over current approaches. They introduce a token-level, model-agnostic watermarking framework leveraging a surrogate model and the Gumbel-Max trick. Experiments on Llama-13B and Mistral-8×7B show competitive performance and robustness, establishing key trade-offs and paving the way for future adversarial-resistant systems.

The paper has some merits; however, it requires major revisions. For example, as the reviewer pointed out, the correctness of the results on the "average perplexity" remains uncertain. Moreover, the comparison to existing methods, such as SIR, is insufficient. Although the authors responded to the reviewer’s comments, there are many existing methods in LLM watermarking, and it would be better to include more comparisons with these methods and discuss the differences. Therefore, the paper requires significant revisions and cannot be accepted in its current form. I encourage the authors to address the reviewer’s comments and resubmit the paper to a future venue.

**Additional Comments On Reviewer Discussion:**

The reviewers raised concerns about the perplexity calculation and the comparison to existing methods. Regarding the perplexity calculation, the authors responded by referencing other papers; however, it remains not fully certain. For the comparison, the authors evaluated their method against one existing approach. Unfortunately, the reviewer who raised this concern did not provide further feedback. Nonetheless, given the existence of numerous LLM watermarking methods, more comprehensive comparisons and discussions are necessary. Additionally, the positive reviewer scores were only "weak accept," and the overall score of the paper is below the borderline (6.0). Therefore, I have decided to recommend rejection.

---

### Decision · Program_Chairs · 2025-01-22

Reject